# High-Dimensional Tensor Regression with Oracle Properties

**Wenbin Wang** [1]  **Yu Shi** [1]  **Ziping Zhao** [1]

## Abstract

Tensor regression has emerged as a powerful framework for modeling linear relationships among multi-dimensional variables by effectively capturing inherent cross-mode interactions within tensor-structured data. In this paper, we introduce a high-dimensional tensor-response tensor regression model under low-dimensional structural assumptions, such as sparsity and low-rankness. Specifically, we assume that the underlying regression tensor lies within an unknown low-dimensional subspace and propose a general least squares estimation framework with non-convex penalties. Theoretically, we establish rigorous risk bounds for the resulting estimators, demonstrating that they attain the oracle statistical rates under mild technical conditions. To ensure computational efficiency, we introduce a proximal gradient algorithm for solving the proposed non-convex optimization problem. Extensive experiments conducted on both synthetic and real-world datasets validate the effectiveness of the proposed regression model and showcase the practical utility of the theoretical findings.

## 1. Introduction

The tensor, a multi-dimensional array generalizing the matrix to higher dimensions, has become a useful tool in many data analysis areas (Kolda & Bader, 2009; Abraham et al., 2012; McConnell, 2014; McCullagh, 2018), including image analysis (Zhou et al., 2013; Li et al., 2018), biology (Hore et al., 2016), spectroscopy data (Amini et al., 2017), economics and finance (Li et al., 2015a; Wang et al., 2022), business (Hao et al., 2021; Bi et al., 2018), etc. Among tensor-based methods, the tensor regression is especially useful for revealing linear relationships among high-dimensional variable sets and has been successfully applied

in many domains (Han et al., 2022; Liu et al., 2022; Wang et al., 2024). For example, in image processing, tensor regression techniques have addressed critical tasks such as denoising (Zhang et al., 2021a), image inpainting (Bertalmio et al., 2001), and medical image analysis (Zhou et al., 2013; Li & Zhang, 2017). In recommendation systems, tensor regression leverages shared item information, substantially enhancing prediction accuracy and outperforming models that treat items independently (Zhang et al., 2021b). Additionally, in spatio-temporal analysis, tensor regression has been developed to handle both forecasting and cokriging tasks (Bahadori et al., 2014; Yu & Liu, 2016; Yu et al., 2018; Su et al., 2020).

Despite their wide applications, tensor regression models encounter significant challenges in estimation. This problem is more prominent in high-dimensional settings, where the number of parameters substantially exceeds the number of observations, which makes the model estimation ill-posed (Raskutti et al., 2019; Chen et al., 2019). To address such challenges, imposing structural assumptions that capture the underlying low-dimensional structures is critical, such as sparsity or low-rankness (Rabusseau & Kadri, 2016). Sparsity refers to the phenomenon where most entries are either exactly zero or near zero, which is commonly leveraged in fields such as recommendation systems (Lee, 2001) and compressed sensing (Chen et al., 2023). On the other hand, low-rankness indicates the rank is significantly smaller than its informative dimensions, which is frequently applied in areas like image compression (Li & Li, 2010) and collaborative filtering (Li et al., 2017). However, defining these low-dimensional structures for tensors is a key challenge, as the concepts of sparsity and low-rankness have multiple nontrivial extensions in the tensor setting (Kolda & Bader, 2009). For instance, tensor sparsity can be defined either entry-wise or group-wise, such as at the fiber level (Raskutti et al., 2019) or the slice level (Zhang et al., 2019). Similarly, low-rankness can be imposed on different forms of tensors, such as mode-wise (Chen et al., 2019) and slice-wise (Farias & Li, 2017; Luo & Zhang, 2024). Raskutti et al. (2019) investigated all of the previously mentioned sparsity and low-rankness structures, establishing both general risk bounds and specific upper bounds in various scenarios. Alternatively, tensor decomposition techniques, such as Canonical Polyadic (CP) decomposition (Carroll & Chang, 1970) and

[1] School of Information Science and Technology, ShanghaiTech University, Shanghai 201210, China. Correspondence to: Ziping Zhao <zipingzhao@shanghaitech.edu.cn>.

*Proceedings of the 42nd International Conference on Machine Learning*, Vancouver, Canada. PMLR 267, 2025. Copyright 2025 by the author(s).

Tucker decomposition (Tucker, 1966), offer another way to enforce structure. However, these methods often face challenges due to their nonconvex nature of the optimization problems involved (Luo & Zhang, 2023). To further reduce the number of estimated parameters and improve model performance, several studies argue for decomposition-based estimators with regularizers (He et al., 2018; Ahmed et al., 2020; Xu, 2020).

In this paper, we investigate a high-dimensional tensor-on-tensor regression model and propose to estimate the coefficient with penalty regularizers. While convex methods, such as Lasso (Tibshirani, 1996) and nuclear norm minimization (Recht et al., 2010; Candes & Recht, 2012), are widely employed due to their strong theoretical guarantees (Zhang et al., 2019; Raskutti et al., 2019), nonconvex approaches have recently garnered attention for their advantages in unbiased estimation and improved theoretical properties in high-dimensional settings. Although nonconvex regularizers are widely used, their benefits for high-dimensional tensor regression problems remain unclear. This paper closes this gap by proposing a general and unifying estimation framework. In particular, we discuss a class of decomposable nonconvex penalty functions, including the smoothly clipped absolute deviation (SCAD) (Fan & Li, 2001) and the minimax concave penalty (MCP) (Zhang, 2010). Leveraging these univariate penalty functions, we impose distinct low-dimensional structures—sparsity or low-rank constraints—on the regression coefficient tensor. To effectively solve the resulting optimization problems, we present a proximal gradient algorithm. Our analysis shows that these estimators enjoy oracle properties under mild assumptions which is faster than the estimators (Raskutti et al., 2019). Extensive numerical experiments on both synthetic data and real-world datasets validate the theoretical results and demonstrate the practical advantages and breadth of the proposed methods. Proofs are deferred to the Supplementary Materials.

## 2. Preliminary

Throughout the paper, we use boldface calligraphy letters for tensors, such as $\boldsymbol{\mathcal{A}}$, boldface uppercase letters for matrices, such as $\boldsymbol{A}$, and standard lowercase letters for scalars, such as $x$. The order (or degree) of a tensor is defined as the number of modes it has; hence, matrices, vectors, and scalars are order-2, order-1, and order-0 tensors, respectively. For an order-$N$ tensor $\boldsymbol{\mathcal{A}} \in \mathbb{R}^{d_1 \times \cdots \times d_N}$, where the mode-$i$ dimension is $d_i$, $i \in \{1, \ldots, N\}$, the entry of $\boldsymbol{\mathcal{A}}$ at position $(i_1, \ldots, i_N)$ is denoted by $a_{i_1, \ldots, i_N}$ or $[\boldsymbol{\mathcal{A}}]_{i_1, \ldots, i_N}$.

**Fibers and Slices** By fixing all indices of a tensor $\boldsymbol{\mathcal{A}}$ except for the $k$-th mode, we obtain mode-$(k)$ fibers, which are vectors. Mode-$(j, k)$ slices are obtained by fixing all indices except for the $j$-th and $k$-th modes, resulting in

matrices.

**Tensor Unfolding** A tensor $\boldsymbol{\mathcal{A}}$ with order higher than 3 can be unfolded into an order-3 tensor along the $(j, k)$-th mode, denoted as $\boldsymbol{\mathcal{A}}_{(j,k)} \in \mathbb{R}^{d_j \times d_k \times \prod_{s \neq j, k} d_s}$. This unfolding arranges the mode-$(j, k)$ slices of $\boldsymbol{\mathcal{A}}$ as the frontal slices of $\boldsymbol{\mathcal{A}}_{(j,k)}$. Specifically, its entry satisfy $\left[\boldsymbol{\mathcal{A}}_{(j,k)}\right]_{i_j, i_k, l} = a_{i_1, \ldots, i_N}$, where $l = 1 + \sum_{s=1, s \neq j, s \neq k}^{N} (i_s - 1) \prod_{m=1, m \neq j, m \neq k}^{s-1} d_m$. A tensor can also be unfolded into a matrix, which is also known as tensor matricization or flattening. The mode-$(k)$ unfolding of $\boldsymbol{\mathcal{A}}$ is denoted by $\boldsymbol{A}_{(k)} \in \mathbb{R}^{d_k \times \prod_{i \neq k} d_i}$, where each column corresponds to a mode-$(k)$ fiber of $\boldsymbol{\mathcal{A}}$. Specifically, the entries of $\boldsymbol{A}_{(k)}$ satisfy $\left[\boldsymbol{A}_{(k)}\right]_{i_k, l} = a_{i_1, \ldots, i_k, \ldots, i_N}$, where $l = 1 + \sum_{s=1, s \neq k}^{N} (i_s - 1) \prod_{m=1, m \neq k}^{s-1} d_m$. Finally, a tensor can also be reshaped into a vector, an operation commonly known as tensor vectorization. We denote the vectorization of $\boldsymbol{\mathcal{A}}$ as $\text{vec}(\boldsymbol{\mathcal{A}})$, which is equivalent to vectorizing its mode-1 unfolding: $\text{vec}(\boldsymbol{\mathcal{A}}) = \text{vec}(\boldsymbol{A}_{(1)})$.

We use calligraphic letters to represent sets, such as $\mathcal{S}$. The support of a set $\mathcal{S}$ is denoted by $|\mathcal{S}|$. $\Pi_{\mathcal{S}}(\cdot)$ denotes the projection onto the set $\mathcal{S}$. For a function $f$, $f'$ denotes its derivative, $\nabla f$ represents its gradient, and $\nabla^2 f$ denotes its Hessian. For functionals $f(x)$ and $g(x)$, we denote $f(x) \gtrsim g(x)$ if $f(x) \geq cg(x)$, $f(x) \lesssim g(x)$ if $f(x) \leq Cg(x)$, and $f(x) \asymp g(x)$ if $cg(x) \leq f(x) \leq Cg(x)$ for some positive constants $c$ and $C$.

## 3. Problem Formulation

In this section, we present a unified framework for high-dimensional tensor regression with nonconvex regularizers.

### 3.1. Tensor Regression

We consider the following generic tensor-on-tensor regression model (Lock, 2018; Raskutti et al., 2019; Miao et al., 2022) with tensor coefficient $\boldsymbol{\mathcal{A}}^\star \in \mathbb{R}^{d_1 \times \cdots \times d_N}$:

$$\boldsymbol{\mathcal{Y}} = \langle \boldsymbol{\mathcal{X}}, \boldsymbol{\mathcal{A}}^\star \rangle + \boldsymbol{\mathcal{E}}, \tag{1}$$

where $\boldsymbol{\mathcal{X}} \in \mathbb{R}^{d_1 \times \cdots \times d_M}$ is the predictor variable with $M \leq N$, $\boldsymbol{\mathcal{Y}} \in \mathbb{R}^{d_{M+1} \times \cdots \times d_N}$ is the response variable, and $\boldsymbol{\mathcal{E}} \in \mathbb{R}^{d_{M+1} \times \cdots \times d_N}$ is the noise. $\langle \cdot, \cdot \rangle$ is the tensor contraction product between two tensors, where its $(i_{M+1}, \ldots, i_N)$-th entry is defined as

$$[\langle \boldsymbol{\mathcal{X}}, \boldsymbol{\mathcal{A}}^\star \rangle]_{i_{M+1}, \ldots, i_N}$$
$$= \sum_{i_1=1}^{d_1} \cdots \sum_{i_M=1}^{d_M} x_{i_1, \ldots, i_M} a^\star_{i_1, \ldots, i_M, i_{M+1}, \ldots, i_N}.$$

Specifically, when $M = N$, the output is a scalar, in which case (1) becomes a scalar-on-tensor regression model (Zhou et al., 2013; Gui et al., 2016).

## 3.2. Proposed Problem

Given a collection of $n$ samples $\left\{ \left( \boldsymbol{\mathcal{Y}}^{(i)}, \boldsymbol{\mathcal{X}}^{(i)} \right) \right\}_{i=1}^{n}$, which is assumed to be generated from the observation model (1), our goal is to estimate the unknown coefficient tensor $\boldsymbol{\mathcal{A}}^{\star}$ by solving the following regularized least squares estimation problem:

$$\min_{\boldsymbol{\mathcal{A}} \in \mathbb{R}^{d_1 \times \cdots \times d_N}} \left\{ \frac{1}{2n} \sum_{i=1}^{n} \left\| \boldsymbol{\mathcal{Y}}^{(i)} - \left\langle \boldsymbol{\mathcal{X}}^{(i)}, \boldsymbol{\mathcal{A}} \right\rangle \right\|_{\mathrm{F}}^{2} + \mathcal{R}_{\lambda}\left(\boldsymbol{\mathcal{A}}\right) \right\},$$
(2)

where $\mathcal{R}_{\lambda}\left(\boldsymbol{\mathcal{A}}\right)$ is a structural regularization term. For a tensor $\boldsymbol{\mathcal{A}}$, $\|\boldsymbol{\mathcal{A}}\|_{\mathrm{F}} = \langle \boldsymbol{\mathcal{A}}, \boldsymbol{\mathcal{A}} \rangle^{\frac{1}{2}}$.

## 3.3. Nonconvex Regularization

In this paper, we consider a class of regularizers $\mathcal{R}_{\lambda}\left(\boldsymbol{\mathcal{A}}\right)$, which is defined based on a nonconvex penalty function $p_{\lambda}(\cdot)$ with parameter $\lambda \geq 0$. Prototype examples of such regularizers include the SCAD (Fan & Li, 2001) and MCP (Zhang, 2010), and we introduce them in the following.

The SCAD penalty function, proposed by (Fan & Li, 2001), is defined as

$$p_{\lambda}\left(t\right) = \begin{cases} \lambda|t|, & \text{for}|t| \leq \lambda, \\ -\frac{t^2 - 2a\lambda|t| + \lambda^2}{2(a-1)}, & \text{for}\lambda < |t| \leq a\lambda, \\ \frac{(a+1)\lambda^2}{2}, & \text{for}|t| > a\lambda, \end{cases}$$

where $a > 2$ is a tuning parameter. The MCP penalty, introduced by (Zhang, 2010), is defined as

$$p_{\lambda}\left(t\right) = \begin{cases} \lambda|t| - \frac{t^2}{2a}, & |t| \leq a\lambda, \\ \frac{a\lambda^2}{2}, & t > a\lambda, \end{cases}$$

for some constant $a > 1$. Note that both the above nonconvex functions $p_{\lambda}(t)$ can be decomposed as $p_{\lambda}(t) = \lambda|t| + q_{\lambda}(t)$, where $|t|$ denotes the absolute value of $t$ and $q_{\lambda}(t)$ is a concave function. Specifically, for SCAD, $q_{\lambda}(t)$ is given by

$$q_{\lambda}\left(t\right) = \frac{-\left(|t| + \lambda\right)^2}{2\left(a-1\right)} \mathbf{1} \left(\lambda \leq |t| \leq a\lambda\right) \\ + \left(\frac{1}{2\left(a+1\right)\lambda^2} - \lambda|t|\right) \mathbf{1} \left(|t| \geq a\lambda\right),$$

where $\mathbf{1}\left(\cdot\right)$ denotes the indicator function; for MCP, $q_{\lambda}(t)$ is

$$q_{\lambda}\left(t\right) = -\frac{t^2}{2a} \mathbf{1} \left(|t| \leq a\lambda\right) + \left(\frac{a\lambda^2}{2} - \lambda|t|\right) \mathbf{1} \left(|t| \geq a\lambda\right).$$

Furthermore, we impose the following regularity conditions on $p_{\lambda}\left(\cdot\right)$ and $q_{\lambda}\left(\cdot\right)$, as detailed in Assumption 1.

**Assumption 1.** *The functions $p_{\lambda}(t)$ and $q_{\lambda}(t)$ satisfy the following conditions:*

- *There exists a constant $\nu > 0$ such that the penalty function satisfies $p_{\lambda}'(t) = 0$ for all $t \geq \nu$;*

- *$q_{\lambda}(t)$ is symmetric, i.e., $q_{\lambda}(-t) = q_{\lambda}(t)$ for all $t$;*

- *$q_{\lambda}'(t)$ is monotone and Lipschitz continuous, i.e., for $t_2 \geq t_1$, there exists a nonnegative constant $\zeta$ such that $-\zeta \leq \frac{q_{\lambda}'(t_2) - q_{\lambda}'(t_1)}{t_2 - t_1} \leq 0$;*

- *$q_{\lambda}(t)$ and $q_{\lambda}'(t)$ pass through the origin, i.e. $q_{\lambda}(0) = q_{\lambda}'(0) = 0$;*

- *There exists a positive constant $\lambda$ such that $|q_{\lambda}'(t)| \leq \lambda$ for all $t$.*

Such conditions are commonly employed in the analysis of nonconvex statistical estimation problems (Wang et al., 2014; Gui et al., 2016; Fan et al., 2018). The third condition introduces a curvature property that governs the degree of concavity of $q_{\lambda}(\cdot)$, and consequently, the level of nonconvexity of $p_{\lambda}(\cdot)$. For SCAD, these conditions are satisfied with $\nu = a\lambda$ and $\zeta = \frac{1}{a-1}$, while for MCP, we have $\nu = a\lambda$ and $\zeta = \frac{1}{a}$.

### 3.3.1. SPARSITY REGULARIZATION

A straightforward approach to induce sparsity within a tensor $\boldsymbol{\mathcal{A}}$ is to enforce entry-wise sparsity. This strategy draws inspiration from the well-known Lasso regression (Tibshirani, 1996) and has been extensively studied using convex regularization methods (Zhang et al., 2019; Raskutti et al., 2019). In contrast to prior works, we employ a nonconvex penalty. Specifically, the entry-wise sparsity regularizer is defined as

$$\mathcal{R}_{\lambda}\left(\boldsymbol{\mathcal{A}}\right) = \sum_{i_1=1}^{d_1} \cdots \sum_{i_N=1}^{d_N} p_{\lambda}\left(a_{i_1,\ldots,i_N}\right).$$

In addition to promoting entry-wise sparsity, this regularization framework can be extended to incorporate other forms of sparsity in a general $N$-mode tensor, such as fiber-wise sparsity and slice-wise sparsity. Further formulations and discussions of these alternative sparsity-inducing regularizations are presented in Appendix A.

### 3.3.2. LOW-RANKNESS REGULARIZATION

In addition to promoting sparsity, encouraging low-rank structure in tensors has demonstrated significant benefits in various applications (Nion & Sidiropoulos, 2010; Li & Li, 2010; Collins & Cohen, 2012; Semerci et al., 2014). There are multiple notions of rank for higher-order tensors (Kolda & Bader, 2009). In this section, we focus on mode-wise low-rankness, which involves penalizing the singular values of the mode-$(k)$ unfoldings. However, the commonly used tensor nuclear norm penalty, which applies the $\ell_1$ norm

to the singular values of the unfolded matrices, inevitably introduces a non-negligible bias (Raskutti et al., 2019). To alleviate this issue, we propose using a nonconvex penalty applied to each singular value. Specifically, the mode-wise low-rankness regularizer is defined as

$$\mathcal{R}_\lambda\left(\boldsymbol{\mathcal{A}}\right) = \sum_{i=1}^{\min\left\{d_k, \prod_{j \neq k} d_j\right\}} p_\lambda\left(\sigma_i\left(\boldsymbol{A}_{(k)}\right)\right),$$

where $\sigma_i\left(\boldsymbol{A}_{(k)}\right)$ denotes the $i$-th singular value of $\boldsymbol{A}_{(k)}$. Additionally, this framework can be extended to accommodate alternative forms of low-rankness regularization. Further problem formulations and theoretical analyses of these extensions are provided in Appendix A.

## 4. Main Theory

In this section, we present the theoretical results for the estimators from (2) under different scenarios and derive their corresponding estimation error bounds. We begin by presenting some preliminary assumptions.

### 4.1. Preliminaries

To facilitate the subsequent discussion, we define a local region as

$$\mathcal{C} = \left\{\boldsymbol{\mathcal{B}} \in \mathbb{R}^{d_1 \times \cdots \times d_N} \mid \mathcal{D}\left(\boldsymbol{\mathcal{B}}, \boldsymbol{\mathcal{A}}^\star\right) \leq r\right\},$$

where $\mathcal{D}\left(\cdot\right)$ denotes a distance function. For example, in the discussion of entry-wise sparsity,

$$\mathcal{D}\left(\boldsymbol{\mathcal{B}}, \boldsymbol{\mathcal{A}}^\star\right) := \left\|\boldsymbol{\mathcal{B}} - \boldsymbol{\mathcal{A}}^\star\right\|_{\mathrm{F}};$$

and in the discussion of mode-wise low-rankness, we define

$$\mathcal{D}\left(\boldsymbol{\mathcal{B}}, \boldsymbol{\mathcal{A}}^\star\right) := \frac{\left\|\Pi_{\mathcal{F}_{\boldsymbol{\mathcal{A}}^\star}^\perp}\left(\boldsymbol{\mathcal{B}}\right)\right\|_{\mathrm{nuc}}}{\left\|\Pi_{\mathcal{F}_{\boldsymbol{\mathcal{A}}^\star}}\left(\boldsymbol{\mathcal{B}}\right)\right\|_{\mathrm{nuc}}},$$

where $\mathcal{F}_{\boldsymbol{\mathcal{A}}^\star}$ is a subspace associated with the unfolding of $\boldsymbol{\mathcal{A}}^\star$ (to be defined explicitly later), $\mathcal{F}_{\boldsymbol{\mathcal{A}}^\star}^\perp$ is its orthogonal complement, and $\|\cdot\|_{\mathrm{nuc}}$ denotes the nuclear norm, i.e., $\|\cdot\|_{\mathrm{nuc}} = \sum_i \sigma_i(\cdot)$.

Define the empirical loss function as $\mathcal{L}\left(\boldsymbol{\mathcal{A}}\right) = \frac{1}{2n} \sum_{i=1}^n \left\|\boldsymbol{\mathcal{Y}}^{(i)} - \left\langle \boldsymbol{\mathcal{X}}^{(i)}, \boldsymbol{\mathcal{A}}\right\rangle\right\|_{\mathrm{F}}^2$. We make the following assumptions on this loss function.

**Assumption 2** (Restricted strong convexity (RSC)). *For any $\boldsymbol{\mathcal{A}}, \boldsymbol{\mathcal{B}} \in \mathcal{C}$, there exists a constant $\mu$ satisfying $\mu > 0$ such that*

$$\mathcal{L}\left(\boldsymbol{\mathcal{B}}\right) \geq \mathcal{L}\left(\boldsymbol{\mathcal{A}}\right) + \left\langle \nabla\mathcal{L}\left(\boldsymbol{\mathcal{A}}\right), \boldsymbol{\mathcal{B}} - \boldsymbol{\mathcal{A}}\right\rangle + \frac{\mu}{2}\|\boldsymbol{\mathcal{B}} - \boldsymbol{\mathcal{A}}\|_{\mathrm{F}}^2.$$

**Assumption 3** (Restricted smoothness (RSM)). *For any $\boldsymbol{\mathcal{A}}, \boldsymbol{\mathcal{B}} \in \mathcal{C}$, there exists a constant $L$ satisfying $L > 0$ such that*

$$\mathcal{L}\left(\boldsymbol{\mathcal{B}}\right) \leq \mathcal{L}\left(\boldsymbol{\mathcal{A}}\right) + \left\langle \nabla\mathcal{L}\left(\boldsymbol{\mathcal{A}}\right), \boldsymbol{\mathcal{B}} - \boldsymbol{\mathcal{A}}\right\rangle + \frac{L}{2}\|\boldsymbol{\mathcal{B}} - \boldsymbol{\mathcal{A}}\|_{\mathrm{F}}^2.$$

Assumptions 2 and 3, which characterize the curvature properties of the empirical loss function $\mathcal{L}$, are analogous to the classical RSC and RSM conditions commonly used in the literature on linear regression problems (Wang et al., 2014; Gui et al., 2016; Elenberg et al., 2018). If Assumptions 2 and 3 hold at the same time, it implies that $L \geq \mu$. Leveraging the methodology of (Candes & Tao, 2007), it can be proven that the empirical loss function $\mathcal{L}$ satisfies both the RSC and RSM conditions with high probability.

**Assumption 4.** *Assume that the concatenation of vectorized covariates from $n$ samples, denoted as $[\mathrm{vec}(\boldsymbol{\mathcal{X}}^{(1)})^\top, \ldots, \mathrm{vec}(\boldsymbol{\mathcal{X}}^{(n)})^\top]$, follows a multivariate Gaussian distribution with zero mean and covariance matrix $\boldsymbol{\Sigma}$. We assume that there exist constants $\kappa \geq 1$ such that the eigenvalues of $\boldsymbol{\Sigma}$ satisfy:*

$$\kappa^{-1} \leq \sigma_{\min}\left(\boldsymbol{\Sigma}\right) \leq \sigma_{\max}\left(\boldsymbol{\Sigma}\right) \leq \kappa.$$

Assumption 4 ensures that the covariance matrix $\boldsymbol{\Sigma}$ is positive definite and well-conditioned, which is crucial for avoiding degeneracies in the parameter space. Such conditions are commonly met in a range of statistical estimation problems (Liu et al., 2014; Raskutti et al., 2019; Wei & Zhao, 2023). If the covariates $\left\{\boldsymbol{\mathcal{X}}^{(i)}\right\}_{i=1}^n$ are independent and identically distributed, the covariance matrix $\boldsymbol{\Sigma}$ is block-diagonal, and Assumption 4 reduces to similar conditions on the covariance matrix of each individual sample.

### 4.2. Statistical Error Analysis

In the following, we establish statistical error bounds for different regularizers $\mathcal{R}_\lambda\left(\cdot\right)$, providing insights into their performance under different conditions.

#### 4.2.1. SPARSITY REGULARIZATION

Before presenting a detailed analysis of the convergence rates associated with the entry-wise sparsity regularizer, we first introduce the notion of the oracle rate. The oracle rate refers to the statistical convergence rate achieved by the oracle estimator, which serves as an idealized benchmark under the assumption that the true parameter support is known a priori. This assumption allows the oracle estimator to attain the best possible theoretical performance.

Assuming the true parameter support set for the entry-wise sparsity is $\mathcal{S}_1$, the entry-wise sparse oracle estimator is defined as

$$\widehat{\boldsymbol{\mathcal{A}}}^O = \arg\min_{\boldsymbol{\mathcal{A}}:\boldsymbol{\mathcal{A}}_{\overline{\mathcal{S}_1}} = \boldsymbol{0}} \mathcal{L}\left(\boldsymbol{\mathcal{A}}\right),$$

where $\overline{\mathcal{S}_1}$ denotes the complement of the support set $\mathcal{S}_1 = \left\{(i_1, \ldots, i_N) \mid a_{i_1, \ldots, i_N}^\star \neq 0\right\}$. By the mean value theorem, it is easy to obtain that $\widehat{\boldsymbol{\mathcal{A}}}^O$ satisfies $\|\widehat{\boldsymbol{\mathcal{A}}}^O - \boldsymbol{\mathcal{A}}^\star\|_{\mathrm{F}} \lesssim$

$\left\| \nabla \mathcal{L} \left( \mathcal{A}^\star \right)_{\mathcal{S}_1} \right\|_{\mathrm{F}} \lesssim \sqrt{\frac{|\mathcal{S}_1|}{n}}$. Now, we have the following result.

**Theorem 5** (Entry-wise sparsity)**.** *Suppose that Assumptions 1∼4 hold. If $\mu > \zeta$,*   *$\lambda \asymp \sqrt{\frac{\log(d_1 d_2 \cdots d_M)}{n}}$, and the true parameter tensor $\mathcal{A}^\star$ satisfies*

$$\min_{(i_1,\ldots,i_N)\in\mathcal{S}_1} \left| a^\star_{i_1,\ldots,i_N} \right| \geq \nu, \tag{3}$$

*then the optimal solution $\widehat{\mathcal{A}}$ to problem* (2) *satisfies*

$$\left\| \widehat{\mathcal{A}} - \mathcal{A}^\star \right\|_{\mathrm{F}} \lesssim \sqrt{\frac{|\mathcal{S}_1|}{n}}.$$

Theorem 5 implies that the proposed estimator achieves the oracle rate under relatively mild assumptions. This performance is superior to that of the existing estimator, which uses an $\ell_1$ norm penalty (Raskutti et al., 2019). In fact, condition (3) is referred to as the minimum signal strength condition, and $\nu$ denotes the minimum signal strength. This condition is commonly employed in the analysis of nonconvex penalized regression problems (Fan & Li, 2001; Zhang, 2010; Fan et al., 2018), and it is considered rather mild because in our analysis, we take $\nu \asymp \lambda$ to be of the order $\sqrt{\frac{\log(d_1 d_2 \cdots d_M)}{n}}$, which can be very small as the sample size $n$ increases.

### 4.2.2. Low-rankness Regularization

Consider the matrix $\boldsymbol{X} \in \mathbb{R}^{m \times n}$ of rank $r$. Its singular value decomposition is given by $\boldsymbol{X} = \boldsymbol{U}\boldsymbol{\Sigma}\boldsymbol{V}^\top$, where $\boldsymbol{U} \in \mathbb{R}^{m \times r}$ contains the left singular vectors, $\boldsymbol{V} \in \mathbb{R}^{n \times r}$ contains the right singular vectors, and $\boldsymbol{\Sigma} = \mathrm{Diag}\left(\sigma_1\left(\boldsymbol{X}\right), \ldots, \sigma_r\left(\boldsymbol{X}\right)\right) \in \mathbb{R}^{r \times r}$ is a diagonal matrix of the singular values. We further define a subspace $\mathcal{F}\left(\boldsymbol{X}\right)$ and its orthogonal complement $\mathcal{F}^\perp\left(\boldsymbol{X}\right)$ as follows:[1]

$$\mathcal{F}\left(\boldsymbol{X}\right) = \left\{ \boldsymbol{W} \mid \mathrm{span}\left(\boldsymbol{W}\right) \subseteq \boldsymbol{U}, \mathrm{span}\left(\boldsymbol{W}^\top\right) \subseteq \boldsymbol{V} \right\},$$

$$\mathcal{F}^\perp\left(\boldsymbol{X}\right) = \left\{ \boldsymbol{W} \mid \mathrm{span}\left(\boldsymbol{W}\right) \perp \boldsymbol{U}, \mathrm{span}\left(\boldsymbol{W}^\top\right) \perp \boldsymbol{V} \right\}.$$

where $\mathrm{span}(\boldsymbol{W})$ denotes the subspace spanned by the columns of $\boldsymbol{W}$. The projection operators onto the subspace $\mathcal{F}$ and its orthogonal complement $\mathcal{F}^\perp$ are defined as follows:

$$\Pi_\mathcal{F}\left(\boldsymbol{X}\right) = \boldsymbol{U}\boldsymbol{U}^\top \boldsymbol{X} \boldsymbol{V}\boldsymbol{V}^\top,$$

$$\Pi_{\mathcal{F}^\perp}\left(\boldsymbol{X}\right) = \left(\boldsymbol{I} - \boldsymbol{U}\boldsymbol{U}^\top\right) \boldsymbol{X} \left(\boldsymbol{I} - \boldsymbol{V}\boldsymbol{V}^\top\right),$$

where $\boldsymbol{I}$ denotes the identity matrix with contextually appropriate dimensions. Additionally, we introduce the linear

---

[1]For brevity, we adopt the shorthand notations $\mathcal{F}$ and $\mathcal{F}^\perp$ when the dependence on $\boldsymbol{X}$ is clear from the context.

operator $\mathfrak{X}\left(\mathcal{A}\right): \mathbb{R}^{d_1 \times \cdots \times d_N} \to \mathbb{R}^{n \times d_{M+1} \times \cdots \times d_N}$, defined as

$$\mathfrak{X}(\mathcal{A}) = \left[ \langle \boldsymbol{\mathcal{X}}^{(1)}, \mathcal{A} \rangle, \ldots, \langle \boldsymbol{\mathcal{X}}^{(n)}, \mathcal{A} \rangle \right],$$

along with its adjoint operator $\mathfrak{X}^*\left(\boldsymbol{\mathcal{E}}^{(1:n)}\right)$ : $\mathbb{R}^{n \times d_{M+1} \times \cdots \times d_N} \to \mathbb{R}^{d_1 \times \cdots \times d_N}$, which is defined as $\mathfrak{X}^*\left(\boldsymbol{\mathcal{E}}^{(1:n)}\right) = \sum_{i=1}^n \boldsymbol{\mathcal{E}}^{(i)} \oplus \boldsymbol{\mathcal{X}}^{(i)}$, where $\left(\boldsymbol{\mathcal{E}} \oplus \boldsymbol{\mathcal{X}}\right)_{i_1,\ldots,i_M,i_{M+1},\ldots,i_N} = e_{i_1,\ldots,i_M} x_{i_{M+1},\ldots,i_N}$.

Then, we introduce the oracle statistical convergence rate of the mode-wise low-rank estimator, which is assumed to know the true singular value support $\mathcal{S}_2 = \left\{ i \mid \sigma_i \left( \Pi_\mathcal{F}\left(A^\star_{(k)}\right) \right) \neq 0 \right\}$ in advance. Specifically, the mode-wise low-rank oracle estimator is defined as

$$\widehat{\mathcal{A}}^O = \arg \min_{\mathcal{A}: \mathcal{A}_{\overline{\mathcal{S}_2}} = \mathbf{0}} \mathcal{L}\left(\mathcal{A}\right).$$

By the mean value theorem, it is easy to obtain that $\widehat{\mathcal{A}}^O$ satisfies $\|\widehat{\mathcal{A}}^O - \mathcal{A}^\star\|_{\mathrm{F}} \lesssim \|\nabla \mathcal{L}\left(\mathcal{A}^\star\right)_{\mathcal{S}_2}\|_{\mathrm{F}} \lesssim \sqrt{\frac{|\mathcal{S}_2|}{n}}$. Then, we have the following result.

**Theorem 6** (Mode-wise low-rankness)**.** *Suppose that Assumptions 1∼4 hold.   If $\mu > \zeta$,   $\lambda \gtrsim \frac{\sqrt{|\mathcal{S}_2|}}{n} \left\| \left[ \mathfrak{X}^*\left(\boldsymbol{\mathcal{E}}^{(1:n)}\right) \right]_{(k)} \right\|_{\mathrm{sp}}$, where $\| \cdot \|_{\mathrm{sp}}$ is the spectral norm, and the true parameter tensor $\mathcal{A}^\star$ satisfies*

$$\min_i \sigma_i(A^\star_{(k)}) \geq \nu + \frac{2\sqrt{|\mathcal{S}_2|}}{n\mu} \|[\mathfrak{X}^*(\boldsymbol{\mathcal{E}}^{(1:n)})]_{(k)}\|_{\mathrm{sp}},$$

*then the optimal solution $\widehat{\mathcal{A}}$ to problem* (2) *satisfies*

$$\left\| \widehat{\mathcal{A}} - \mathcal{A}^\star \right\|_{\mathrm{F}} \lesssim \frac{\tau_k \sqrt{|\mathcal{S}_2|}}{n} \asymp \frac{\sqrt{|\mathcal{S}_2|}}{n},$$

*where $\tau_k = \left\| \Pi_\mathcal{F}\left( \left[ \mathfrak{X}^*\left(\boldsymbol{\mathcal{E}}^{(1:n)}\right) \right]_{(k)} \right) \right\|_{\mathrm{sp}}$.*

This result suggests that, with an appropriately chosen regularization parameter $\lambda$ and provided that the smallest nonzero singular value is sufficiently large, the estimator will achieve the same rate as the oracle estimator.

## 5. Optimization Algorithm

In this section, we present an accelerated proximal gradient algorithm, adapted from (Yao et al., 2017), for solving the estimation problem (2). The algorithm iteratively combines a gradient descent step on the loss function $\mathcal{L}(\mathcal{A})$ with a proximal step on the regularization term $\mathcal{R}_\lambda(\mathcal{A})$. To accelerate convergence, we incorporate an extrapolation step based on Nesterov's accelerated gradient method (Nesterov, 2013).

**Algorithm 1** Accelerated Proximal Gradient Algorithm

**Require:** $\eta \in (0, \frac{1}{L}), \delta \in (0, \frac{1}{\eta} - L), \lambda$;

1   $\boldsymbol{\mathcal{A}}_0 = \boldsymbol{\mathcal{A}}_1 = \boldsymbol{0}$;
2   $t = 1$;
3   **repeat**
4     $\boldsymbol{\mathcal{Y}}_t = \boldsymbol{\mathcal{A}}_t + \frac{t-1}{t+2}(\boldsymbol{\mathcal{A}}_t - \boldsymbol{\mathcal{A}}_{t-1})$;
5     $\Delta_t = \max_{s=\max(1,t-q),\ldots,t} \mathcal{L}(\boldsymbol{\mathcal{A}}_t) + \mathcal{R}_\lambda(\boldsymbol{\mathcal{A}}_t)$;
6     **if** $\mathcal{L}(\boldsymbol{\mathcal{Y}}_t) + \mathcal{R}_\lambda(\boldsymbol{\mathcal{Y}}_t) \leq \Delta_t$ **then**
7       $\boldsymbol{\mathcal{V}}_t = \boldsymbol{\mathcal{Y}}_t$;
8     **else**
9       $\boldsymbol{\mathcal{V}}_t = \boldsymbol{\mathcal{A}}_t$;
10    **end**
11    $\boldsymbol{\mathcal{V}}_t = \boldsymbol{\mathcal{A}}_t + \frac{t-1}{t+2}(\boldsymbol{\mathcal{A}}_t - \boldsymbol{\mathcal{A}}_{t-1})$;
12    $\boldsymbol{\mathcal{Z}}_t = \boldsymbol{\mathcal{V}}_t - \eta \nabla \mathcal{L}(\boldsymbol{\mathcal{V}}_t)$;
13    $\boldsymbol{\mathcal{A}}_t = \text{prox}_\lambda(\boldsymbol{\mathcal{Z}}_t)$
14 **until** $\omega_\lambda(\boldsymbol{\mathcal{A}}_t) \leq \epsilon$;
    **Output:** $\boldsymbol{\mathcal{A}}_{T+1}$

**Accelerated Proximal Gradient Algorithm**   Define the proximal operator associated with $\mathcal{R}_\lambda(\cdot)$ as

$$\text{prox}_\lambda(\boldsymbol{\mathcal{V}}) = \arg\min_{\boldsymbol{\mathcal{X}}} \frac{1}{2} \|\boldsymbol{\mathcal{X}} - \boldsymbol{\mathcal{V}}\|_{\text{F}}^2 + \mathcal{R}_\lambda(\boldsymbol{\mathcal{X}}).$$

Let $\{\boldsymbol{\mathcal{X}}_t\}$ be the sequence of iterates. At the iteration $t$, the algorithm updates $\boldsymbol{\mathcal{X}}_{t+1}$ through the following steps:

$$\boldsymbol{\mathcal{Y}}_t = \boldsymbol{\mathcal{X}}_t + \theta_t(\boldsymbol{\mathcal{X}}_t - \boldsymbol{\mathcal{X}}_{t-1}),$$
$$\boldsymbol{\mathcal{X}}_{t+1} = \text{prox}_\lambda(\boldsymbol{\mathcal{Y}}_t - \eta \nabla \mathcal{L}(\boldsymbol{\mathcal{Y}}_t)),$$

where $\theta_t = \frac{t-1}{t+2}$ and $\eta$ is the stepsize (Beck & Teboulle, 2009; Nesterov, 2013). When $\theta_t = 0$, the algorithm reduces to the standard proximal gradient algorithm. However, in the non-convex tensor setting, the extrapolation step $\boldsymbol{\mathcal{Y}}_t$ can prevent a guaranteed sufficient decrease of the objective function in each iteration. To address this issue, Yao et al. (2017) proposed to evaluate the objective value before the proximal step, instead of checking after it as was previously done, demonstrating that this modification guarantees a sufficient decrease under certain mild conditions. To effectively implement the algorithm, we need to choose an appropriate step size $\eta$. The parameter $\eta$ is related to the Lipschitz constant $L$ of $\mathcal{L}(\cdot)$ (i.e., $\|\nabla\mathcal{L}(\boldsymbol{\mathcal{X}}) - \nabla\mathcal{L}(\boldsymbol{\mathcal{Y}})\|_{\text{F}} \leq L\|\boldsymbol{\mathcal{X}} - \boldsymbol{\mathcal{Y}}\|_{\text{F}}$). Choosing $\eta \leq 1/L$ ensures that the gradient step does not overshoot, which is critical for the convergence of the algorithm. The complete procedure is summarized in Algorithm 1. In line 5 of Algorithm 1, $q$ is an adjustable parameter that governs nonmonotonic behavior of the objective sequences. Setting $q = 0$ represents the simplest approach enforcing a strictly monotone objective sequence, while a larger $q$ permits the objective function $\mathcal{L}(\boldsymbol{\mathcal{Y}}_t) + \mathcal{R}_\lambda(\boldsymbol{\mathcal{Y}}_t)$ to occasionally increase. This flexibility is inspired by the Barzilai-Borwein scheme for unconstrained smooth minimization (Grippo & Sciandrone, 2002). In our experiments,

we adopt $q = 5$ to balance the convergence speed and stability.

**Optimality Conditions**   Recall that the non-convex penalty $p_\lambda(t)$ can be decomposed as the sum of a convex term $\lambda|t|$ and a concave component $q_\lambda(t)$. Hence, the problem (2) can be reformulated as

$$\min_{\boldsymbol{\mathcal{A}}} \mathcal{L}(\boldsymbol{\mathcal{A}}) + \mathcal{Q}_\lambda(\boldsymbol{\mathcal{A}}) + \lambda\|\boldsymbol{\mathcal{A}}\|,$$

where $\mathcal{Q}_\lambda(\boldsymbol{\mathcal{A}})$ is the concave component of the $R_\lambda(\boldsymbol{\mathcal{A}})$, and $\|\boldsymbol{\mathcal{A}}\|$ is a generic convex norm. For instance, in the entry-wise sparsity regularizer case,

$$\mathcal{Q}_\lambda(\boldsymbol{\mathcal{A}}) = \sum_{i_1=1}^{d_1} \cdots \sum_{i_N=1}^{d_N} q_\lambda(a_{i_1,\ldots,i_N})$$

and $\|\boldsymbol{\mathcal{A}}\|$ is the $\ell_1$ norm. In the mode-wise low-rankness regularizer case,

$$\mathcal{Q}_\lambda(\boldsymbol{\mathcal{A}}) = \sum_{i=1}^{\min\{I_k, \prod_{j\neq k} d_j\}} p_\lambda(\sigma_i(\boldsymbol{A}_{(k)}))$$

and $\|\boldsymbol{\mathcal{A}}\|$ is the nuclear norm of mode-$(k)$ unfolded matrix.

After this decomposition, the optimization task simplifies considerably if the term $\mathcal{Q}_\lambda(\boldsymbol{\mathcal{A}})$ is omitted, reducing to the classical linear programming problem. Motivated by this observation, the objective function can be reformulated as

$$\min_{\boldsymbol{\mathcal{A}}} \widetilde{\mathcal{L}}(\boldsymbol{\mathcal{A}}) + \|\boldsymbol{\mathcal{A}}\|,$$

where $\widetilde{\mathcal{L}}(\boldsymbol{\mathcal{A}}) = \mathcal{L}(\boldsymbol{\mathcal{A}}) + \mathcal{Q}_\lambda(\boldsymbol{\mathcal{A}})$ can be served as a surrogate function and $\|\boldsymbol{\mathcal{A}}\|$ as a new convex penalty.

Since $\widehat{\boldsymbol{\mathcal{A}}}$ is the exact global solution to the optimization problem (2). By the Karush-Kuhn-Tucker (KKT) conditions, $\widehat{\boldsymbol{\mathcal{A}}}$ satisfies the following first-order optimal condition:

$$\nabla\widetilde{\mathcal{L}}(\widehat{\boldsymbol{\mathcal{A}}}) + \lambda\widehat{\boldsymbol{\mathcal{G}}} = \boldsymbol{0}, \tag{4}$$

where $\widehat{\boldsymbol{\mathcal{G}}} \in \partial\|\widehat{\boldsymbol{\mathcal{A}}}\|$ is a subgradient of $\|\cdot\|$. Equivalently, for all $\boldsymbol{\mathcal{A}}$,

$$\langle\widehat{\boldsymbol{\mathcal{A}}} - \boldsymbol{\mathcal{A}}, \nabla\widetilde{\mathcal{L}}(\widehat{\boldsymbol{\mathcal{A}}}) + \lambda\widehat{\boldsymbol{\mathcal{G}}}\rangle \leq 0. \tag{5}$$

Based on the optimality condition in (5), we measure the suboptimality of an $\boldsymbol{\mathcal{A}}$ using

$$\omega_\lambda(\boldsymbol{\mathcal{A}}) = \min_{\boldsymbol{\mathcal{G}} \in \partial\|\widehat{\boldsymbol{\mathcal{A}}}\|_*} \left\|\nabla\widetilde{\mathcal{L}}(\boldsymbol{\mathcal{A}}) + \lambda\boldsymbol{\mathcal{G}}\right\|_*,$$

where $\|\cdot\|_*$ denote the dual norm of $\|\cdot\|$. We say $\boldsymbol{\mathcal{A}}$ is an $\epsilon$-optimal solution to (2) if $\omega_\lambda(\boldsymbol{\mathcal{A}}) \leq \epsilon$ with $\epsilon$ a small positive number. Intuitively, when $\boldsymbol{\mathcal{A}}$ is the exact global optimum, $\omega_\lambda(\boldsymbol{\mathcal{A}}) \leq 0$ by the KKT condition (4); otherwise, if $\boldsymbol{\mathcal{A}}$ is near-optimal, then $\omega_\lambda(\boldsymbol{\mathcal{A}})$ remains small but slightly positive.

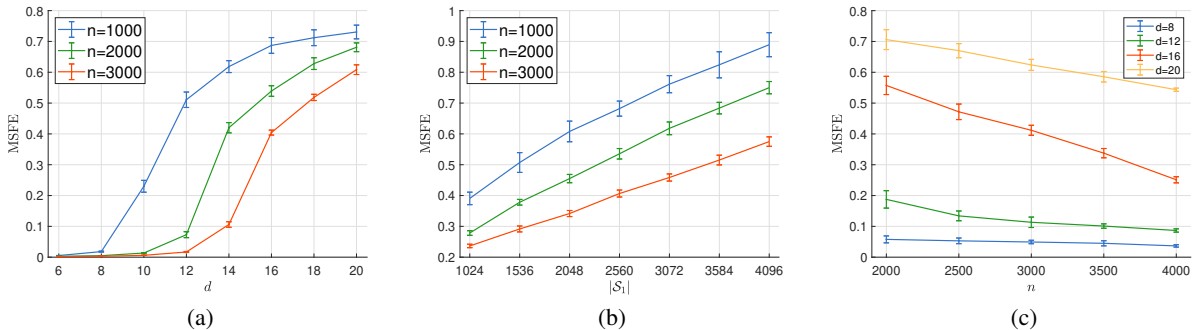

Figure 1: Entry-wise sparsity regularizer with the error bars of MSFE $\pm$ standard deviation.

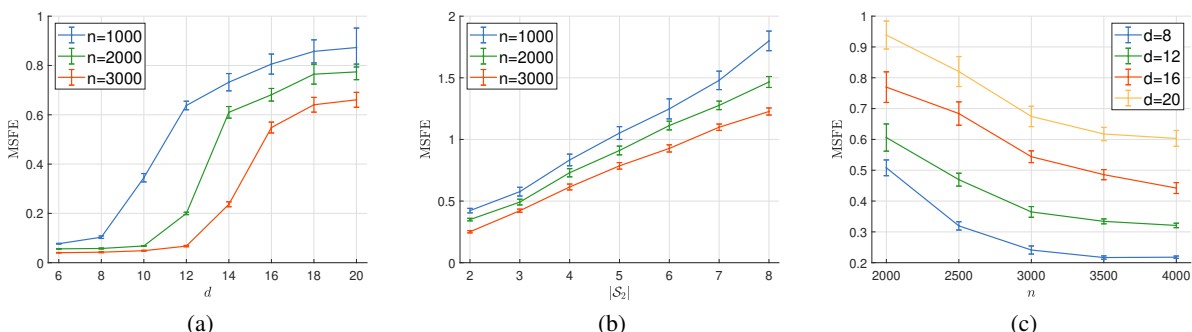

Figure 2: Mode-wise lowrankness regularizer with the error bars of MSFE $\pm$ standard deviation.

# 6. Numerical Experiments

In this section, we evaluate the performance of the proposed tensor regression model with various regularization schemes, as well as the optimization algorithm. In all experiments, the SCAD penalty is employed as the nonconvex regularizer. The estimation performance is measured by the Mean Squared Frobenius norm Error (MSFE) and the Root Mean Square Error (RMSE). Specifically, the MSFE is defined as

$$\text{MSFE} = \frac{1}{\prod_{i=1}^{M} d_i} \left\| \boldsymbol{\mathcal{A}}^{\star} - \widehat{\boldsymbol{\mathcal{A}}} \right\|_{\text{F}}^2,$$

where $\widehat{\boldsymbol{\mathcal{A}}}^2$ is the estimated results and $\boldsymbol{\mathcal{A}}^{\star}$ is the true value, and the RMSE is defined as

$$\text{RMSE} = \sqrt{\frac{1}{n} \sum_{i=1}^{n} \left\| \boldsymbol{\mathcal{Y}}^{(i)} - \left\langle \boldsymbol{\mathcal{X}}^{(i)}, \widehat{\boldsymbol{\mathcal{A}}} \right\rangle \right\|_{\text{F}}^2}.$$

The tuning parameter $\lambda$ and the hyperparameter within the SCAD penalty are selected via ten-fold cross-validation,

---

$^2$In the whole section, we use $\widehat{\boldsymbol{\mathcal{A}}}$ to denote a generic tensor estimator, which can be the estimation results obtained by various algorithms and estimators.

aiming to minimize the estimation error. All the reported results are averaged on 100 Monte Carlo realizations to ensure statistical robustness.

### 6.1. Synthetic Data

We first evaluate our proposed estimator through comprehensive synthetic experiments. We generate a set of independent covariate tensors $\{\boldsymbol{\mathcal{X}}^{(i)}\}_{i=1}^{n}$, each entry independently drawn from a standard Gaussian distribution. The response variables are obtained according to model (1), with additive independent Gaussian noise having zero mean and variance parameterized by $\eta^2$. For all synthetic experiments, we employ 3rd-order tensors $\boldsymbol{\mathcal{A}} \in \mathbb{R}^{d \times d \times d}$. For Figures 1 and 2, the noise parameter $\eta$ is set to 0.1.

Figure 1 shows the estimation performance when employing the entry-wise sparsity regularizer, varying the dimension $d$, the number of nonzero entries $|\mathcal{S}_1|$ and the sample size $n$, respectively. In Figure 1a and 1b, three lines correspond to sample size $n = \{1000, 2000, 3000\}$. The proportion of non-zero entries $s^{\star}/d^3$ is set to 0.5 for Figures 1a and 1c. Figure 1a demonstrates that a large sample size $n$ consistently lower estimation error. In contrast, Figure 1b shows that higher sparsity levels raise estimation error.

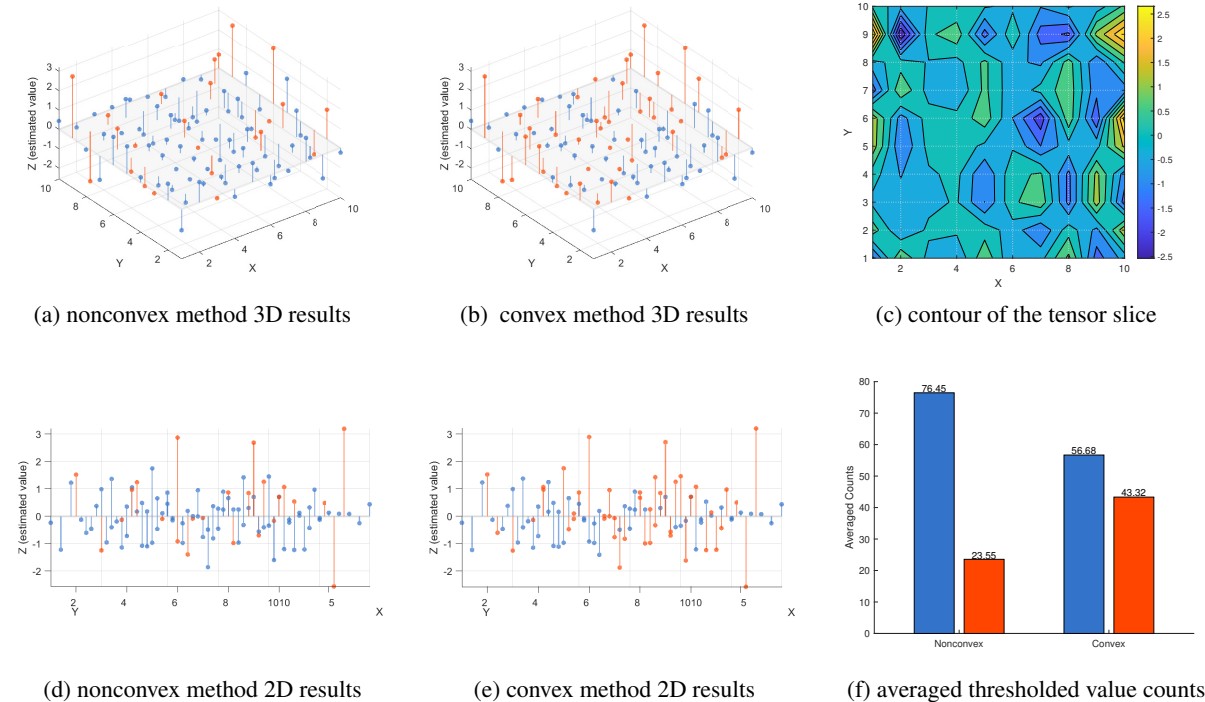

| (a) nonconvex method 3D results | (b) convex method 3D results | (c) contour of the tensor slice |
|---|---|---|
| (d) nonconvex method 2D results | (e) convex method 2D results | (f) averaged thresholded value counts |

Figure 3: Visualization of the estimated tensors with nonconvex and convex methods.

Table 1: Comparisons between proposed nonconvex regularizers and convex regularizers.

| Structures | Methods | Synthetic Data | | | | | Real-world Data | |
|---|---|---|---|---|---|---|---|---|
| | | size | $|\mathcal{S}|$ | $\eta$ | MSFE | RMSE | MSFE | MPRE |
| entry-wise sparsity | Nonconvex | $16 \times 16 \times 16$ | 2048 | 0.1 | **0.4042 ± 0.0201** | 0.0992 ± 0.0021 | **134.5864 ± 11.2950** | **7.6072 ± 0.0301** |
| | Convex | | | | 0.6938 ± 0.0297 | 0.1004 ± 0.0023 | 144.7160 ± 14.9947 | 7.7498 ± 0.0457 |
| mode-wise low-rankness | Nonconvex | $16 \times 16 \times 16$ | 5 | 1 | **0.5482 ± 0.0395** | **0.1002 ± 0.0012** | **35.5536 ± 1.4889** | **1.0330 ± 0.0022** |
| | Convex | | | | 1.7411 ± 0.0953 | 0.1096 ± 0.0020 | 41.2719 ± 3.5079 | 1.1027 ± 0.0024 |

Figure 2 illustrates the performance of the mode-wise low-rankness penalty. In Figure 2b, the $x$-axis $|\mathcal{S}_2|$ represent the rank of the mode-$(k)$ unfolded matrix. Figures 2a and 2c set the rank of at 5, while Figure 2b and 2c fix the dimension of each mode to 16. Each figure displays three distinct lines that correspond to the estimation errors for sample sizes of $n = \{1000, 2000, 3000\}$.

In Figure 3, we plot each point in the matrix using its indices as coordinates, with the corresponding value on the $z$-axis. A threshold color points to blue if their absolute error is below this threshold, and red indicates otherwise. Figure 3a and 3b compare nonconvex and convex methods on the tensor slice based on the mode-wise low-rank structure, where the size is $10 \times 10$, the sample number $n = 1000$ and the noise parameter $\eta = 0.1$. Figure 3c shows the contour of the original tensor slice, with the estimation threshold set to $10^{-2}$. Figures 3d and 3e visualize the thresholded results of the estimation in 2D, and Figure 3f illustrates average

counts of values above or below the threshold.

Table 1 compares the performance of our proposed non-convex regularizers against traditional convex regularizers (Raskutti et al., 2019). For sparsity, we set the $\eta = 0.1$, and for low-rankness, $\eta = 1$. We configure the tensor dimension such $\mathcal{A} \in \mathbb{R}^{d \times d \times s}$ or $\mathcal{A} \in \mathbb{R}^{d \times d \times d}$, with $d = 16$, $s = 20$. In all settings, our proposed regularizers achieve lower MSFE and RMSE than convex methods, aligning with our theoretical analysis.

### 6.2. Real-world Datasets

We validate our method on ImageNet 2012 dataset (Russakovsky et al., 2015) for image denoising using a 3rd-order tensor $\mathcal{A} \in \mathbb{R}^{64 \times 64 \times 3}$ with $n = 4000$ samples. In addition to MSFE, we also report Mean Prediction Relative Error (MPRE), defined as $\frac{1}{n} \sum_{i=1}^{n} \frac{\left\| \mathcal{Y}^{\star} - \hat{\mathcal{Y}}^{(i)} \right\|_{\mathrm{F}}}{\left\| \mathcal{Y}^{\star} \right\|_{\mathrm{F}}}$, where

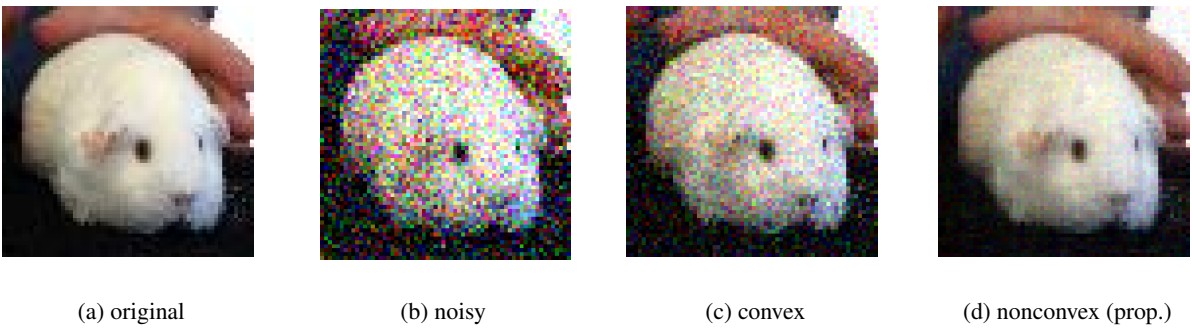

(a) original     (b) noisy     (c) convex     (d) nonconvex (prop.)

Figure 4: The original image, noisy image, and denoised images using convex and nonconvex methods.

$\mathcal{Y}^{\star} = \langle \mathcal{A}^{\star}, \mathcal{X} \rangle$ and $\widehat{\mathcal{Y}}^{(i)}$ is the $i$-th prediction. Figure 4 shows the estimated image, and Table 1 presents the comparative results.

## 7. Conclusions and Future Work

In this paper, we propose a comprehensive framework for tensor regression estimation using nonconvex regularizers. Our findings demonstrate that estimators employing nonconvex regularizers exhibit faster convergence rates compared to those with convex regularizers. Furthermore, we show that under several mild conditions, our proposed estimator possesses the oracle property. Extensive experimental results validate our theoretical claims, showcasing a close alignment between the theoretical predictions and the observed numerical performance of our estimators. Currently, we are limited to applying regularization regularizers to tensor regression models. It would be desirable to derive some theoretical guarantees for alternative methods that capture structure in the tensor regression models, such as tensor decomposition; this is the aim of our future work. To conclude, our work effectively bridges the gap between practical applications and theoretical analysis of tensor-on-tensor regression with nonconvex regularizers. To the best of our knowledge, this is the first work to obtain the oracle statistical rate of convergence for the tensor regression problem.

## Impact Statement

Tensor regression provides substantial advantages over traditional regression methods, particularly in settings involving multi-way data such as video analysis, multi-modal signals, or high-dimensional biological datasets. This work aims to advance the field of tensor regression by developing more effective and scalable models tailored to such complex data structures. While the proposed methods may have broad societal implications in various domains (e.g., health-care, environmental monitoring, and multimedia), we do not identify any specific foreseeable consequences that require explicit attention at this time.

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

# Supplementary Materials for
# "High-Dimensional Tensor Regression with Oracle Properties"

## Content of Appendix

# A. More Low-Dimensional Structures

In many applications, tensor coefficients exhibit structures beyond the aforementioned. For instance, an entire fiber or slice of a tensor might be zero, or a slice might be low-rank (Li et al., 2015b; Raskutti et al., 2019; Chen et al., 2019). Below, we introduce several penalties that promote these more intricate structures, followed by a corresponding theoretical analysis.

We now provide oracle-style guarantees for estimators employing these penalties. Our analysis parallels that of Section 4 and leverages similar assumptions.

## A.1. Fiber-wise sparsity

Consider unfolding the tensor $\boldsymbol{\mathcal{A}}$ along mode-$(k)$, the fiber-wise sparsity regularizer is defined as

$$\mathcal{R}_\lambda\left(\boldsymbol{\mathcal{A}}\right) = \sum_{l=1}^{\prod_{j\neq k} d_j} p_\lambda\left(\left\|\left[\boldsymbol{A}_{(k)}\right]_{\cdot,l}\right\|_2\right),$$

where the $[\boldsymbol{X}]_{\cdot,j}$ denotes the $j$-th column of $\boldsymbol{X}$ and $\|\cdot\|_2$ is the vector 2-norm. This penalty encourages entire fibers to be zero—analogous to the group Lasso (Yuan & Lin, 2006)

The oracle rate refers to the statistical convergence rate of the fiber-wise sparse oracle estimator, which is assumed to know the true support set $\mathcal{S}_3$ in advance, where $\mathcal{S}_3 = \left\{i \mid \left\|\left[\boldsymbol{A}_{(k)}\right]_{\cdot,i}\right\|_2 \neq 0\right\}$. Specifically, the fiber-wise sparse oracle estimator is defined as

$$\widehat{\boldsymbol{\mathcal{A}}}^O = \arg\min_{\boldsymbol{\mathcal{A}}:\boldsymbol{\mathcal{A}}_{\overline{\mathcal{S}_3}}=\mathbf{0}} \mathcal{L}\left(\boldsymbol{\mathcal{A}}\right).$$

**Theorem 7** (Fiber-wise sparsity). *Suppose that Assumptions 1~4 hold. If $\mu > \zeta$,    $\lambda \asymp \sqrt{\frac{d_k}{n}}$, and $\left[\boldsymbol{A}_{(k)}\right]_{\cdot,l}$ satisfies the condition that*

$$\min_{l\in\mathcal{S}_3}\left\|\left[\boldsymbol{A}_{(k)}\right]_{\cdot,l}\right\|_2 \geq \nu,$$

*the estimator $\widehat{\boldsymbol{\mathcal{A}}}$ to problem (2) satisfies*

$$\left\|\widehat{\boldsymbol{\mathcal{A}}} - \boldsymbol{\mathcal{A}}^\star\right\|_{\mathrm{F}} \lesssim \sqrt{\frac{|\mathcal{S}_3|\,d_k}{n}}.$$

## A.2. Slice-wise sparsity

In other scenarios, one may expect entire tensor slices to be zero (Raskutti et al., 2019). To promote such slice-wise sparsity, we introduce the slice-wise sparsity regularizer, defined as

$$\mathcal{R}_\lambda\left(\boldsymbol{\mathcal{A}}\right) = \sum_{l=1}^{\prod_{s\neq j,k} d_s} p_\lambda\left(\left\|\left[\boldsymbol{\mathcal{A}}_{(j,k)}\right]_{\cdot,\cdot,l}\right\|_{\mathrm{F}}\right).$$

The oracle rate refers to the statistical convergence rate of the slice-wise sparse oracle estimator, which is assumed to know the true support set $\mathcal{S}_4$ in advance, where $\mathcal{S}_4 = \left\{l \mid \left\|\left[\boldsymbol{\mathcal{A}}_{(j,k)}\right]_{\cdot,\cdot,l}\right\|_{\mathrm{F}} \neq 0\right\}$. Specifically, the slice-wise sparse oracle estimator is defined as

$$\widehat{\boldsymbol{\mathcal{A}}}^O = \arg\min_{\boldsymbol{\mathcal{A}}:\boldsymbol{\mathcal{A}}_{\overline{\mathcal{S}_4}}=\mathbf{0}} \mathcal{L}\left(\boldsymbol{\mathcal{A}}\right).$$

**Theorem 8** (Slice-wise sparsity). *Suppose that Assumptions 1~4 hold. If $\mu > \zeta$, $\lambda \asymp \sqrt{\frac{d_j d_k}{n}}$, and $\left[\boldsymbol{\mathcal{A}}_{(j,k)}\right]_{\cdot,\cdot,l}$ satisfies the condition that*

$$\min_{l\in\mathcal{S}_4}\left\|\left[\boldsymbol{\mathcal{A}}_{(j,k)}\right]_{\cdot,\cdot,l}\right\|_{\mathrm{F}} \geq \nu,$$

*the estimator $\widehat{\boldsymbol{\mathcal{A}}}$ to problem (2) satisfies*

$$\left\|\widehat{\boldsymbol{\mathcal{A}}} - \boldsymbol{\mathcal{A}}^\star\right\|_{\mathrm{F}} \lesssim \sqrt{\frac{|\mathcal{S}_4|\,d_j d_k}{n}}.$$

### A.3. Slice-wise low-rankness

Beyond mode-wise low-rankness, some applications benefit from low-rankness within individual slices (Lock, 2018; Raskutti et al., 2019). To capture and exploit this structure, we introduce the slice-wise low-rankness regularizer, defined as

$$\mathcal{R}_\lambda\left(\boldsymbol{\mathcal{A}}\right) = \sum_{l=1}^{\prod_{m\neq j,k} d_m} \sum_{s=1}^{\min\{d_j d_k, \prod_{l\neq j,k} d_l\}} p_\lambda\left(\sigma_s\left(\left[\boldsymbol{\mathcal{A}}_{(j,k)}\right]_{\cdot,\cdot,l}\right)\right).$$

The oracle rate refers to the statistical convergence rate of the slice-wise low-rank oracle estimator, which is assumed to know the true rank $\mathcal{S}_5 = \left\{s \mid \sigma_s\left(\Pi_{\mathcal{F}}\left(\left[\boldsymbol{\mathcal{A}}^\star_{(j,k)}\right]_{\cdot,\cdot,l}\right)\right) \neq 0\right\}$ in advance. Specifically, the slice-wise low-rank oracle estimator is defined as

$$\widehat{\boldsymbol{\mathcal{A}}}^O = \arg\min_{\boldsymbol{\mathcal{A}}:\boldsymbol{\mathcal{A}}_{\overline{\mathcal{S}_5}}=\mathbf{0}} \mathcal{L}\left(\boldsymbol{\mathcal{A}}\right).$$

**Theorem 9** (Slice-wise low-rankness)**.** *Suppose that Assumptions 1~4 hold. If $\mu > \zeta$, $\lambda \gtrsim \frac{1}{n}\sqrt{|\mathcal{S}_5|}\left\|\left[\left[\mathfrak{X}^\star\left(\boldsymbol{\mathcal{E}}^{(1:n)}\right)\right]_{(j,k)}\right]_{\cdot,\cdot,l}\right\|_{\mathrm{sp}}$, and $\sigma_s\left(\left[\boldsymbol{\mathcal{A}}^\star_{(j,k)}\right]_{\cdot,\cdot,l}\right)$ satisfies the condition*

$$\left|\sigma_s\left(\left[\boldsymbol{\mathcal{A}}^\star_{(j,k)}\right]_{\cdot,\cdot,l}\right)\right| \geq \nu + \frac{2\sqrt{|\mathcal{S}_5|}}{n\mu}\left\|\left[\left[\mathfrak{X}^\star\left(\boldsymbol{\mathcal{E}}^{(1:n)}\right)\right]_{(j,k)}\right]_{\cdot,\cdot,l}\right\|_{\mathrm{sp}},$$

*the estimator $\widehat{\boldsymbol{\mathcal{A}}}$ to problem* (2) *satisfies*

$$\left\|\widehat{\boldsymbol{\mathcal{A}}} - \boldsymbol{\mathcal{A}}^\star\right\|_{\mathrm{F}} \lesssim \frac{\tau_{(j,k)}\sqrt{|\mathcal{S}_5|}}{n},$$

*where $\tau_{(j,k)} = \max_l\left\|\Pi_{\mathcal{F}}\left(\left[\mathfrak{X}^\star\left(\boldsymbol{\mathcal{E}}\right)\right]_{(j,k)}\right)_{\cdot,\cdot,l}\right\|_{\mathrm{sp}}.$*

## B. Proofs of the Theoretical Results

### B.1. Proof of Theorem 5

We begin by demonstrating that the entry-wise sparsity regularizer can be reformulated as the sum of the $\ell_1$ penalty and a concave part. Specifically, we have

$$\mathcal{R}_\lambda\left(\boldsymbol{\mathcal{A}}\right) = \sum_{i_1=1}^{d_1}\cdots\sum_{i_N=1}^{d_N} p_\lambda\left(a_{i_1,\dots,i_N}\right) = \lambda\left\|a_{i_1,\dots,i_N}\right\|_1 + \mathcal{Q}_\lambda\left(\boldsymbol{\mathcal{A}}\right),$$

where $\left\|a_{i_1,\dots,i_N}\right\|_1 := \sum_{i_1=1}^{d_1}\cdots\sum_{i_N=1}^{d_N}\left|a_{i_1,\dots,i_N}\right|$ is the $\ell_1$ norm and $\mathcal{Q}_\lambda\left(\boldsymbol{\mathcal{A}}\right) = \sum_{i_1=1}^{d_1}\cdots\sum_{i_N=1}^{d_N} q_\lambda\left(a_{i_1,\dots,i_N}\right).$

**Lemma 1.** *Under Assumptions 2 and 3, the loss function $\widetilde{\mathcal{L}}\left(\boldsymbol{\mathcal{A}}\right)$ satisfies the restricted strong convexity*

$$\widetilde{\mathcal{L}}\left(\boldsymbol{\mathcal{A}}'\right) \geq \widetilde{\mathcal{L}}\left(\boldsymbol{\mathcal{A}}\right) + \left\langle\nabla\widetilde{\mathcal{L}}\left(\boldsymbol{\mathcal{A}}\right), \boldsymbol{\mathcal{A}}' - \boldsymbol{\mathcal{A}}\right\rangle + \frac{\mu - \zeta}{2}\left\|\boldsymbol{\mathcal{A}}' - \boldsymbol{\mathcal{A}}\right\|_{\mathrm{F}}^2,$$

*and the restricted smoothness*

$$\widetilde{\mathcal{L}}\left(\boldsymbol{\mathcal{A}}'\right) \leq \widetilde{\mathcal{L}}\left(\boldsymbol{\mathcal{A}}\right) + \left\langle\nabla\widetilde{\mathcal{L}}\left(\boldsymbol{\mathcal{A}}\right), \boldsymbol{\mathcal{A}}' - \boldsymbol{\mathcal{A}}\right\rangle + \frac{L}{2}\left\|\boldsymbol{\mathcal{A}}' - \boldsymbol{\mathcal{A}}\right\|_{\mathrm{F}}^2.$$

*Proof.* Recall that $\mathcal{Q}_\lambda\left(\boldsymbol{\mathcal{A}}\right)$ represents the concave component of the non-convex penalty $\mathcal{R}_\lambda\left(a_{i_1,\dots,i_N}\right)$, implying that $-\mathcal{Q}_\lambda\left(\boldsymbol{\mathcal{A}}\right)$ is convex. Specifically, $\mathcal{Q}_\lambda\left(\boldsymbol{\mathcal{A}}\right)$ can be expressed as a sum over its entries $\mathcal{Q}_\lambda\left(\boldsymbol{\mathcal{A}}\right) = \sum_{i_1=1}^{d_1}\cdots\sum_{i_N=1}^{d_N} q_\lambda\left(a_{i_1,\dots,i_N}\right)$, where $q_\lambda\left(a_{i_1,\dots,i_N}\right)$ satisfies the third regularity condition specified in Assumption 1. From this assumption, we have

$$-\zeta\left(a'_{i_1,\dots,i_N} - a_{i_1,\dots,i_N}\right)^2 \leq \left(q'_\lambda\left(a'_{i_1,\dots,i_N}\right) - q'_\lambda\left(a_{i_1,\dots,i_N}\right)\right)\left(a'_{i_1,\dots,i_N} - a_{i_1,\dots,i_N}\right) \leq 0.$$

By aggregating over all entries, we deduce that the convex function $-\mathcal{Q}_\lambda\left(\boldsymbol{\mathcal{A}}\right)$ satisfies the following inequality

$$\left\langle\left(\nabla\left(-\mathcal{Q}_\lambda\left(\boldsymbol{\mathcal{A}}'\right)\right)-\nabla\left(-\mathcal{Q}_\lambda\left(\boldsymbol{\mathcal{A}}\right)\right)\right)^\top,\boldsymbol{\mathcal{A}}'-\boldsymbol{\mathcal{A}}\right\rangle\le\zeta\left\|\boldsymbol{\mathcal{A}}'-\boldsymbol{\mathcal{A}}\right\|_{\mathrm{F}}^2,\tag{6}$$

$$\left\langle\left(\nabla\left(-\mathcal{Q}_\lambda\left(\boldsymbol{\mathcal{A}}'\right)\right)-\nabla\left(-\mathcal{Q}_\lambda\left(\boldsymbol{\mathcal{A}}\right)\right)\right)^\top,\boldsymbol{\mathcal{A}}'-\boldsymbol{\mathcal{A}}\right\rangle\ge0.\tag{7}$$

Inequalities (6) and (7) correspond to the definitions of RSC and RSM for the function $-\mathcal{Q}_\lambda\left(\boldsymbol{\mathcal{A}}\right)$, respectively. Specifically, they imply that $-\mathcal{Q}_\lambda\left(\boldsymbol{\mathcal{A}}\right)$ is both $\zeta$-smooth and 0-strongly convex. Consequently, we have the following

$$-\mathcal{Q}_\lambda\left(\boldsymbol{\mathcal{A}}'\right)\le-\mathcal{Q}_\lambda\left(\boldsymbol{\mathcal{A}}\right)-\left\langle\nabla\mathcal{Q}_\lambda\left(\boldsymbol{\mathcal{A}}\right),\boldsymbol{\mathcal{A}}'-\boldsymbol{\mathcal{A}}\right\rangle+\frac{\zeta}{2}\left\|\boldsymbol{\mathcal{A}}'-\boldsymbol{\mathcal{A}}\right\|_{\mathrm{F}}^2,$$

$$-\mathcal{Q}_\lambda\left(\boldsymbol{\mathcal{A}}'\right)\ge-\mathcal{Q}_\lambda\left(\boldsymbol{\mathcal{A}}\right)-\left\langle\nabla\mathcal{Q}_\lambda\left(\boldsymbol{\mathcal{A}}\right),\boldsymbol{\mathcal{A}}'-\boldsymbol{\mathcal{A}}\right\rangle.$$

For the loss function $\mathcal{L}\left(\boldsymbol{\mathcal{A}}\right)$, applying Taylor's theorem and the mean value theorem yields

$$\mathcal{L}\left(\boldsymbol{\mathcal{A}}'\right)=\mathcal{L}\left(\boldsymbol{\mathcal{A}}\right)+\left\langle\nabla\mathcal{L}\left(\boldsymbol{\mathcal{A}}\right),\boldsymbol{\mathcal{A}}'-\boldsymbol{\mathcal{A}}\right\rangle+\frac{1}{2}\left\langle\nabla^2\mathcal{L}\left(\beta\boldsymbol{\mathcal{A}}'+(1-\beta)\boldsymbol{\mathcal{A}}\right),\left(\boldsymbol{\mathcal{A}}'-\boldsymbol{\mathcal{A}}\right)\otimes\left(\boldsymbol{\mathcal{A}}'-\boldsymbol{\mathcal{A}}\right)\right\rangle,$$

for some $\beta\in[0,1]$. Here, $\otimes$ denotes the Kronecker product. Given two tensors $\boldsymbol{\mathcal{A}},\boldsymbol{\mathcal{A}}'\in\mathbb{R}^{d_1\times\cdots\times d_N}$, their Kronecker product results in a tensor $\boldsymbol{\mathcal{A}}''$ of dimension $(d_1d_1)\times\ldots(d_Nd_N)$. Each entry $a''_{i_1j_1,i_2j_2,\ldots,i_Nj_N}$ is defined as $a_{i_1,i_2,\ldots,i_N}\times a'_{j_1,j_2,\ldots,j_N}$.

Under Assumptions 2 and 3, we have

$$\mathcal{L}\left(\boldsymbol{\mathcal{B}}\right)-\mathcal{L}\left(\boldsymbol{\mathcal{A}}\right)\ge\left\langle\nabla\mathcal{L}\left(\boldsymbol{\mathcal{A}}\right),\boldsymbol{\mathcal{B}}-\boldsymbol{\mathcal{A}}\right\rangle+\frac{\mu}{2}\|\boldsymbol{\mathcal{B}}-\boldsymbol{\mathcal{A}}\|_{\mathrm{F}}^2,$$

$$\mathcal{L}\left(\boldsymbol{\mathcal{B}}\right)-\mathcal{L}\left(\boldsymbol{\mathcal{A}}\right)\le\left\langle\nabla\mathcal{L}\left(\boldsymbol{\mathcal{A}}\right),\boldsymbol{\mathcal{B}}-\boldsymbol{\mathcal{A}}\right\rangle+\frac{L}{2}\|\boldsymbol{\mathcal{B}}-\boldsymbol{\mathcal{A}}\|_{\mathrm{F}}^2.$$

Recall that $\widetilde{\mathcal{L}}\left(\boldsymbol{\mathcal{A}}\right)=\mathcal{L}\left(\boldsymbol{\mathcal{A}}\right)+\mathcal{Q}_\lambda\left(\boldsymbol{\mathcal{A}}\right)$. Thus, we obtain

$$\widetilde{\mathcal{L}}\left(\boldsymbol{\mathcal{A}}'\right)\ge\widetilde{\mathcal{L}}\left(\boldsymbol{\mathcal{A}}\right)+\left\langle\nabla\widetilde{\mathcal{L}}\left(\boldsymbol{\mathcal{A}}\right),\boldsymbol{\mathcal{A}}'-\boldsymbol{\mathcal{A}}\right\rangle+\frac{\mu-\zeta}{2}\left\|\boldsymbol{\mathcal{A}}'-\boldsymbol{\mathcal{A}}\right\|_{\mathrm{F}}^2,$$

$$\widetilde{\mathcal{L}}\left(\boldsymbol{\mathcal{A}}'\right)\le\widetilde{\mathcal{L}}\left(\boldsymbol{\mathcal{A}}\right)+\left\langle\nabla\widetilde{\mathcal{L}}\left(\boldsymbol{\mathcal{A}}\right),\boldsymbol{\mathcal{A}}'-\boldsymbol{\mathcal{A}}\right\rangle+\frac{L}{2}\left\|\boldsymbol{\mathcal{A}}'-\boldsymbol{\mathcal{A}}\right\|_{\mathrm{F}}^2,$$

$\square$

**Lemma 2.** *Suppose there exists an integer $\widetilde{s}_1>C\left|\mathcal{S}_1\right|$, where $C$ is a constant, and that $\boldsymbol{\mathcal{A}}$ satisfies $\left\|\boldsymbol{\mathcal{A}}_{\overline{\mathcal{S}_1}}\right\|_0\le\widetilde{s}_1$, $\omega\left(\boldsymbol{\mathcal{A}}\right)\le\frac{\lambda}{2}$, and $\left\|\nabla\mathcal{L}\left(\boldsymbol{\mathcal{A}}^\star\right)\right\|_{\max}\le\lambda/8$, where $\|\cdot\|_{\max}$ denotes the maximal element of the tensor. Under Assumptions 2 and 3, $\boldsymbol{\mathcal{A}}$ satisfies*

$$\|\boldsymbol{\mathcal{A}}-\boldsymbol{\mathcal{A}}^\star\|_{\mathrm{F}}\le\frac{21/8}{\mu-\zeta}\lambda\sqrt{\left|\mathcal{S}_1\right|}.$$

*Proof.* Given that $\left\|\boldsymbol{\mathcal{A}}_{\overline{\mathcal{S}_1}}\right\|_0\le\widetilde{s}_1$ and $\left\|\boldsymbol{\mathcal{A}}^\star_{\overline{\mathcal{S}_1}}\right\|_0=0$, it follows that $\left\|(\boldsymbol{\mathcal{A}}-\boldsymbol{\mathcal{A}}^\star)_{\overline{\mathcal{S}_1}}\right\|_0\le\widetilde{s}_1$. Based on Lemma 1, we can derive the following inequalities

$$\widetilde{\mathcal{L}}\left(\boldsymbol{\mathcal{A}}^\star\right)\ge\widetilde{\mathcal{L}}\left(\boldsymbol{\mathcal{A}}\right)+\left\langle\nabla\widetilde{\mathcal{L}}\left(\boldsymbol{\mathcal{A}}\right),\boldsymbol{\mathcal{A}}^\star-\boldsymbol{\mathcal{A}}\right\rangle+\frac{\mu-\zeta}{2}\left\|\boldsymbol{\mathcal{A}}^\star-\boldsymbol{\mathcal{A}}\right\|_{\mathrm{F}}^2,\tag{8}$$

$$\widetilde{\mathcal{L}}\left(\boldsymbol{\mathcal{A}}\right)\ge\widetilde{\mathcal{L}}\left(\boldsymbol{\mathcal{A}}^\star\right)+\left\langle\nabla\widetilde{\mathcal{L}}\left(\boldsymbol{\mathcal{A}}^\star\right),\boldsymbol{\mathcal{A}}-\boldsymbol{\mathcal{A}}^\star\right\rangle+\frac{\mu-\zeta}{2}\left\|\boldsymbol{\mathcal{A}}-\boldsymbol{\mathcal{A}}^\star\right\|_{\mathrm{F}}^2.\tag{9}$$

Adding (8) and (9), we obtain

$$\left\langle\nabla\widetilde{\mathcal{L}}\left(\boldsymbol{\mathcal{A}}\right),\boldsymbol{\mathcal{A}}^\star-\boldsymbol{\mathcal{A}}\right\rangle\ge\left\langle\nabla\widetilde{\mathcal{L}}\left(\boldsymbol{\mathcal{A}}^\star\right),\boldsymbol{\mathcal{A}}-\boldsymbol{\mathcal{A}}^\star\right\rangle+(\mu-\zeta)\left\|\boldsymbol{\mathcal{A}}^\star-\boldsymbol{\mathcal{A}}\right\|_{\mathrm{F}}^2.\tag{10}$$

Let $\mathcal{G} \in \partial \|\mathcal{A}\|_1$ denote the sub-gradient and $\mathcal{S}$ be a set. According to the Karush-Kuhn-Tucker (KKT) condition, we have

$$\nabla \widetilde{\mathcal{L}}(\mathcal{A}) + \lambda \mathcal{G} = \mathbf{0}.$$

Determining the optimal solution is challenging, therefore, we introduce a measure of sub-optimality

$$\omega(\mathcal{A}) = \min_{\mathcal{G}' \in \partial \|\mathcal{A}\|_1} \max_{\mathcal{A}' \in \mathcal{S}} \left\{ \frac{1}{\|\mathcal{A} - \mathcal{A}'\|_1} \left\langle \mathcal{A} - \mathcal{A}', \nabla \widetilde{\mathcal{L}}(\mathcal{A}) + \lambda \mathcal{G}' \right\rangle \right\}.$$

We define our algorithm's stopping criterion as $\omega(\mathcal{A}) \leq \varepsilon$. Consequently, the sub-optimality can be expressed as

$$\omega(\mathcal{A}) = \max_{\mathcal{A}' \in \mathcal{S}} \left\{ \frac{1}{\|\mathcal{A} - \mathcal{A}'\|_1} \left\langle \mathcal{A} - \mathcal{A}', \nabla \widetilde{\mathcal{L}}(\mathcal{A}) + \lambda \mathcal{G} \right\rangle \right\}.$$

Adding $\lambda \left\langle \mathcal{A} - \mathcal{A}^\star, \mathcal{G}' \right\rangle$ to the both sides of (10), we obtain

$$\left\langle \mathcal{A} - \mathcal{A}^\star, \nabla \widetilde{\mathcal{L}}(\mathcal{A}) + \lambda \mathcal{G} \right\rangle \geq \left\langle \mathcal{A} - \mathcal{A}^\star, \nabla \widetilde{\mathcal{L}}(\mathcal{A}) \right\rangle + (\mu - \zeta) \|\mathcal{A}^\star - \mathcal{A}\|_{\mathrm{F}}^2 + \lambda \left\langle \mathcal{A} - \mathcal{A}^\star, \mathcal{G} \right\rangle.$$

Since $\mathcal{A}^\star \in \mathcal{S}$, we have

$$\frac{1}{\|\mathcal{A} - \mathcal{A}^\star\|_1} \left\langle \mathcal{A} - \mathcal{A}^\star, \nabla \widetilde{\mathcal{L}}(\mathcal{A}) + \lambda \mathcal{G} \right\rangle \leq \max_{\mathcal{A}' \in \mathcal{S}} \left\{ \frac{1}{\|\mathcal{A} - \mathcal{A}'\|_1} \left\langle \mathcal{A} - \mathcal{A}', \nabla \widetilde{\mathcal{L}}(\mathcal{A}) + \lambda \mathcal{G} \right\rangle \right\} = v(\mathcal{A}).$$

Recall that we assume $v(\mathcal{A}) \leq \frac{\lambda}{2}$, we obtain

$$\left\langle \mathcal{A} - \mathcal{A}^\star, \nabla \widetilde{\mathcal{L}}(\mathcal{A}) + \lambda \mathcal{G} \right\rangle \leq v(\mathcal{A}) \leq \frac{\lambda}{2} \|\mathcal{A} - \mathcal{A}^\star\|_1. \tag{11}$$

Combining (10) and (11), we obtain

$$\frac{\lambda}{2} \|\mathcal{A} - \mathcal{A}^\star\|_1 \geq \underbrace{\left\langle \mathcal{A} - \mathcal{A}^\star, \nabla \widetilde{\mathcal{L}}(\mathcal{A}^\star) \right\rangle + (\mu - \zeta) \|\mathcal{A}^\star - \mathcal{A}\|_{\mathrm{F}}^2}_{\mathrm{I}} + \underbrace{\lambda \left\langle \mathcal{A} - \mathcal{A}^\star, \mathcal{G} \right\rangle}_{\mathrm{II}}.$$

For term I, separating the support of $\mathcal{A} - \mathcal{A}^\star$ into $\mathcal{S}_1$ and $\overline{\mathcal{S}_1}$, we have

$$\begin{aligned}
\left\langle \mathcal{A} - \mathcal{A}^\star, \nabla \widetilde{\mathcal{L}}(\mathcal{A}^\star) \right\rangle &= \left\langle \mathcal{A} - \mathcal{A}^\star, \nabla \widetilde{\mathcal{L}}(\mathcal{A}^\star) \right\rangle + \left\langle \mathcal{A} - \mathcal{A}^\star, \nabla \mathcal{Q}_\lambda(\mathcal{A}^\star) \right\rangle \\
&\geq - \|\mathcal{A} - \mathcal{A}^\star\|_1 \|\nabla \mathcal{L}(\mathcal{A}^\star)\|_{\max} + \left\langle \mathcal{A} - \mathcal{A}^\star, \nabla \mathcal{Q}_\lambda(\mathcal{A}^\star) \right\rangle \\
&= - \left\|(\mathcal{A} - \mathcal{A}^\star)_{\overline{\mathcal{S}_1}}\right\|_1 \|\nabla \mathcal{L}(\mathcal{A}^\star)\|_{\max} - \left\|(\mathcal{A} - \mathcal{A}^\star)_{\mathcal{S}_1}\right\|_1 \|\nabla \mathcal{L}(\mathcal{A}^\star)\|_{\max} \\
&\quad + \left\langle (\mathcal{A} - \mathcal{A}^\star)_{\mathcal{S}_1}, (\nabla \mathcal{Q}_\lambda(\mathcal{A}^\star))_{\mathcal{S}_1} \right\rangle + \left\langle (\mathcal{A} - \mathcal{A}^\star)_{\overline{\mathcal{S}_1}}, (\nabla \mathcal{Q}_\lambda(\mathcal{A}^\star))_{\overline{\mathcal{S}_1}} \right\rangle \\
&\geq - \left\|(\mathcal{A} - \mathcal{A}^\star)_{\overline{\mathcal{S}_1}}\right\|_1 \|\nabla \mathcal{L}(\mathcal{A}^\star)\|_{\max} - \left\|(\mathcal{A} - \mathcal{A}^\star)_{\mathcal{S}_1}\right\|_1 \|\nabla \mathcal{L}(\mathcal{A}^\star)\|_{\max} \\
&\quad - \left\|(\mathcal{A} - \mathcal{A}^\star)_{\mathcal{S}_1}\right\|_1 \|\nabla \mathcal{Q}_\lambda(\mathcal{A}^\star)\|_{\max}.
\end{aligned}$$

For term II, separating the support of $\mathcal{A} - \mathcal{A}^\star$ into $\mathcal{S}_1$ and $\overline{\mathcal{S}_1}$, we have

$$\begin{aligned}
\lambda \left\langle \mathcal{A} - \mathcal{A}^\star, \mathcal{G} \right\rangle &= \lambda \left\langle (\mathcal{A} - \mathcal{A}^\star)_{\mathcal{S}_1}, \mathcal{G}_{\mathcal{S}_1} \right\rangle + \lambda \left\langle (\mathcal{A} - \mathcal{A}^\star)_{\overline{\mathcal{S}_1}}, \mathcal{G}_{\overline{\mathcal{S}_1}} \right\rangle \\
&\geq - \lambda \left\|(\mathcal{A} - \mathcal{A}^\star)_{\mathcal{S}_1}\right\|_1 \|\mathcal{G}_{\mathcal{S}_1}\|_{\max} + \lambda \left\langle \mathcal{A}_{\overline{\mathcal{S}_1}}, \mathcal{G}_{\overline{\mathcal{S}_1}} \right\rangle \\
&\geq - \lambda \left\|(\mathcal{A} - \mathcal{A}^\star)_{\mathcal{S}_1}\right\|_1 + \lambda \sum_{(i_1, \ldots, i_N \in \overline{\mathcal{S}_1})} |a_{i_1, \ldots, i_N}| \\
&= - \lambda \left\|(\mathcal{A} - \mathcal{A}^\star)_{\mathcal{S}_1}\right\|_1 + \lambda \left\|(\mathcal{A} - \mathcal{A}^\star)_{\overline{\mathcal{S}_1}}\right\|_1.
\end{aligned}$$

Thus, we obtain

$$\frac{\lambda}{2}\left\|\mathcal{A}-\mathcal{A}^{\star}\right\|_{1} \geq -\left\|(\mathcal{A}-\mathcal{A}^{\star})_{\overline{\mathcal{S}_{1}}}\right\|_{1}\left\|\nabla\mathcal{L}\left(\mathcal{A}^{\star}\right)\right\|_{\max} - \left\|(\mathcal{A}-\mathcal{A}^{\star})_{\mathcal{S}_{1}}\right\|_{1}\left\|\nabla\mathcal{L}\left(\mathcal{A}^{\star}\right)\right\|_{\max} \tag{12}$$
$$- \left\|(\mathcal{A}-\mathcal{A}^{\star})_{\mathcal{S}_{1}}\right\|_{1}\left\|\nabla\mathcal{Q}_{\lambda}\left(\mathcal{A}^{\star}\right)\right\|_{\max} + (\mu - \zeta)\left\|\mathcal{A}^{\star}-\mathcal{A}\right\|_{\mathrm{F}}^{2}$$
$$- \lambda\left\|(\mathcal{A}-\mathcal{A}^{\star})_{\mathcal{S}_{1}}\right\|_{1} + \lambda\left\|(\mathcal{A}-\mathcal{A}^{\star})_{\overline{\mathcal{S}_{1}}}\right\|_{1}.$$

We separate the left-hand side of (12) as

$$\frac{\lambda}{2}\left\|\mathcal{A}-\mathcal{A}^{\star}\right\|_{1} = \frac{\lambda}{2}\left\|(\mathcal{A}-\mathcal{A}^{\star})_{\mathcal{S}_{1}}\right\|_{1} + \frac{\lambda}{2}\left\|(\mathcal{A}-\mathcal{A}^{\star})_{\overline{\mathcal{S}_{1}}}\right\|_{1}.$$

Rearranging the terms, we obtain

$$(\mu - \zeta)\left\|\mathcal{A}^{\star}-\mathcal{A}\right\|_{\mathrm{F}}^{2} + \left(\frac{\lambda}{2} - \left\|\nabla\mathcal{L}\left(\mathcal{A}^{\star}\right)\right\|_{\max}\right)\left\|(\mathcal{A}-\mathcal{A}^{\star})_{\overline{\mathcal{S}_{1}}}\right\|_{1}$$
$$\leq \left(\frac{3\lambda}{2} + \left\|\nabla\mathcal{L}\left(\mathcal{A}^{\star}\right)\right\|_{\max} + \left\|\nabla\mathcal{Q}_{\lambda}\left(\mathcal{A}^{\star}\right)\right\|_{\max}\right)\left\|(\mathcal{A}-\mathcal{A}^{\star})_{\mathcal{S}_{1}}\right\|_{1}.$$

Recall that $\left\|\nabla\mathcal{L}\left(\mathcal{A}^{\star}\right)\right\|_{\max} \leq \frac{\lambda}{8}$, we have

$$(\mu - \zeta)\left\|\mathcal{A}^{\star}-\mathcal{A}\right\|_{\mathrm{F}}^{2} \leq \left(\frac{3\lambda}{2} + \left\|\nabla\mathcal{L}\left(\mathcal{A}^{\star}\right)\right\|_{\max} + \left\|\nabla\mathcal{Q}_{\lambda}\left(\mathcal{A}^{\star}\right)\right\|_{\max}\right)\left\|(\mathcal{A}-\mathcal{A}^{\star})_{\mathcal{S}_{1}}\right\|_{1}$$
$$\leq \left(\frac{3\lambda}{2} + \frac{\lambda}{8} + \lambda\right)\left\|(\mathcal{A}-\mathcal{A}^{\star})_{\mathcal{S}_{1}}\right\|_{1}$$
$$\leq \frac{21\lambda}{8}\sqrt{|\mathcal{S}_{1}|}\left\|(\mathcal{A}-\mathcal{A}^{\star})_{\mathcal{S}_{1}}\right\|_{\mathrm{F}}$$
$$\leq \frac{21\lambda}{8}\sqrt{|\mathcal{S}_{1}|}\left\|\mathcal{A}-\mathcal{A}^{\star}\right\|_{\mathrm{F}}.$$

Given that $\mu - \zeta > 0$, we have

$$\left\|\mathcal{A}-\mathcal{A}^{\star}\right\|_{\mathrm{F}} \leq \frac{21/8}{\mu - \zeta}\lambda\sqrt{|\mathcal{S}_{1}|}.$$

$\square$

**Lemma 3.** *Consider the regularization parameter $\lambda$ and assume that the derivative of the non-convex penalty satisfies $p'_{\lambda}\left(a_{i_{1},\ldots,i_{N}}\right) = 0$ whenever $\left|a_{i_{1},\ldots,i_{N}}\right| \geq \nu$ for some $\nu > 0$. Let $\mathcal{S}_{1}^{\mathrm{I}} \cup \mathcal{S}_{1}^{\mathrm{II}} = \mathcal{S}_{1}$. For indices $(i_{1},\ldots,i_{N}) \in \mathcal{S}_{1}^{\mathrm{I}} \subseteq \mathcal{S}_{1}$, we assume $\left|a_{i_{1},\ldots,i_{N}}^{\star}\right| \geq \nu$, and for indices $(i_{1},\ldots,i_{N}) \in \mathcal{S}_{1}^{\mathrm{II}} \subseteq \mathcal{S}_{1}$, we assume $\left|a_{i_{1},\ldots,i_{N}}^{\star}\right| \leq \nu$. Under Assumptions 2$\sim$4, we derive the following bound*

$$\left\|\widehat{\mathcal{A}}-\mathcal{A}^{\star}\right\|_{\mathrm{F}} \leq \frac{1}{\mu - \zeta}\left\|(\nabla\mathcal{L}\left(\mathcal{A}^{\star}\right))_{\mathcal{S}_{1}^{\mathrm{I}}}\right\|_{\mathrm{F}} + \frac{3}{\mu - \zeta}\lambda\sqrt{|\mathcal{S}_{1}^{\mathrm{II}}|}.$$

*Proof.* Define the sub-gradients $\mathcal{G}^{\star} \in \partial\left\|\mathcal{A}^{\star}\right\|_{1}$ and $\widehat{\mathcal{G}} \in \partial\left\|\widehat{\mathcal{A}}\right\|_{1}$.

Note that $\widehat{\mathcal{A}}$ satisfies the optimality condition that $\omega\left(\widehat{\mathcal{A}}\right) \leq 0$, we have

$$\max_{\mathcal{A}'\in\mathcal{S}}\left\{\left\langle\widehat{\mathcal{A}}-\mathcal{A}', \nabla\widetilde{\mathcal{L}}\left(\widehat{\mathcal{A}}\right)+\lambda\widehat{\mathcal{G}}\right\rangle\right\} \leq 0. \tag{13}$$

Given that $\left\|\widehat{\mathcal{A}}_{\overline{\mathcal{S}_1}}\right\|_0 \leq \widetilde{s}_1$, since $\left\|\left(\widehat{\mathcal{A}} - \mathcal{A}^\star\right)_{\overline{\mathcal{S}_1}}\right\|_0 \leq \widetilde{s}_1$, according to Lemma 1, we obtain

$$\widetilde{\mathcal{L}}\left(\widehat{\mathcal{A}}\right) \geq \widetilde{\mathcal{L}}\left(\mathcal{A}^\star\right) + \left\langle \nabla\widetilde{\mathcal{L}}\left(\mathcal{A}^\star\right), \widehat{\mathcal{A}} - \mathcal{A}^\star \right\rangle + \frac{\mu - \zeta}{2}\|\widehat{\mathcal{A}} - \mathcal{A}^\star\|_{\mathrm{F}}^2, \tag{14}$$

$$\widetilde{\mathcal{L}}\left(\mathcal{A}^\star\right) \geq \widetilde{\mathcal{L}}\left(\widehat{\mathcal{A}}\right) + \left\langle \nabla\widetilde{\mathcal{L}}\left(\widehat{\mathcal{A}}\right), \mathcal{A}^\star - \widehat{\mathcal{A}} \right\rangle + \frac{\mu - \zeta}{2}\|\mathcal{A}^\star - \widehat{\mathcal{A}}\|_{\mathrm{F}}^2.$$

By the convexity of $\ell_1$ norm, we have

$$\lambda\left\|\widehat{\mathcal{A}}\right\|_1 \leq \lambda\|\mathcal{A}^\star\|_1 + \lambda\left\langle \widehat{\mathcal{A}} - \mathcal{A}^\star, \mathcal{G}^\star \right\rangle,$$

$$\lambda\|\mathcal{A}^\star\|_1 \leq \lambda\left\|\widehat{\mathcal{A}}\right\|_1 + \lambda\left\langle \mathcal{A}^\star - \widehat{\mathcal{A}}, \widehat{\mathcal{G}} \right\rangle. \tag{15}$$

Adding (14) $\sim$ (15), we obtain

$$0 \geq \underbrace{\left\langle \nabla\mathcal{L}\left(\mathcal{A}^\star\right) + \nabla\mathcal{Q}_\lambda\left(a^\star_{i_1\ldots i_N}\right) + \lambda\mathcal{G}^\star, \widehat{\mathcal{A}} - \mathcal{A}^\star \right\rangle}_{(i)} + \underbrace{\left\langle \nabla\widetilde{\mathcal{L}}\left(\widehat{\mathcal{A}}\right) + \lambda\mathcal{G}^\star, \mathcal{A}^\star - \widehat{\mathcal{A}} \right\rangle}_{(ii)} + (\mu - \zeta)\left\|\widehat{\mathcal{A}} - \mathcal{A}^\star\right\|_{\mathrm{F}}^2. \tag{16}$$

From the optimality condition (13), we have

$$\left\langle \nabla\widetilde{\mathcal{L}}\left(\widehat{\mathcal{A}}\right) + \lambda\widehat{\mathcal{G}}, \mathcal{A}^\star - \widehat{\mathcal{A}} \right\rangle \leq \max_{\mathcal{A}' \in \mathcal{S}}\left\{\left\langle \widehat{\mathcal{A}} - \mathcal{A}', \nabla\widetilde{\mathcal{L}}\left(\widehat{\mathcal{A}}\right) + \lambda\widehat{\mathcal{G}} \right\rangle\right\} \leq 0,$$

which implies the term (ii) in (16) is non-negative. Consequently, we can arrange (16) to obtain

$$(\mu - \zeta)\left\|\widehat{\mathcal{A}} - \mathcal{A}^\star\right\|_{\mathrm{F}}^2$$

$$\leq \left\langle \nabla\mathcal{L}\left(\mathcal{A}^\star\right) + \nabla\mathcal{Q}_\lambda\left(\mathcal{A}^\star\right) + \lambda_t\mathcal{G}^\star, \widehat{\mathcal{A}} - \mathcal{A}^\star \right\rangle$$

$$\leq \min_{\mathcal{G}^\star \in \partial\|\mathcal{A}^\star\|_1}\left\{\sum_{i_1=1}^{I_1}\cdots\sum_{i_N=1}^{I_N}\left|\left(\nabla\mathcal{L}\left(\mathcal{A}^\star\right) + \nabla\mathcal{Q}_\lambda\left(\mathcal{A}^\star\right) + \lambda_t\mathcal{G}^\star\right)_{i_1\ldots i_N}\right| \cdot \left|\left(\mathcal{A}^\star - \widehat{\mathcal{A}}\right)_{i_1\ldots i_N}\right|\right\}. \tag{17}$$

We proceed by decomposing the summation on the right-hand side of (17) into three distinct parts

- $(i_1, \ldots, i_N) \in \overline{\mathcal{S}_1}$,

- $(i_1, \ldots, i_N) \in \mathcal{S}_1^{\mathrm{I}}$,

- $(i_1, \ldots, i_N) \in \mathcal{S}_1^{\mathrm{II}}$.

Here, $\mathcal{S}_1^{\mathrm{I}} = \{(i_1, \ldots, i_N) \mid |a_{i_1,\ldots,i_N}| \geq \nu\}$, $\mathcal{S}_1^{\mathrm{I}} = \{(i_1, \ldots, i_N) \mid |a_{i_1,\ldots,i_N}| < \nu\}$, and $\nu > 0$ is defined in Assumption 1. (i) For any index $(i_1, \ldots, i_N) \in \overline{\mathcal{S}_1}$, the regularity condition yields

$$\nabla\mathcal{Q}_\lambda a^\star_{i_1\ldots i_N} = q'_\lambda\left(a^\star_{i_1\ldots i_N}\right) = q'_\lambda(0), \quad \text{for} \quad j \in \overline{\mathcal{S}_1}.$$

Assuming that $\|\nabla\mathcal{L}\left(\mathcal{A}^\star\right)\|_{\max} \leq \frac{\lambda}{8}$, it follows that

$$\max_{(i_1,\ldots,i_N) \in \overline{\mathcal{S}_1}}\left|\left(\nabla\mathcal{L}\left(\mathcal{A}^\star\right)\right)_{i_1,\ldots,i_N}\right| \leq \|\nabla\mathcal{L}\left(\mathcal{A}^\star\right)\|_{\max} \leq \frac{\lambda}{8} \leq \lambda.$$

Therefore,

$$\max_{(i_1,\ldots,i_N) \in \overline{\mathcal{S}_1}}\left|\left(\nabla\mathcal{L}\left(\mathcal{A}^\star\right) + \mathcal{Q}_\lambda\left(\mathcal{A}^\star\right)\right)_{i_1,\ldots,i_N}\right| \leq \lambda.$$

Moreover, since $\boldsymbol{\mathcal{G}}^\star \in \partial \|\boldsymbol{\mathcal{A}}^\star\|_1$, it holds that $\lambda \boldsymbol{\mathcal{G}}^\star_{i_1,\ldots,i_N} \in [-\lambda, \lambda]$. Consequently, for each $(i_1,\ldots,i_N) \in \overline{\mathcal{S}_1}$, we can select $\boldsymbol{\mathcal{G}}^\star_{i_1,\ldots,i_N}$ such that

$$\left| \left(\nabla \mathcal{L}\left(\boldsymbol{\mathcal{A}}^\star\right) + \nabla \mathcal{Q}_\lambda\left(\boldsymbol{\mathcal{A}}^\star\right)\right)_{i_1\ldots i_N} + \lambda g^\star_{i_1\ldots i_N} \right| = 0.$$

This implies

$$\min_{\boldsymbol{\mathcal{G}}^\star \in \partial \|\boldsymbol{\mathcal{A}}^\star\|_1} \left\{ \left| \left(\nabla \mathcal{L}\left(\boldsymbol{\mathcal{A}}^\star\right) + \nabla \mathcal{Q}_\lambda\left(\boldsymbol{\mathcal{A}}^\star\right) + \lambda \boldsymbol{\mathcal{G}}^\star\right)_{i_1\ldots i_N} \right| \right\} = 0, \quad \text{for} \quad (i_1 \ldots i_N) \in \overline{\mathcal{S}_1}.$$

Therefore, we obtain

$$\min_{\boldsymbol{\mathcal{G}}^\star \in \partial \|\boldsymbol{\mathcal{A}}^\star\|_1} \left\{ \sum_{(i_1,\ldots,i_N) \in \overline{\mathcal{S}_1}} \left| \left(\nabla \mathcal{L}\left(\boldsymbol{\mathcal{A}}^\star\right) + \nabla \mathcal{Q}_\lambda\left(\boldsymbol{\mathcal{A}}^\star\right) + \lambda \boldsymbol{\mathcal{G}}^\star\right)_{i_1\ldots i_N} \right| \cdot \left| \left(\boldsymbol{\mathcal{A}}^\star - \widehat{\boldsymbol{\mathcal{A}}}\right)_{i_1\ldots i_N} \right| \right\} = 0. \tag{18}$$

(ii) For indices $(i_1,\ldots,i_N) \in \mathcal{S}_1^{\mathrm{I}}$, we have $\left|a^\star_{i_1,\ldots,i_N}\right| \geq \nu$. Given that $R(\boldsymbol{\mathcal{A}}) = \lambda \|\boldsymbol{\mathcal{A}}\|_1 + \mathcal{Q}_\lambda\left(a_{i_1\ldots i_N}\right)$, our assumption on $R(\boldsymbol{\mathcal{A}})$ ensures that

This leads to

$$\min_{\boldsymbol{\mathcal{G}}^\star \in \partial \|\boldsymbol{\mathcal{A}}^\star\|_1} \left\{ \sum_{(i_1\ldots i_N) \in \mathcal{S}_1^{\mathrm{I}}} \left| \left(\nabla \mathcal{L}\left(\boldsymbol{\mathcal{A}}^\star\right) + \nabla \mathcal{Q}_\lambda\left(\boldsymbol{\mathcal{A}}^\star\right) + \lambda \boldsymbol{\mathcal{G}}^\star\right)_{i_1\ldots i_N} \right| \cdot \left| \left(\boldsymbol{\mathcal{A}}^\star - \widehat{\boldsymbol{\mathcal{A}}}\right)_{i_1\ldots i_N} \right| \right\}$$

$$= \sum_{(i_1\ldots i_N) \in \mathcal{S}_1^{\mathrm{I}}} \left| \left(\nabla \mathcal{L}\left(\boldsymbol{\mathcal{A}}^\star\right)\right)_{i_1\ldots i_N} \right| \cdot \left| \left(\boldsymbol{\mathcal{A}}^\star - \widehat{\boldsymbol{\mathcal{A}}}\right)_{i_1\ldots i_N} \right|$$

$$\leq \left\| \left(\nabla \mathcal{L}\left(\boldsymbol{\mathcal{A}}^\star\right)\right)_{\mathcal{S}_1^{\mathrm{I}}} \right\|_{\mathrm{F}} \cdot \left\| \boldsymbol{\mathcal{A}}^\star - \widehat{\boldsymbol{\mathcal{A}}} \right\|_{\mathrm{F}}.$$

(iii) For indices $(i_1,\ldots,i_N) \in \mathcal{S}_1^{\mathrm{II}}$, we have $\left|a^\star_{i_1,\ldots,i_N}\right| < \nu$. Given that $\|\nabla \mathcal{L}\left(\boldsymbol{\mathcal{A}}^\star\right)\|_{\max} \leq \frac{\lambda}{8}$, we have

$$\max_{(i_1\ldots i_N) \in \mathcal{S}_1^{\mathrm{II}}} \left| \left(\nabla \mathcal{L}\left(\boldsymbol{\mathcal{A}}^\star\right)\right)_{i_1\ldots i_N} \right| \leq \left| \left(\nabla \mathcal{L}\left(\boldsymbol{\mathcal{A}}^\star\right)\right)_{i_1\ldots i_N} \right|_{\max} \leq \lambda/8.$$

Meanwhile, we have

$$\max_{(i_1\ldots i_N) \in \mathcal{S}_1^{\mathrm{II}}} \left| \left(\nabla \mathcal{Q}_\lambda\left(\boldsymbol{\mathcal{A}}^\star\right)\right)_{i_1\ldots i_N} \right| = \max_{(i_1\ldots i_N) \in \mathcal{S}_1^{\mathrm{II}}} \left| q'_\lambda\left(a^\star_{i_1\ldots i_N}\right) \right| \leq \max \left| q'_\lambda\left(a^\star_{i_1\ldots i_N}\right) \right| \leq \lambda,$$

Additionally, since $\boldsymbol{\mathcal{G}}^\star \in \partial \|\boldsymbol{\mathcal{A}}^\star\|_1$, it follows that $\left|g^\star_{i_1,\ldots,i_N}\right| \leq 1$. Therefore, for each $(i_1,\ldots,i_N) \in \mathcal{S}_1^{\mathrm{II}}$, we obtain

$$\left| \left(\nabla \mathcal{L}\left(\boldsymbol{\mathcal{A}}^\star\right) + \nabla \mathcal{Q}_\lambda\left(\boldsymbol{\mathcal{A}}^\star\right) + \lambda \boldsymbol{\mathcal{G}}^\star\right)_{i_1\ldots i_N} \right| \leq \max_{(i_1\ldots i_N) \in \mathcal{S}_1^{\mathrm{II}}} \left| \left(\nabla \mathcal{L}\left(\boldsymbol{\mathcal{A}}^\star\right)\right)_{i_1\ldots i_N} \right| + \max_{(i_1\ldots i_N) \in \mathcal{S}_1^{\mathrm{II}}} \left| \left(\nabla \mathcal{Q}_\lambda\left(\boldsymbol{\mathcal{A}}^\star\right)\right)_{i_1\ldots i_N} \right| + \lambda$$

$$\leq 3\lambda,$$

which implies

$$\min_{\boldsymbol{\mathcal{G}}^\star \in \partial \|\boldsymbol{\mathcal{A}}^\star\|_1} \left\{ \sum_{(i_1\ldots i_N) \in \mathcal{S}_1^{\mathrm{II}}} \left| \left(\nabla \mathcal{L}\left(\boldsymbol{\mathcal{A}}^\star\right) + \nabla \mathcal{Q}_\lambda\left(\boldsymbol{\mathcal{A}}^\star\right) + \lambda \boldsymbol{\mathcal{G}}^\star\right)_{i_1\ldots i_N} \right| \cdot \left| \left(\boldsymbol{\mathcal{A}}^\star - \widehat{\boldsymbol{\mathcal{A}}}\right)_{i_1\ldots i_N} \right| \right\}$$

$$\leq 3\lambda \left| \left(\boldsymbol{\mathcal{A}}^\star - \widehat{\boldsymbol{\mathcal{A}}}\right)_{i_1\ldots i_N} \right|$$

$$= 3\lambda \left\| \left(\boldsymbol{\mathcal{A}}^\star - \widehat{\boldsymbol{\mathcal{A}}}\right)_{\overline{\mathcal{S}_1^{\mathrm{II}}}} \right\|_{\mathrm{F}}$$

$$\leq 3\lambda \sqrt{|\mathcal{S}_1|} \left\| \left(\boldsymbol{\mathcal{A}}^\star - \widehat{\boldsymbol{\mathcal{A}}}\right)_{\overline{\mathcal{S}_1^{\mathrm{II}}}} \right\|_{\mathrm{F}}$$

$$\leq 3\lambda \sqrt{|\mathcal{S}_1|} \left\| \boldsymbol{\mathcal{A}}^\star - \widehat{\boldsymbol{\mathcal{A}}} \right\|_{\mathrm{F}}. \tag{19}$$

Substituting the bounds from (18) to (19) into the right-hand side of (17), we obtain

$$\left\| \boldsymbol{\mathcal{A}}^\star - \widehat{\boldsymbol{\mathcal{A}}} \right\|_{\mathrm{F}} \leq \frac{1}{\mu - \zeta} \left( \left\| \left( \nabla \mathcal{L} \left( \boldsymbol{\mathcal{A}}^\star \right) \right)_{\mathcal{S}_1^{\mathrm{I}}} \right\|_{\mathrm{F}} + 3\lambda \sqrt{\left| \mathcal{S}_1^{\mathrm{II}} \right|} \right).$$

$\square$

**Lemma 4.** *For least-squares regression with sub-Gaussian noise, we assume that the columns of $\widetilde{\boldsymbol{\mathcal{X}}}$ are normalized in such a way that $\max_{j \in \{1, \ldots, d_1 \times d_2 \times \cdots \times d_N\}} \left\| \widetilde{\boldsymbol{\mathcal{X}}}_{\cdot j} \right\|_2 \leq \sqrt{n}$, where $\widetilde{\boldsymbol{\mathcal{X}}} = \left( \mathrm{vec} \left( \boldsymbol{\mathcal{X}}^{(1)} \right), \ldots, \mathrm{vec} \left( \boldsymbol{\mathcal{X}}^{(n)} \right) \right)^\top$. If $\lambda \asymp \sqrt{\frac{\log(d_1 d_2 \cdots d_N)}{n}}$, then we have*

$$\left\| \nabla \mathcal{L} \left( \boldsymbol{\mathcal{A}}^\star \right) \right\|_{\mathrm{F}} \lesssim \sqrt{\frac{|\mathcal{S}_1|}{n}}.$$

*Proof.* We begin by establishing an upper bound on the probability that the maximum entry of the gradient $\mathbb{P} \left( \left\| \nabla \mathcal{L} \left( \boldsymbol{\mathcal{A}} \right) \right\|_{\max} \geq \frac{\lambda}{8} \right)$, where $\nabla \mathcal{L} \left( \boldsymbol{\mathcal{A}} \right) = \frac{1}{n} \left\langle \widetilde{\boldsymbol{\mathcal{X}}}, \widetilde{\boldsymbol{\mathcal{E}}} \right\rangle$ and $\widetilde{\boldsymbol{\mathcal{E}}} = \left( \mathrm{vec} \left( \boldsymbol{\mathcal{E}}^{(1)} \right), \ldots, \mathrm{vec} \left( \boldsymbol{\mathcal{E}}^{(n)} \right) \right)^\top$.

For $\lambda \asymp \sqrt{\frac{\log(d_1 d_2 \cdots d_N)}{n}}$, using the union bound, we obtain

$$\mathbb{P} \left( \left\| \nabla \mathcal{L} \left( \boldsymbol{\mathcal{A}} \right) \right\|_{\max} \geq \frac{\lambda}{8} \right) \leq \mathbb{P} \left( \left\| \frac{1}{n} \left\langle \widetilde{\boldsymbol{\mathcal{X}}}, \widetilde{\boldsymbol{\mathcal{E}}} \right\rangle \right\|_{\max} \geq \frac{c\sqrt{\log d / n}}{8} \right)$$

$$\leq \sum_{j=1}^{d_1 \times d_2 \times \cdots \times d_N} \mathbb{P} \left( \left| \frac{1}{n} \left\langle \widetilde{\boldsymbol{\mathcal{X}}}, \widetilde{\boldsymbol{\mathcal{E}}} \right\rangle \right|_j \geq \frac{c\sqrt{\log d / n}}{8} \right).$$

Let's define $\theta_k = \left| \left\langle \widetilde{\boldsymbol{\mathcal{X}}}, \widetilde{\boldsymbol{\mathcal{E}}} \right\rangle \right|_k$, where $k$ is composite coordinate. Since $\widetilde{\boldsymbol{\mathcal{E}}}_j$ is sub-Gaussian$\left( 0, \eta^2 \right)$, it follows that for any $t_0 > 0$,

$$\mathbb{E} \left( \exp \left\{ t_0 \theta_k \right\} + \exp \left\{ -t_0 \theta_k \right\} \right) \leq 2 \exp \left\{ \frac{1}{n^2} \left\| \widetilde{\boldsymbol{\mathcal{X}}}_{\cdot k} \right\|^2 \eta^2 t_0^2 / 2 \right\}.$$

Taking $t_0 = \frac{t n^2}{\left\| \widetilde{\boldsymbol{\mathcal{X}}}_{\cdot k} \right\|^2 \eta^2 t_0^2}$ yields that

$$\mathbb{P} \left( |\theta_k| \geq t \right) \leq 2 \exp \left\{ -\frac{n^2 t^2}{2 \left\| \widetilde{\boldsymbol{\mathcal{X}}}_{\cdot k} \right\|^2 \eta^2} \right\}.$$

Further taking $t = \frac{\lambda}{8}$ results

$$\mathbb{P} \left( \left\| \nabla \mathcal{L} \left( \boldsymbol{\mathcal{A}} \right) \right\|_{\max} \geq \frac{\lambda}{8} \right) \leq 2 \left( d_1 \times \cdots \times d_N \right)^{-c^2 / \left( 128 \eta^2 \right)}.$$

Applying the Hanson-Wright inequality yields that

$$\mathbb{P} \left( \left| \left\langle \boldsymbol{\mathcal{E}}, \left\langle \boldsymbol{\mathcal{A}}, \boldsymbol{\mathcal{E}} \right\rangle \right\rangle - \mathbb{E} \left\langle \boldsymbol{\mathcal{E}}, \left\langle \boldsymbol{\mathcal{A}}, \boldsymbol{\mathcal{E}} \right\rangle \right\rangle \right| > \mathbb{E} \left\langle \boldsymbol{\mathcal{E}}, \left\langle \boldsymbol{\mathcal{A}}, \boldsymbol{\mathcal{E}} \right\rangle \right\rangle \right)$$

$$\leq 2 \exp \left[ -C \min \left\{ \frac{\mathbb{E} \left\langle \boldsymbol{\mathcal{E}}, \left\langle \boldsymbol{\mathcal{A}}, \boldsymbol{\mathcal{E}} \right\rangle \right\rangle}{\eta^2 \left\| \boldsymbol{\mathcal{A}} \right\|_{\mathrm{F}}}, \left( \frac{\mathbb{E} \left\langle \boldsymbol{\mathcal{E}}, \left\langle \boldsymbol{\mathcal{A}}, \boldsymbol{\mathcal{E}} \right\rangle \right\rangle}{\eta^2 \left\| \boldsymbol{\mathcal{A}} \right\|_{\mathrm{F}}} \right)^2 \right\} \right],$$

where $C$ is a universal constant.

Combining the above two inequalities, we have

$$\left\|\nabla\mathcal{L}\left(\boldsymbol{\mathcal{A}}^{\star}\right)\right\|_{\mathrm{F}} = \sqrt{\frac{\left\langle\widetilde{\boldsymbol{\mathcal{E}}},\widetilde{\boldsymbol{\mathcal{X}}}\right\rangle}{n}} \leq \sqrt{\frac{2\mathbb{E}\left\langle\widetilde{\boldsymbol{\mathcal{E}}},\widetilde{\boldsymbol{\mathcal{X}}}\right\rangle}{n}} \leq \sqrt{2L}\eta\sqrt{\frac{|\mathcal{S}_1|}{n}}. \tag{20}$$

$\square$

Building upon Lemma 1 through Lemma 4, we derive Theorem 5.

## B.2. Proof of Theorem 7

We begin by demonstrating that the fiber-wise sparsity regularizer can be reformulated as the sum of the $\ell_1$ penalty and a concave part. Specifically, we have:

$$\mathcal{R}_\lambda\left(\boldsymbol{\mathcal{A}}\right) = \sum_{l=1}^{\Pi_{j\neq k}d_j} p_\lambda\left(\left\|\left[\boldsymbol{A}_{(k)}\right]_{\cdot,l}\right\|_2\right) = \sum_{l=1}^{\Pi_{j\neq k}d_j} \lambda\left\|\left[\boldsymbol{A}_{(k)}\right]_{\cdot,l}\right\|_2 + \mathcal{Q}_\lambda\left(\left\|\left[\boldsymbol{A}_{(k)}\right]_{\cdot,l}\right\|_2\right),$$

where $\mathcal{Q}_\lambda\left(\left\|\left[\boldsymbol{A}_{(k)}\right]_{\cdot,l}\right\|_2\right) = \sum_{l=1}^{\Pi_{j\neq k}d_j} q_\lambda\left(\left\|\left[\boldsymbol{A}_{(k)}\right]_{\cdot,l}\right\|_2\right)$.

**Lemma 5.** *Under Assumptions 2 and 3, the loss function $\widetilde{\mathcal{L}}\left(\boldsymbol{\mathcal{A}}\right)$ satisfies the restricted strong convexity*

$$\widetilde{\mathcal{L}}\left(\boldsymbol{\mathcal{A}}'\right) \geq \widetilde{\mathcal{L}}\left(\boldsymbol{\mathcal{A}}\right) + \left\langle\nabla\widetilde{\mathcal{L}}\left(\boldsymbol{\mathcal{A}}\right),\boldsymbol{\mathcal{A}}'-\boldsymbol{\mathcal{A}}\right\rangle + \frac{\mu-\zeta}{2}\left\|\boldsymbol{\mathcal{A}}'-\boldsymbol{\mathcal{A}}\right\|_{\mathrm{F}}^2,$$

*and the restricted smoothness*

$$\widetilde{\mathcal{L}}\left(\boldsymbol{\mathcal{A}}'\right) \leq \widetilde{\mathcal{L}}\left(\boldsymbol{\mathcal{A}}\right) + \left\langle\nabla\widetilde{\mathcal{L}}\left(\boldsymbol{\mathcal{A}}\right),\boldsymbol{\mathcal{A}}'-\boldsymbol{\mathcal{A}}\right\rangle + \frac{L}{2}\left\|\boldsymbol{\mathcal{A}}'-\boldsymbol{\mathcal{A}}\right\|_{\mathrm{F}}^2.$$

*Proof.* Since the proof closely mirrors that of Lemma 1, it is omitted here for brevity. $\square$

**Lemma 6.** *Suppose there exists an integer $\widetilde{s}_3 > C\left|\mathcal{S}_3\right|$, where $C$ is a constant, and that $\boldsymbol{\mathcal{A}}$ satisfies $\left\|\boldsymbol{\mathcal{A}}_{\overline{\mathcal{S}_3}}\right\|_0 \leq \widetilde{s}_3$, $\omega\left(\boldsymbol{\mathcal{A}}\right) \leq \frac{\lambda}{2}$, where $\omega\left(\boldsymbol{\mathcal{A}}\right) = \min\limits_{\boldsymbol{\mathcal{G}}\in\partial\left\|\left[\boldsymbol{A}_{(k)}\right]_{\cdot,l}\right\|_2}\left\{\left\|\nabla\widetilde{\mathcal{L}}\left(\boldsymbol{\mathcal{A}}\right)+\lambda\boldsymbol{\mathcal{G}}\right\|_{\max}\right\}$, and $\left\|\nabla\mathcal{L}\left(\boldsymbol{\mathcal{A}}^*\right)\right\|_{\max} \leq \lambda/8$. Under Assumptions 2 and 3, $\boldsymbol{\mathcal{A}}$ satisfies*

$$\left\|\boldsymbol{\mathcal{A}}-\boldsymbol{\mathcal{A}}^{\star}\right\|_{\mathrm{F}} \leq \frac{21/8}{\mu-\zeta}\lambda\sqrt{|\mathcal{S}_3|}.$$

*Proof.* We omit the proof here for brevity, as it closely mirrors that of Lemma 2. $\square$

**Lemma 7.** *Consider the regularization parameter $\lambda$ and assume that the derivative of the non-convex penalty satisfies $p'_\lambda\left(\left\|\left[\boldsymbol{A}_{(k)}\right]_{\cdot,l}\right\|_2\right) = 0$ whenever $\left\|\left[\boldsymbol{A}_{(k)}\right]_{\cdot,l}\right\|_2 \geq \nu$ for some $\nu > 0$. Let $\mathcal{S}_3^{\mathrm{I}}\cup\mathcal{S}_3^{\mathrm{II}} = \mathcal{S}_3$. For indices $(i_1,\ldots,i_N) \in \mathcal{S}_3^{\mathrm{I}} \subseteq \mathcal{S}_3$, we assume $\min\limits_l\left[\boldsymbol{A}_{(k)}^{\star}\right]_{\cdot,l} \geq \nu$, and for indices $(i_1,\ldots,i_N) \in \mathcal{S}_3^{\mathrm{II}} \subseteq \mathcal{S}_3$, we assume $\min\limits_l\left[\boldsymbol{A}_{(k)}^{\star}\right]_{\cdot,l} \leq \nu$. Under Assumptions 2~4, we derive the following bound:*

$$\left\|\widehat{\boldsymbol{\mathcal{A}}}-\boldsymbol{\mathcal{A}}^{\star}\right\|_{\mathrm{F}} \leq \frac{1}{\mu-\zeta}\left\|\left(\nabla\mathcal{L}\left(\boldsymbol{\mathcal{A}}^{\star}\right)\right)_{\mathcal{S}_3^{\mathrm{I}}}\right\|_{\mathrm{F}} + \frac{3}{\mu-\zeta}\lambda\sqrt{\left|\mathcal{S}_3^{\mathrm{II}}\right|}.$$

*Proof.* For brevity, we omit the proof here, as it closely resembles that of Lemma 3. $\square$

**Lemma 8.** *For least-squares regression with sub-Gaussian noise, we assume that the columns of $\widetilde{\boldsymbol{\mathcal{X}}}$ are normalized in such a way that $\max_{j\in\{1,\ldots,d_1d_2\ldots d_N\}}\left\|\widetilde{\boldsymbol{\mathcal{X}}}_{\cdot j}\right\|_2 \leq \sqrt{n}$, where $\widetilde{\boldsymbol{\mathcal{X}}} = \left(\mathrm{vec}\left(\boldsymbol{\mathcal{X}}^{(1)}\right),\ldots,\mathrm{vec}\left(\boldsymbol{\mathcal{X}}^{(n)}\right)\right)^{\top}$. If $\lambda \asymp \sqrt{\frac{\log d_k}{n}}$, then we have*

$$\left\|\nabla\mathcal{L}\left(\boldsymbol{\mathcal{A}}^{\star}\right)\right\|_{\mathrm{F}} \lesssim \sqrt{\frac{|\mathcal{S}_3|}{n}}.$$

*Proof.* For brevity, the proof is omitted here as it closely follows the methodology established in Lemma 4. $\square$

## B.3. Proof of Theorem 8

We begin by demonstrating that the fiber-wise sparsity regularizer can be reformulated as the sum of the $\ell_1$ penalty and a concave part. Specifically, we have:

$$
\mathcal{R}_\lambda\left(\boldsymbol{\mathcal{A}}\right) = \sum_{i=1}^{\Pi_{s\neq j,k} d_s} p_\lambda\left(\left\|\left[\boldsymbol{\mathcal{A}}_{(j,k)}\right]_{\cdot,\cdot,l}\right\|_{\mathrm{F}}\right) = \sum_{i=1}^{\Pi_{s\neq j,k} d_s} \lambda\left\|\left[\boldsymbol{\mathcal{A}}_{(j,k)}\right]_{\cdot,\cdot,l}\right\|_{\mathrm{F}} + \mathcal{Q}_\lambda\left(\left\|\left[\boldsymbol{\mathcal{A}}_{(j,k)}\right]_{\cdot,\cdot,l}\right\|_{\mathrm{F}}\right),
$$

where $\mathcal{Q}_\lambda\left(\left\|\left[\boldsymbol{\mathcal{A}}_{(j,k)}\right]_{\cdot,\cdot,l}\right\|_{\mathrm{F}}\right) = \sum_{i=1}^{\Pi_{s\neq j,k} d_s} q_\lambda\left(\left\|\left[\boldsymbol{\mathcal{A}}_{(j,k)}\right]_{\cdot,\cdot,l}\right\|_{\mathrm{F}}\right).$

**Lemma 9.** *Under Assumptions 2 and 3, the loss function $\widetilde{\mathcal{L}}\left(\boldsymbol{\mathcal{A}}\right)$ satisfies the restricted strong convexity*

$$
\widetilde{\mathcal{L}}\left(\boldsymbol{\mathcal{A}}'\right) \geq \widetilde{\mathcal{L}}\left(\boldsymbol{\mathcal{A}}\right) + \left\langle\nabla\widetilde{\mathcal{L}}\left(\boldsymbol{\mathcal{A}}\right), \boldsymbol{\mathcal{A}}' - \boldsymbol{\mathcal{A}}\right\rangle + \frac{\mu - \zeta}{2}\left\|\boldsymbol{\mathcal{A}}' - \boldsymbol{\mathcal{A}}\right\|_{\mathrm{F}}^2,
$$

*and the restricted smoothness*

$$
\widetilde{\mathcal{L}}\left(\boldsymbol{\mathcal{A}}'\right) \leq \widetilde{\mathcal{L}}\left(\boldsymbol{\mathcal{A}}\right) + \left\langle\nabla\widetilde{\mathcal{L}}\left(\boldsymbol{\mathcal{A}}\right), \boldsymbol{\mathcal{A}}' - \boldsymbol{\mathcal{A}}\right\rangle + \frac{L}{2}\left\|\boldsymbol{\mathcal{A}}' - \boldsymbol{\mathcal{A}}\right\|_{\mathrm{F}}^2.
$$

*Proof.* The proof can be demonstrated similarly to the proof in Lemma 1. Hence, we omit it here. $\qquad\square$

**Lemma 10.** *Suppose there exists an integer $\widetilde{s}_4 > C\left|\mathcal{S}_4\right|$, where $C$ is a constant, and that $\boldsymbol{\mathcal{A}}$ satisfies $\left\|\boldsymbol{\mathcal{A}}_{\overline{\mathcal{S}_4}}\right\|_0 \leq \widetilde{s}_4$, $\omega\left(\boldsymbol{\mathcal{A}}\right) \leq \frac{\lambda}{2}$, where $\omega\left(\boldsymbol{\mathcal{A}}\right) = \min\limits_{\boldsymbol{\mathcal{G}}\in\partial\left\|\left[\boldsymbol{\mathcal{A}}_{(j,k)}\right]_{\cdot,\cdot,l}\right\|_{\mathrm{F}}}\left\{\left\|\nabla\widetilde{\mathcal{L}}\left(\boldsymbol{\mathcal{A}}\right) + \lambda\boldsymbol{\mathcal{G}}\right\|_{\max}\right\}$, and $\left\|\nabla\mathcal{L}\left(\boldsymbol{\mathcal{A}}^*\right)\right\|_{\max} \leq \lambda/8$. Under Assumptions 2 and 3, $\boldsymbol{\mathcal{A}}$ satisfies*

$$
\left\|\boldsymbol{\mathcal{A}} - \boldsymbol{\mathcal{A}}^\star\right\|_{\mathrm{F}} \leq \frac{21/8}{\mu - \zeta}\lambda\sqrt{\left|\mathcal{S}_4\right|}.
$$

*Proof.* For the sake of brevity, we omit the proof here, as it closely follows that of Lemma 2. $\qquad\square$

**Lemma 11.** *Consider the regularization parameter $\lambda$ and assume that the derivative of the non-convex penalty satisfies $p'_\lambda\left(\left\|\left[\boldsymbol{\mathcal{A}}_{(j,k)}\right]_{\cdot,\cdot,l}\right\|_{\mathrm{F}}\right) = 0$ whenever $\left\|\left[\boldsymbol{\mathcal{A}}_{(j,k)}\right]_{\cdot,\cdot,l}\right\|_{\mathrm{F}} \geq \nu$ for some $\nu > 0$. Let $\mathcal{S}_4^{\mathrm{I}} \cup \mathcal{S}_4^{\mathrm{II}} = \mathcal{S}_4$. For indices $(i_1,\ldots,i_N) \in \mathcal{S}_4^{\mathrm{I}} \subseteq \mathcal{S}_4$, we assume $\min\limits_l\left\|\left[\boldsymbol{\mathcal{A}}_{(j,k)}\right]_{\cdot,\cdot,l}\right\|_{\mathrm{F}} \geq \nu$, and for indices $(i_1,\ldots,i_N) \in \mathcal{S}_4^{\mathrm{II}} \subseteq \mathcal{S}_4$, we assume $\min\limits_l\left\|\left[\boldsymbol{\mathcal{A}}_{(j,k)}\right]_{\cdot,\cdot,l}\right\|_{\mathrm{F}} \leq \nu$. Under Assumptions 2$\sim$4, we derive the following bound:*

$$
\left\|\widehat{\boldsymbol{\mathcal{A}}} - \boldsymbol{\mathcal{A}}^\star\right\|_{\mathrm{F}} \leq \frac{1}{\mu - \zeta}\left\|\left(\nabla\mathcal{L}\left(\boldsymbol{\mathcal{A}}^\star\right)\right)_{\mathcal{S}_4^{\mathrm{I}}}\right\|_{\mathrm{F}} + \frac{3}{\mu - \zeta}\lambda\sqrt{\left|\mathcal{S}_4^{\mathrm{II}}\right|}.
$$

*Proof.* For brevity, we omit the proof here, as it closely resembles that of Lemma 3. $\qquad\square$

**Lemma 12.** *For least-squares regression with sub-Gaussian noise, we assume that the columns of $\widetilde{\boldsymbol{\mathcal{X}}}$ are normalized in such a way that $\max_{j\in\{1,\ldots,d_1 d_2\ldots d_N\}}\left\|\widetilde{\boldsymbol{\mathcal{X}}}_{\cdot j}\right\|_2 \leq \sqrt{n}$, where $\widetilde{\boldsymbol{\mathcal{X}}} = \left(\mathrm{vec}\left(\boldsymbol{\mathcal{X}}^{(1)}\right), \ldots, \mathrm{vec}\left(\boldsymbol{\mathcal{X}}^{(n)}\right)\right)^\top$. If $\lambda \asymp \sqrt{\frac{\log(d_j d_k)}{n}}$, then we have*

$$
\left\|\nabla\mathcal{L}\left(\boldsymbol{\mathcal{A}}^\star\right)\right\|_{\mathrm{F}} \lesssim \sqrt{\frac{\left|\mathcal{S}_4\right|}{n}}.
$$

*Proof.* For brevity, the proof is omitted here as it closely follows the methodology established in Lemma 4. $\qquad\square$

## B.4. Proof of Theorem 6

The proposed mode-wise low-rankness penalty can be reformulated as the sum of a scaled norm and a concave function. Specifically, we have

$$\mathcal{R}_\lambda \left(\boldsymbol{\mathcal{A}}\right) = \sum_{i=1}^{\min\left\{d_k, \prod_{j\neq k} d_j\right\}} p_\lambda \left(\sigma_i \left(\boldsymbol{A}_{(k)}\right)\right) = \lambda \left\|\boldsymbol{A}_{(k)}\right\|_{\mathrm{nuc}} + \mathcal{Q}_\lambda \left(\boldsymbol{A}_{(k)}\right),$$

where $\sigma_i \left(\boldsymbol{A}_{(k)}\right)$ denotes the $i$-th singular value of the mode-$(k)$ unfolding $\boldsymbol{A}_{(k)}$. For the estimation problem, we define

$$\widetilde{\mathcal{L}} \left(\boldsymbol{\mathcal{A}}\right) = \mathcal{L} \left(\boldsymbol{\mathcal{A}}\right) + \mathcal{Q}_\lambda \left(\boldsymbol{A}_{(k)}\right),$$

where $\mathcal{Q}_\lambda \left(\boldsymbol{A}_{(k)}\right) = \sum_{i=1}^{\min\left\{I_k, \prod_{j\neq k} d_j\right\}} q_\lambda \left(\sigma_i \left(\boldsymbol{A}_{(k)}\right)\right)$.

Based on the restrict strongly convexity of $\mathcal{L}\left(\cdot\right)$ in Assumption 2 and the parameter for regularity condition in Assumption 1, if $\mu > \zeta$, we have the restrict strongly convexity of $\widetilde{\mathcal{L}}\left(\cdot\right)$.

Besides, for the RSC and RSM assumption, we define the following cone of directions

$$\mathcal{C} = \left\{\boldsymbol{\mathcal{B}} \in \mathbb{R}^{d_1 \cdots d_N} \mid \left\|\Pi_{\mathcal{F}^\perp} \left(\boldsymbol{\mathcal{B}}\right)\right\|_{\mathrm{nuc}} \leq 5 \left\|\Pi_{\mathcal{F}} \left(\boldsymbol{\mathcal{B}}\right)\right\|_{\mathrm{nuc}}\right\}$$

**Lemma 13.** *Under Assumption 2, if* $\boldsymbol{\mathcal{B}} \in \mathcal{C}$,*we have*

$$\widetilde{\mathcal{L}} \left(\boldsymbol{\mathcal{A}} + \boldsymbol{\mathcal{B}}\right) \geq \widetilde{\mathcal{L}} \left(\boldsymbol{\mathcal{A}}\right) + \left\langle \nabla\widetilde{\mathcal{L}} \left(\boldsymbol{\mathcal{A}}\right), \boldsymbol{\mathcal{B}}\right\rangle + \frac{\mu - \zeta}{2} \left\|\boldsymbol{\mathcal{B}}\right\|_{\mathrm{F}}^2.$$

*Proof.* Based on Assumption 2, we have

$$\mathcal{L} \left(\boldsymbol{\mathcal{A}} + \boldsymbol{\mathcal{B}}\right) \leq \mathcal{L} \left(\boldsymbol{\mathcal{A}}\right) + \left\langle \nabla\mathcal{L} \left(\boldsymbol{\mathcal{A}}\right), \boldsymbol{\mathcal{B}}\right\rangle + \frac{\mu}{2} \left\|\boldsymbol{\mathcal{B}}\right\|_{\mathrm{F}}. \tag{21}$$

Moreover, considering the singular values of the unfolded matrices $\boldsymbol{A}_{(k)}$ and $\boldsymbol{B}_{(k)}$, we obtain

$$-\zeta \leq \frac{q_\lambda' \left(\sigma_i \left(\boldsymbol{A}_{(k)}\right)\right) - q_\lambda' \left(\sigma_i \left([\boldsymbol{\mathcal{A}} + \boldsymbol{\mathcal{B}}]_{(k)}\right)\right)}{\sigma_i \left(\boldsymbol{A}_{(k)}\right) - \sigma_i \left([\boldsymbol{\mathcal{A}} + \boldsymbol{\mathcal{B}}]_{(k)}\right)},$$

which is similar to the proof for Lemma 1. This inequality leads to

$$\left\langle \left(-\nabla\mathcal{Q}_\lambda \left(\boldsymbol{A}_{(k)}\right)\right) - \left(-\nabla\mathcal{Q}_\lambda \left([\boldsymbol{\mathcal{A}} + \boldsymbol{\mathcal{B}}]_{(k)}\right)\right), \boldsymbol{B}_{(k)}\right\rangle \leq \zeta \left\|\boldsymbol{B}_{(k)}\right\|_{\mathrm{F}}.$$

This inequality characterizes the smoothness of $-Q(\cdot)$, which is equivalent to

$$\mathcal{Q}_\lambda \left([\boldsymbol{\mathcal{A}} + \boldsymbol{\mathcal{B}}]_{(k)}\right) = \mathcal{Q}_\lambda \left(\boldsymbol{A}_{(k)} + \boldsymbol{B}_{(k)}\right) \geq \mathcal{Q}_\lambda \left(\boldsymbol{A}_{(k)}\right) + \left\langle \nabla\mathcal{Q}_\lambda \left(\boldsymbol{A}_{(k)}\right), \boldsymbol{B}_{(k)}\right\rangle - \frac{\zeta}{2} \left\|\boldsymbol{B}_{(k)}\right\|_{\mathrm{F}}^2. \tag{22}$$

Noting that the Frobenius norm satisfies $\|\boldsymbol{B}_{(k)}\|_{\mathrm{F}}^2 = \|\boldsymbol{\mathcal{B}}\|_{\mathrm{F}}^2$. Let $\boldsymbol{\mathcal{A}}' = \boldsymbol{\mathcal{A}} + \boldsymbol{\mathcal{B}}$, adding (21) and (22), we have

$$\begin{aligned}
\widetilde{\mathcal{L}} \left(\boldsymbol{\mathcal{A}}'\right) &= \mathcal{L} \left(\boldsymbol{\mathcal{A}}'\right) + \mathcal{Q}_\lambda \left(\boldsymbol{A}'_{(k)}\right) \\
&\geq \mathcal{L} \left(\boldsymbol{\mathcal{A}}\right) + \mathcal{Q}_\lambda \left(\boldsymbol{A}_{(k)}\right) + \left\langle \nabla\mathcal{L} \left(\boldsymbol{\mathcal{A}}\right), \boldsymbol{\mathcal{B}}\right\rangle + \left\langle \nabla\mathcal{Q}_\lambda \left(\boldsymbol{A}_{(k)}\right), \boldsymbol{B}_{(k)}\right\rangle + \frac{\mu - \zeta}{2} \left\|\boldsymbol{\mathcal{B}}\right\|_{\mathrm{F}}^2 \\
&= \widetilde{\mathcal{L}} \left(\boldsymbol{\mathcal{A}}\right) + \left\langle \nabla\mathcal{L} \left(\boldsymbol{\mathcal{A}}\right), \boldsymbol{\mathcal{B}}\right\rangle + \left\langle \nabla\mathcal{Q}_\lambda \left(\boldsymbol{A}_{(k)}\right), \boldsymbol{B}_{(k)}\right\rangle + \frac{\mu - \zeta}{2} \left\|\boldsymbol{\mathcal{B}}\right\|_{\mathrm{F}}^2.
\end{aligned}$$

$\square$

**Lemma 14.** *Under Assumption 2, if $\mu > \zeta$ and the regularization parameter $\lambda \geq \frac{\left\|\mathfrak{X}^\star(\mathcal{E})_{(k)}\right\|_{\mathrm{sp}}}{2n}$, we have*

$$\left\|\Pi_{\mathcal{F}^\perp}\left(\widehat{\boldsymbol{A}}_{(k)} - \boldsymbol{A}^\star_{(k)}\right)\right\|_{\mathrm{nuc}} \leq 5\left\|\Pi_{\mathcal{F}}\left(\widehat{\boldsymbol{A}}_{(k)} - \boldsymbol{A}^\star_{(k)}\right)\right\|_{\mathrm{nuc}}.$$

*Proof.* By Lemma 13 , we have

$$\widetilde{\mathcal{L}}(\widehat{\boldsymbol{\mathcal{A}}}) - \widetilde{\mathcal{L}}(\boldsymbol{\mathcal{A}}^\star) \geq \left\langle \nabla\mathcal{L}(\boldsymbol{\mathcal{A}}^\star), \widehat{\boldsymbol{\mathcal{A}}} - \boldsymbol{\mathcal{A}}^\star\right\rangle + \left\langle \nabla\mathcal{Q}_\lambda\left(\boldsymbol{A}^\star_{(k)}\right), \left[\widehat{\boldsymbol{\mathcal{A}}} - \boldsymbol{\mathcal{A}}^\star\right]_{(k)}\right\rangle. \tag{23}$$

We proceed to bound the right-hand side of inequality (23). By decomposing the inner products using the projections onto two orthogonal subspaces, we have

$$\left\langle \nabla\mathcal{L}(\boldsymbol{\mathcal{A}}^\star), \widehat{\boldsymbol{\mathcal{A}}} - \boldsymbol{\mathcal{A}}^\star\right\rangle + \left\langle \nabla\mathcal{Q}_\lambda\left(\boldsymbol{A}^\star_{(k)}\right), \left[\widehat{\boldsymbol{\mathcal{A}}} - \boldsymbol{\mathcal{A}}^\star\right]_{(k)}\right\rangle$$

$$= \left\langle [\nabla\mathcal{L}(\boldsymbol{\mathcal{A}}^\star)]_{(k)} + \nabla\mathcal{Q}_\lambda\left(\boldsymbol{A}^\star_{(k)}\right), \Pi_{\mathcal{F}}\left(\left[\widehat{\boldsymbol{\mathcal{A}}} - \boldsymbol{\mathcal{A}}^\star\right]_{(k)}\right)\right\rangle + \left\langle [\nabla\mathcal{L}(\boldsymbol{\mathcal{A}}^\star)]_{(k)} + \nabla\mathcal{Q}_\lambda\left(\boldsymbol{A}^\star_{(k)}\right), \Pi_{\mathcal{F}^\perp}\left(\left[\widehat{\boldsymbol{\mathcal{A}}} - \boldsymbol{\mathcal{A}}^\star\right]_{(k)}\right)\right\rangle$$

$$\geq -\left\|\Pi_{\mathcal{F}}\left([\nabla\mathcal{L}(\boldsymbol{\mathcal{A}}^\star)]_{(k)} + \nabla\mathcal{Q}_\lambda\left(\boldsymbol{A}^\star_{(k)}\right)\right)\right\|_{\mathrm{sp}}\left\|\Pi_{\mathcal{F}}\left(\left[\widehat{\boldsymbol{\mathcal{A}}} - \boldsymbol{\mathcal{A}}^\star\right]_{(k)}\right)\right\|_{\mathrm{nuc}} \tag{24}$$

$$-\left\|\Pi_{\mathcal{F}^\perp}\left([\nabla\mathcal{L}(\boldsymbol{\mathcal{A}}^\star)]_{(k)} + \nabla\mathcal{Q}_\lambda\left(\boldsymbol{A}^\star_{(k)}\right)\right)\right\|_{\mathrm{sp}}\left\|\Pi_{\mathcal{F}^\perp}\left(\left[\widehat{\boldsymbol{\mathcal{A}}} - \boldsymbol{\mathcal{A}}^\star\right]_{(k)}\right)\right\|_{\mathrm{nuc}}. \tag{25}$$

For (24), due to $\lambda \geq \frac{1}{2n}\left\|[\mathfrak{X}^\star(\mathcal{E})]_{(k)}\right\|_{\mathrm{sp}}$, we see that $\left\|[\nabla\mathcal{L}(\boldsymbol{\mathcal{A}}^\star)]_{(k)}\right\|_{\mathrm{sp}} \leq \lambda/2$. According to Assumption 1, we have

$$\left\|\Pi_{\mathcal{F}}\left([\nabla\mathcal{L}(\boldsymbol{\mathcal{A}}^\star)]_{(k)} + \nabla\mathcal{Q}_\lambda\left(\boldsymbol{A}^\star_{(k)}\right)\right)\right\|_{\mathrm{sp}} \leq \frac{3}{2}\lambda. \tag{26}$$

For (25), since $\Pi_{\mathcal{F}^\perp}\left(\boldsymbol{A}^\star_{(k)}\right) = 0$, we obtain

$$\left\|\Pi_{\mathcal{F}^\perp}\left([\nabla\mathcal{L}(\boldsymbol{\mathcal{A}}^\star)]_{(k)} + \nabla\mathcal{Q}_\lambda\left(\boldsymbol{A}^\star_{(k)}\right)\right)\right\|_{\mathrm{sp}} \leq \frac{1}{2}\lambda. \tag{27}$$

Combine (26) and (27), we have

$$\left\langle \nabla\mathcal{L}(\boldsymbol{\mathcal{A}}^\star), \widehat{\boldsymbol{\mathcal{A}}} - \boldsymbol{\mathcal{A}}^\star\right\rangle + \left\langle \nabla\mathcal{Q}_\lambda\left(\boldsymbol{A}^\star_{(k)}\right), \left[\widehat{\boldsymbol{\mathcal{A}}} - \boldsymbol{\mathcal{A}}^\star\right]_{(k)}\right\rangle$$

$$\geq -\frac{3}{2}\lambda\left\|\Pi_{\mathcal{F}}\left(\left[\widehat{\boldsymbol{\mathcal{A}}} - \boldsymbol{\mathcal{A}}^\star\right]_{(k)}\right)\right\|_{\mathrm{nuc}} - \frac{1}{2}\lambda\left\|\Pi_{\mathcal{F}^\perp}\left(\left[\widehat{\boldsymbol{\mathcal{A}}} - \boldsymbol{\mathcal{A}}^\star\right]_{(k)}\right)\right\|_{\mathrm{nuc}}$$

Moreover, noting that $\lambda\left\|\widehat{\boldsymbol{\mathcal{A}}}\right\|_{\mathrm{nuc}} - \lambda\left\|\boldsymbol{\mathcal{A}}^\star\right\|_{\mathrm{nuc}} \geq -\lambda\left\|\Pi_{\mathcal{F}}\left(\left[\widehat{\boldsymbol{\mathcal{A}}} - \boldsymbol{\mathcal{A}}^\star\right]_{(k)}\right)\right\| + \lambda\left\|\Pi_{\mathcal{F}^\perp}\left(\left[\widehat{\boldsymbol{\mathcal{A}}} - \boldsymbol{\mathcal{A}}^\star\right]_{(k)}\right)\right\|_{\mathrm{nuc}}$, and combining with (23) , we obtain

$$\left\langle \nabla\mathcal{L}(\boldsymbol{\mathcal{A}}^\star) + \nabla\mathcal{Q}_\lambda\left(\boldsymbol{A}^\star_{(k)}\right), \widehat{\boldsymbol{\mathcal{A}}} - \boldsymbol{\mathcal{A}}^\star\right\rangle + \lambda\left\|\widehat{\boldsymbol{\mathcal{A}}}\right\|_{\mathrm{nuc}} - \lambda\left\|\boldsymbol{\mathcal{A}}^\star\right\|_{\mathrm{nuc}}$$

$$\geq -\frac{5}{2}\lambda\left\|\Pi_{\mathcal{F}}\left(\left[\widehat{\boldsymbol{\mathcal{A}}} - \boldsymbol{\mathcal{A}}^\star\right]_{(k)}\right)\right\|_{\mathrm{nuc}} + \frac{1}{2}\lambda\left\|\Pi_{\mathcal{F}^\perp}\left(\left[\widehat{\boldsymbol{\mathcal{A}}} - \boldsymbol{\mathcal{A}}^\star\right]_{(k)}\right)\right\|_{\mathrm{nuc}}. \tag{28}$$

Since $\widehat{\boldsymbol{\mathcal{A}}}$ is the global minimizer of the general estimator (2) and given that $\mu > \zeta$, it follows that

$$\widetilde{\mathcal{L}}\left(\widehat{\boldsymbol{\mathcal{A}}}\right) + \lambda\left\|\widehat{\boldsymbol{\mathcal{A}}}\right\|_{\mathrm{nuc}} - \widetilde{\mathcal{L}}(\boldsymbol{\mathcal{A}}^\star) - \lambda\left\|\boldsymbol{\mathcal{A}}^\star\right\|_{\mathrm{nuc}} \leq 0. \tag{29}$$

Substituting (23) and (29) into (28), we obtain

$$
\frac{1}{2}\lambda\left\|\Pi_{\mathcal{F}^{\perp}}\left(\left[\widehat{\boldsymbol{\mathcal{A}}}-\boldsymbol{\mathcal{A}}^{\star}\right]_{(k)}\right)\right\|_{\mathrm{nuc}} \leq \frac{5}{2}\lambda\left\|\Pi_{\mathcal{F}}\left(\left[\widehat{\boldsymbol{\mathcal{A}}}-\boldsymbol{\mathcal{A}}^{\star}\right]_{(k)}\right)\right\|_{\mathrm{nuc}}.
$$

Since $\lambda > 0$, we obtain

$$
\left\|\Pi_{\mathcal{F}^{\perp}}\left(\left[\widehat{\boldsymbol{\mathcal{A}}}-\boldsymbol{\mathcal{A}}^{\star}\right]_{(k)}\right)\right\|_{\mathrm{nuc}} \leq 5\left\|\Pi_{\mathcal{F}}\left(\left[\widehat{\boldsymbol{\mathcal{A}}}-\boldsymbol{\mathcal{A}}^{\star}\right]_{(k)}\right)\right\|_{\mathrm{nuc}}.
$$

$\square$

**Lemma 15.** *Considering the mode-wise low-rankness regularizer, under Assumptions 4∼1, for the estimated parameter tensor $\widehat{\boldsymbol{\mathcal{A}}}$ and the true parameter tensor $\boldsymbol{\mathcal{A}}^{\star}$, we have*

$$
\left\|\widehat{\boldsymbol{\mathcal{A}}}-\boldsymbol{\mathcal{A}}^{\star}\right\|_{\mathrm{F}} \leq \frac{1}{\mu-\zeta}\left[\sqrt{\left|\mathcal{S}_4^{\mathrm{I}}\right|}\left\|\Pi_{\mathcal{F}}\left([\nabla\mathcal{L}\left(\boldsymbol{\mathcal{A}}^{\star}\right)]_{(k)}\right)\right\|_{\mathrm{sp}} + 3\lambda\sqrt{\left|\mathcal{S}_4^{\mathrm{II}}\right|}\right].
$$

*where $\mathcal{S}_4^{\mathrm{I}}$ and $\mathcal{S}_4^{\mathrm{II}}$ are subsets of the support set of $\mathcal{S}_4$. The set $\mathcal{S}_4^{\mathrm{I}}$ include all indices $i \in \mathcal{S}_4^{\mathrm{I}}$ which satisfy $\sigma_i\left(\boldsymbol{A}_{(k)}^{\star}\right) \geq \nu$, and $\mathcal{S}_4^{\mathrm{II}}$ includes all indices with $\sigma_i\left(\boldsymbol{A}_{(k)}^{\star}\right) < \nu$.*

*Proof.* Since $\|\cdot\|_{\mathrm{nuc}}$ is convex, we have

$$
\lambda\left\|\widehat{\boldsymbol{A}}_{(k)}\right\|_{\mathrm{nuc}} \geq \lambda\left\|\boldsymbol{A}_{(k)}^{\star}\right\|_{\mathrm{nuc}} + \lambda\left\langle\left[\widehat{\boldsymbol{\mathcal{A}}}-\boldsymbol{\mathcal{A}}^{\star}\right]_{(k)},\boldsymbol{G}^{\star}\right\rangle, \tag{30}
$$

$$
\lambda\left\|\boldsymbol{A}_{(k)}^{\star}\right\|_{\mathrm{nuc}} \geq \lambda\left\|\widehat{\boldsymbol{A}}_{(k)}\right\|_{\mathrm{nuc}} + \lambda\left\langle\left[\boldsymbol{\mathcal{A}}^{\star}-\widehat{\boldsymbol{\mathcal{A}}}\right]_{(k)},\widehat{\boldsymbol{G}}\right\rangle. \tag{31}
$$

where $\boldsymbol{G}^{\star} \in \partial\|\boldsymbol{A}_{(k)}^{\star}\|_{\mathrm{nuc}}$ and $\widehat{\boldsymbol{G}} \in \partial\|\widehat{\boldsymbol{A}}_{(k)}\|_{\mathrm{nuc}}$. From (30) and (31), we have

$$
\lambda\left\|\widehat{\boldsymbol{A}}_{(k)}\right\|_{\mathrm{nuc}} + \lambda\left\|\boldsymbol{A}_{(k)}^{\star}\right\|_{\mathrm{nuc}} \geq \lambda\left\|\boldsymbol{A}_{(k)}^{\star}\right\|_{\mathrm{nuc}} + \lambda\left\|\widehat{\boldsymbol{A}}_{(k)}\right\|_{\mathrm{nuc}} + \lambda\left\langle\left[\widehat{\boldsymbol{\mathcal{A}}}-\boldsymbol{\mathcal{A}}^{\star}\right]_{(k)},\boldsymbol{G}^{\star}\right\rangle + \lambda\left\langle\left[\boldsymbol{\mathcal{A}}^{\star}-\widehat{\boldsymbol{\mathcal{A}}}\right]_{(k)},\widehat{\boldsymbol{G}}\right\rangle.
$$

This equals to

$$
0 \geq \left(\lambda\left\langle\left[\widehat{\boldsymbol{\mathcal{A}}}-\boldsymbol{\mathcal{A}}^{\star}\right]_{(k)},\boldsymbol{G}^{\star}\right\rangle + \lambda\left\langle\left[\boldsymbol{\mathcal{A}}^{\star}-\widehat{\boldsymbol{\mathcal{A}}}\right]_{(k)},\widehat{\boldsymbol{G}}\right\rangle\right). \tag{32}
$$

Moreover, according to Lemma 13, we have

$$
\widetilde{\mathcal{L}}\left(\widehat{\boldsymbol{\mathcal{A}}}\right) \geq \widetilde{\mathcal{L}}\left(\boldsymbol{\mathcal{A}}^{\star}\right) + \left\langle\nabla\mathcal{L}\left(\boldsymbol{\mathcal{A}}^{\star}\right),\widehat{\boldsymbol{\mathcal{A}}}-\boldsymbol{\mathcal{A}}^{\star}\right\rangle + \left\langle\nabla\mathcal{Q}_{\lambda}\left(\boldsymbol{A}_{(k)}^{\star}\right),\left[\widehat{\boldsymbol{\mathcal{A}}}-\boldsymbol{\mathcal{A}}^{\star}\right]_{(k)}\right\rangle + \frac{\mu-\zeta}{2}\left\|\widehat{\boldsymbol{\mathcal{A}}}-\boldsymbol{\mathcal{A}}^{\star}\right\|_{\mathrm{F}}^{2}, \tag{33}
$$

$$
\widetilde{\mathcal{L}}\left(\boldsymbol{\mathcal{A}}^{\star}\right) \geq \widetilde{\mathcal{L}}\left(\widehat{\boldsymbol{\mathcal{A}}}\right) + \left\langle\nabla\mathcal{L}\left(\widehat{\boldsymbol{\mathcal{A}}}\right),\boldsymbol{\mathcal{A}}^{\star}-\widehat{\boldsymbol{\mathcal{A}}}\right\rangle + \left\langle\nabla\mathcal{Q}_{\lambda}\left(\widehat{\boldsymbol{A}}_{(k)}\right),\left[\boldsymbol{\mathcal{A}}^{\star}-\widehat{\boldsymbol{\mathcal{A}}}\right]_{(k)}\right\rangle + \frac{\mu-\zeta}{2}\left\|\boldsymbol{\mathcal{A}}^{\star}-\widehat{\boldsymbol{\mathcal{A}}}\right\|_{\mathrm{F}}^{2}. \tag{34}
$$

Summing (32), (33), and (34), we have

$$
\begin{aligned}
0 \geq &\left\langle\nabla\mathcal{L}\left(\boldsymbol{\mathcal{A}}^{\star}\right),\widehat{\boldsymbol{\mathcal{A}}}-\boldsymbol{\mathcal{A}}^{\star}\right\rangle + \left\langle\nabla\mathcal{L}\left(\widehat{\boldsymbol{\mathcal{A}}}\right),\boldsymbol{\mathcal{A}}^{\star}-\widehat{\boldsymbol{\mathcal{A}}}\right\rangle + (\mu-\zeta)\left\|\widehat{\boldsymbol{\mathcal{A}}}-\boldsymbol{\mathcal{A}}^{\star}\right\|_{\mathrm{F}}^{2} \\
&+ \left(\left\langle\nabla\mathcal{Q}_{\lambda}\left(\boldsymbol{A}_{(k)}^{\star}\right) + \lambda\boldsymbol{G}^{\star},\left[\widehat{\boldsymbol{\mathcal{A}}}-\boldsymbol{\mathcal{A}}^{\star}\right]_{(k)}\right\rangle + \left\langle\nabla\mathcal{Q}_{\lambda}\left(\widehat{\boldsymbol{A}}_{(k)}\right) + \lambda\widehat{\boldsymbol{G}},\boldsymbol{A}_{(k)}^{\star}-\widehat{\boldsymbol{A}}_{(k)}\right\rangle\right).
\end{aligned}
$$

Since $\widehat{\mathcal{A}}$ is the solution to the estimation problem and $\widehat{\mathcal{A}}$ satisfies the optimality condition, for any $\mathcal{A}' \in \mathbb{R}^{d_1 \times \cdots \times d_N}$, it holds that

$$\max_{\mathcal{A}'} \left\{ \left\langle \nabla \mathcal{L}\left(\widehat{\mathcal{A}}\right), \widehat{\mathcal{A}} - \mathcal{A}' \right\rangle + \left\langle \nabla \mathcal{Q}_\lambda\left(\widehat{\mathbf{A}}_{(k)}\right) + \lambda \widehat{\mathbf{G}}, \left[\widehat{\mathcal{A}} - \mathcal{A}'\right]_{(k)} \right\rangle \right\} \leq 0,$$

which implies

$$\left\langle \nabla \mathcal{L}\left(\widehat{\mathcal{A}}\right), \mathcal{A}^\star - \widehat{\mathcal{A}} \right\rangle + \left\langle \nabla \mathcal{Q}_\lambda\left(\widehat{\mathbf{A}}_{(k)}\right) + \lambda \widehat{\mathbf{G}}, \left[\mathcal{A}^\star - \widehat{\mathcal{A}}\right]_{(k)} \right\rangle \geq 0.$$

Since $\left\langle \nabla \mathcal{L}\left(\mathcal{A}^\star\right), \mathcal{A}^\star - \widehat{\mathcal{A}} \right\rangle = \left\langle \left[\nabla \mathcal{L}\left(\mathcal{A}^\star\right)\right]_{(k)}, \left[\mathcal{A}^\star - \widehat{\mathcal{A}}\right]_{(k)} \right\rangle$, we have

$$\begin{aligned}
(\mu - \zeta) \|\widehat{\mathcal{A}} - \mathcal{A}^\star\|_{\mathrm{F}}^2 &\leq \left[ \left\langle \left[\nabla \mathcal{L}\left(\mathcal{A}^\star\right)\right]_{(k)}, \left[\mathcal{A}^\star - \widehat{\mathcal{A}}\right]_{(k)} \right\rangle + \left\langle \nabla \mathcal{Q}_\lambda\left(\mathbf{A}^\star_{(k)}\right) + \lambda \mathbf{G}^\star, \left[\mathcal{A}^\star - \widehat{\mathcal{A}}\right]_{(k)} \right\rangle \right] \\
&\leq \left\langle \Pi_{\mathcal{F}^\perp} \left[ \left[\nabla \mathcal{L}\left(\mathcal{A}^\star\right)\right]_{(k)} + \nabla \mathcal{Q}_\lambda\left(\mathbf{A}^\star_{(k)}\right) + \lambda \mathbf{G}^\star \right], \left[\mathcal{A}^\star - \widehat{\mathcal{A}}\right]_{(k)} \right\rangle \\
&\quad + \left\langle \Pi_{\mathcal{F}} \left( \left[\nabla \mathcal{L}\left(\mathcal{A}^\star\right)\right]_{(k)} + \nabla \mathcal{Q}_\lambda\left(\mathbf{A}^\star_{(k)}\right) + \lambda \mathbf{G}^\star \right), \left[\mathcal{A}^\star - \widehat{\mathcal{A}}\right]_{(k)} \right\rangle. \quad (35)
\end{aligned}$$

We have defined $\sigma_i\left(\mathbf{A}^\star_{(k)}\right)$ as the $i$-th singular value of matrix $\mathbf{A}^\star_{(k)}$. With regard to the magnitudes of the singular values of $\mathbf{A}^\star_{(k)}$, we can decompose (35) into three parts:

- $i \in \mathcal{S}_4^{\mathrm{I}}$ that $\sigma_i\left(\mathbf{A}^\star_{(k)}\right) \geq \nu$,

- $i \in \mathcal{S}_4^{\mathrm{II}}$ that $\nu \geq \sigma_i\left(\mathbf{A}^\star_{(k)}\right) > 0$,

- $i \in \mathcal{S}_4^{\mathrm{c}}$ that $\sigma_i\left(\mathbf{A}^\star_{(k)}\right) = 0$.

(i) For $i \in \mathcal{S}_4^{\mathrm{I}}$ that $\sigma_i\left(\mathbf{A}^\star_{(k)}\right) \geq \nu$, define a subspace of $\mathcal{F}$ associated with $\mathcal{S}_4^{\mathrm{I}}$ as follows

$$\mathcal{F}_{\mathcal{S}_4^{\mathrm{I}}}\left(\mathbf{U}^\star, \mathbf{V}^\star\right) := \left\{ \mathbf{W} \,\middle|\, \mathrm{row}\left(\mathbf{W}\right) \subset \mathbf{V}_{\mathrm{I}}^\star, \mathrm{col}\left(\mathbf{W}\right) \subset \mathbf{U}_{\mathrm{I}}^\star \right\},$$

where $\mathbf{V}_{\mathrm{I}}^\star$ and $\mathbf{U}_{\mathrm{I}}^\star$ is the matrix with the $i$-th row of $\mathbf{V}_{\mathrm{I}}^\star$ and $\mathbf{U}_{\mathrm{I}}^\star$ with $i \in \mathcal{S}_4^{\mathrm{I}}$.

Recall that $\mathcal{R}_\lambda\left(\mathbf{A}^\star_{(k)}\right) = \lambda \left\|\mathbf{A}^\star_{(k)}\right\|_{\mathrm{nuc}} + \mathcal{Q}_\lambda\left(\mathbf{A}^\star_{(k)}\right)$, we have

$$\nabla \mathcal{R}_\lambda\left(\mathbf{A}^\star_{(k)}\right) = \nabla \mathcal{Q}_\lambda\left(\mathbf{A}^\star_{(k)}\right) + \lambda_k \left(\mathbf{U}_{\mathrm{I}}^\star \mathbf{V}_{\mathrm{I}}^{\star\top} + \mathbf{Z}_{\mathrm{I}}^\star\right),$$

where $\mathbf{Z}_{\mathrm{I}}^\star = -\lambda^{-1} \Pi_{\mathcal{F}_{\mathcal{S}_4^{\mathrm{I}}}} \left( \left[\nabla \mathcal{L}\left(\mathcal{A}^\star\right)\right]_{(k)} \right)$. Since $\|\mathbf{Z}_{\mathrm{I}}^\star\| \leq 1$ and $\mathbf{Z}_{\mathrm{I}}^\star \in \mathcal{F}_{\mathcal{S}_4^{\mathrm{I}}}$, which satisfies the condition of $\mathbf{W}^\star$ to be sub-gradient of $\left\|\mathbf{A}^\star_{(k)}\right\|$. Projecting $\mathcal{R}_\lambda\left(\mathbf{A}^\star_{(k)}\right)$ into the subspace $\mathcal{F}_{\mathcal{S}_4^{\mathrm{I}}}$, we have

$$\begin{aligned}
\Pi_{\mathcal{F}_{\mathcal{S}_4^{\mathrm{I}}}} \left( \nabla \mathcal{R}_\lambda\left(\mathbf{A}^\star_{(k)}\right) \right) &= \Pi_{\mathcal{F}_{\mathcal{S}_4^{\mathrm{I}}}} \left( \nabla \mathcal{Q}_\lambda\left(\mathbf{A}^\star_{(k)}\right) + \lambda \mathbf{U}_{\mathrm{I}}^\star \mathbf{V}_{\mathrm{I}}^{\star\top} + \lambda \mathbf{Z}_{\mathrm{I}}^\star \right) \\
&= \mathbf{U}_{\mathrm{I}}^\star q_\lambda'\left(\mathbf{\Sigma}_{\mathrm{I}}^\star\right) \mathbf{V}_{\mathrm{I}}^{\star\top} + \lambda \mathbf{U}_{\mathrm{I}}^\star \mathbf{V}_{\mathrm{I}}^{\star\top} \\
&= \mathbf{U}_{\mathrm{I}}^\star \left[ q_\lambda'\left(\mathbf{\Sigma}_{\mathrm{I}}^\star\right) + \lambda \mathbf{I}_{\mathrm{I}} \right] \mathbf{V}_{\mathrm{I}}^{\star\top},
\end{aligned}$$

where $\mathbf{I}_{\mathrm{I}}$ is an identity matrix with the size $\min\{d_k, \Pi_{j \neq k} d_j\}$ and $(q_\lambda'\left(\mathbf{\Sigma}_{\mathrm{I}}^\star\right) + \lambda \mathbf{I}_{\mathrm{I}})$ is a diagonal matrix that for $i \notin \mathcal{S}_4^{\mathrm{I}}$, the $i$-th entry on the diagonal equals 0, i.e. $[q_\lambda'\left(\mathbf{\Sigma}_{\mathrm{I}}^\star\right) + \lambda \mathbf{I}_{\mathrm{I}}]_{ii} = 0$, and for all $i \in \mathcal{S}_4^{\mathrm{I}}$, we have

$$\left[q_\lambda'\left(\mathbf{\Sigma}_{\mathrm{I}}^\star\right) + \lambda \mathbf{I}_{\mathrm{I}}\right]_{ii} = q_\lambda'\left(\sigma_i\left(\mathbf{A}^\star_{(k)}\right)\right) + \lambda = p_\lambda'\left(\sigma_i\left(\mathbf{A}^\star_{(k)}\right)\right) = 0.$$

The last equality is derived from fact that $i \in \mathcal{S}_4^{\mathrm{I}}$ satisfies Assumption 1, $p_\lambda'(t) = 0$. Therefore, we have $q_\lambda'(\boldsymbol{\Sigma}_{\mathrm{I}}^\star) + \lambda \boldsymbol{I}_{\mathrm{I}} = 0$, which indicates that $\Pi_{\mathcal{F}_{\mathcal{S}_4^{\mathrm{I}}}}\left(\nabla \mathcal{R}_\lambda\left(\boldsymbol{A}_{(k)}^\star\right)\right) = 0$. For $\boldsymbol{G}^\star = \boldsymbol{U}_{\mathrm{I}}^\star \boldsymbol{V}_{\mathrm{I}}^{\star\top} + \boldsymbol{Z}_{\mathrm{I}}^\star \in \partial\left\|\boldsymbol{A}_{(k)}^\star\right\|_{\mathrm{nuc}}$, we have

$$
\begin{aligned}
&\left\langle \Pi_{\mathcal{F}_{\mathcal{S}_4^{\mathrm{I}}}}\left[[\nabla \mathcal{L}(\boldsymbol{\mathcal{A}}^\star)]_{(k)} + \lambda \boldsymbol{G}^\star + \nabla \mathcal{Q}_\lambda\left(\boldsymbol{A}_{(k)}^\star\right)\right], \left[\boldsymbol{\mathcal{A}}^\star - \widehat{\boldsymbol{\mathcal{A}}}\right]_{(k)}\right\rangle \\
&= \left\langle \Pi_{\mathcal{F}_{\mathcal{S}_4^{\mathrm{I}}}}\left[[\nabla \mathcal{L}(\boldsymbol{\mathcal{A}}^\star)]_{(k)} + \nabla \mathcal{R}_\lambda\left(\boldsymbol{A}_{(k)}^\star\right)\right], \left[\boldsymbol{\mathcal{A}}^\star - \widehat{\boldsymbol{\mathcal{A}}}\right]_{(k)}\right\rangle \\
&= \left\langle \Pi_{\mathcal{F}_{\mathcal{S}_4^{\mathrm{I}}}}\left([\nabla \mathcal{L}(\boldsymbol{\mathcal{A}}^\star)]_{(k)}\right), \Pi_{\mathcal{F}_{\mathcal{S}_4^{\mathrm{I}}}}\left(\left[\boldsymbol{\mathcal{A}}^\star - \widehat{\boldsymbol{\mathcal{A}}}\right]_{(k)}\right)\right\rangle \\
&\leq \left\|\Pi_{\mathcal{F}_{\mathcal{S}_4^{\mathrm{I}}}}\left([\nabla \mathcal{L}(\boldsymbol{\mathcal{A}}^\star)]_{(k)}\right)\right\|_{\mathrm{sp}} \cdot \left\|\Pi_{\mathcal{F}_{\mathcal{S}_4^{\mathrm{I}}}}\left(\left[\boldsymbol{\mathcal{A}}^\star - \widehat{\boldsymbol{\mathcal{A}}}\right]_{(k)}\right)\right\|_{\mathrm{nuc}},
\end{aligned}
$$

where the last inequality is derived from the Hölder inequality. For $\left\|\Pi_{\mathcal{F}_{\mathcal{S}_4^{\mathrm{I}}}}\left(\left[\boldsymbol{\mathcal{A}}^\star - \widehat{\boldsymbol{\mathcal{A}}}\right]_{(k)}\right)\right\|_{\mathrm{nuc}}$, from the properties of projection on to the subspace $\mathcal{F}_{\mathcal{S}_4^{\mathrm{I}}}$, we have

$$
\begin{aligned}
\left\|\Pi_{\mathcal{F}_{\mathcal{S}_4^{\mathrm{I}}}}\left(\left[\boldsymbol{\mathcal{A}}^\star - \widehat{\boldsymbol{\mathcal{A}}}\right]_{(k)}\right)\right\|_{\mathrm{nuc}} &\leq \sqrt{|\mathcal{S}_4^{\mathrm{I}}|}\left\|\Pi_{\mathcal{F}_{\mathcal{S}_4^{\mathrm{I}}}}\left(\left[\boldsymbol{\mathcal{A}}^\star - \widehat{\boldsymbol{\mathcal{A}}}\right]_{(k)}\right)\right\|_{\mathrm{F}} \\
&\leq \sqrt{|\mathcal{S}_4^{\mathrm{I}}|}\left\|\left[\boldsymbol{\mathcal{A}}^\star - \widehat{\boldsymbol{\mathcal{A}}}\right]_{(k)}\right\|_{\mathrm{F}} = \sqrt{|\mathcal{S}_4^{\mathrm{I}}|}\left\|\boldsymbol{\mathcal{A}}^\star - \widehat{\boldsymbol{\mathcal{A}}}\right\|_{\mathrm{F}}.
\end{aligned}
$$

We obtain the second inequality from that the rank of the matrix $\Pi_{\mathcal{F}_{\mathcal{S}_4^{\mathrm{I}}}}\left(\left[\boldsymbol{\mathcal{A}}^\star - \widehat{\boldsymbol{\mathcal{A}}}\right]_{(k)}\right) \leq |\mathcal{S}_4^{\mathrm{I}}|$. Thus, we have

$$
\begin{aligned}
&\left\langle \Pi_{\mathcal{F}_{\mathcal{S}_4^{\mathrm{I}}}}\left[[\nabla \mathcal{L}(\boldsymbol{\mathcal{A}}^\star)]_{(k)} + \nabla \mathcal{Q}_\lambda\left(\boldsymbol{A}_{(k)}^\star\right) + \lambda \boldsymbol{G}^\star\right], \left[\boldsymbol{\mathcal{A}}^\star - \widehat{\boldsymbol{\mathcal{A}}}\right]_{(k)}\right\rangle \\
&\leq \sqrt{|\mathcal{S}_4^{\mathrm{I}}|}\left\|\Pi_{\mathcal{F}_{\mathcal{S}_4^{\mathrm{I}}}}\left(\left[\boldsymbol{\mathcal{A}}^\star - \widehat{\boldsymbol{\mathcal{A}}}\right]_{(k)}\right)\right\|_{\mathrm{sp}} \cdot \left\|\boldsymbol{\mathcal{A}}^\star - \widehat{\boldsymbol{\mathcal{A}}}\right\|_{\mathrm{F}}.
\end{aligned} \tag{36}
$$

(ii) For $i \in \mathcal{S}_4^{\mathrm{II}}$, $\nu \geq \sigma_i\left(\boldsymbol{A}_{(k)}^\star\right) > 0$, define a subspace of $\mathcal{F}$ associated with $\mathcal{S}_4^{\mathrm{II}}$ as follows

$$
\mathcal{F}_{\mathcal{S}_4^{\mathrm{II}}}(\boldsymbol{U}^\star, \boldsymbol{V}^\star) := \left\{\boldsymbol{W} \,|\, \mathrm{row}(\boldsymbol{W}) \subset \boldsymbol{V}_{\mathrm{II}}^\star, \mathrm{col}(\boldsymbol{W}) \subset \boldsymbol{U}_{\mathrm{II}}^\star\right\}.
$$

where $\boldsymbol{V}_{\mathrm{II}}^\star$ and $\boldsymbol{U}_{\mathrm{II}}^\star$ is the matrix with the $i$-th row of $\boldsymbol{U}^\star$ and $\boldsymbol{V}^\star$ with $i \in \mathcal{S}_4^{\mathrm{II}}$. Obviously, for all $\boldsymbol{W}$, the following decomposition holds

$$
\Pi_{\mathcal{F}}(\boldsymbol{W}) = \Pi_{\mathcal{F}_{\mathcal{S}_4^{\mathrm{I}}}}(\boldsymbol{W}) + \Pi_{\mathcal{F}_{\mathcal{S}_4^{\mathrm{II}}}}(\boldsymbol{W}).
$$

In addition, since $\boldsymbol{U}^\star$, $\boldsymbol{V}^\star$ are unitary matrices, for subspace $\mathcal{F}_{\mathcal{S}_4^{\mathrm{I}}}$ and $\mathcal{F}_{\mathcal{S}_4^{\mathrm{II}}}$, we have the complementary subspace $\mathcal{F}_{\mathcal{S}_4^{\mathrm{I}}}^\perp$, $\mathcal{F}_{\mathcal{S}_4^{\mathrm{II}}}^\perp$, thus we have

$$
\mathcal{F}_{\mathcal{S}_4^{\mathrm{I}}} \subset \mathcal{F}_{\mathcal{S}_4^{\mathrm{I}}}^\perp, \text{ and } \mathcal{F}_{\mathcal{S}_4^{\mathrm{II}}} \subset \mathcal{F}_{\mathcal{S}_4^{\mathrm{II}}}^\perp.
$$

Similar to analysis in (i) on $\mathcal{S}_4^{\mathrm{I}}$, we have

$$
\Pi_{\mathcal{F}_{\mathcal{S}_4^{\mathrm{II}}}}\left(\nabla \mathcal{Q}_\lambda\left(\boldsymbol{A}_{(k)}^\star\right)\right) = \boldsymbol{U}_{\mathrm{II}}^\star q_\lambda'(\boldsymbol{\Sigma}_{\mathrm{II}}^\star)\boldsymbol{V}_{\mathrm{II}}^{\star\top}.
$$

where $q_\lambda'(\boldsymbol{\Sigma}_{\mathrm{II}}^\star)$ is a diagonal matrix that $[q_\lambda'(\boldsymbol{\Sigma}_{\mathrm{II}}^\star)]_{ii} = 0$ for $i \notin \mathcal{S}_4^{\mathrm{II}}$, and for all $i \in \mathcal{S}_4^{\mathrm{II}}$,

$$
[q_\lambda'(\boldsymbol{\Sigma}_{\mathrm{II}}^\star)]_{ii} = \left[q_\lambda'\left(\sigma_i\left(\boldsymbol{A}_{(k)}^\star\right)\right)\right]_{ii} \leq \lambda.
$$

Since $\sigma_i \left( \boldsymbol{A}^{\star}_{(k)} \right) \leq \nu$, and $q_\lambda \left( \cdot \right)$ satisfies the regularity Assumption 1, $|q'_\lambda \left( t \right)| \leq \lambda$. Therefore

$$\left\| \Pi_{\mathcal{F}_{\mathcal{S}_4^{\mathrm{II}}}} \left( \nabla \mathcal{Q}_\lambda \left( \boldsymbol{A}^{\star}_{(k)} \right) \right) \right\|_{\mathrm{sp}} = \max_{i \in \mathcal{S}_4^{\mathrm{II}}} [q'_\lambda \left( \boldsymbol{\Sigma}^{\star}_{\mathrm{II}} \right)]_{ii} \leq \lambda.$$

Meanwhile, because of the fact that $\mathcal{F}_{\mathcal{S}_4^{\mathrm{II}}} \subset \mathcal{F}_{\mathcal{S}_4}$, we have

$$\left\| \Pi_{\mathcal{F}_{\mathcal{S}_4^{\mathrm{II}}}} \left( \lambda \boldsymbol{G}^{\star} \right) \right\|_{\mathrm{sp}} \leq \left\| \Pi_{\mathcal{F}} \left( \lambda \boldsymbol{U}^{\star}_{\mathrm{II}} \boldsymbol{V}^{\star\top}_{\mathrm{II}} \right) \right\|_{\mathrm{sp}}. \tag{37}$$

Since $\left\| \boldsymbol{U}^{\star} \boldsymbol{V}^{\star\top} \right\|_{\mathrm{sp}} = 1$, we have

$$\left\| \Pi_{\mathcal{F}} \left( \lambda \boldsymbol{U}^{\star}_{\mathrm{II}} \boldsymbol{V}^{\star\top}_{\mathrm{II}} \right) \right\|_{\mathrm{sp}} = \lambda. \tag{38}$$

Thus, from (37) and (38), we have

$$\left\| \Pi_{\mathcal{F}_{\mathcal{S}_4^{\mathrm{II}}}} \left( \lambda \boldsymbol{G}^{\star} \right) \right\|_{\mathrm{sp}} \leq \lambda. \tag{39}$$

Additionally, due to the fact that $\left\| \Pi_{\mathcal{F}_{\mathcal{S}_4^{\mathrm{II}}}} \left( [\nabla \mathcal{L} \left( \boldsymbol{\mathcal{A}}^{\star} \right)]_{(k)} \right) \right\|_{\mathrm{sp}} \leq \left\| [\nabla \mathcal{L} \left( \boldsymbol{\mathcal{A}}^{\star} \right)]_{(k)} \right\|_{\mathrm{sp}} \leq \lambda$, which indicates that

$$\left\langle \Pi_{\mathcal{F}_{\mathcal{S}_4^{\mathrm{II}}}} \left( [\nabla \mathcal{L} \left( \boldsymbol{\mathcal{A}}^{\star} \right)]_{(k)} + \nabla \mathcal{Q}_\lambda \left( \boldsymbol{A}^{\star}_{(k)} \right) + \lambda \boldsymbol{G}^{\star} \right), \left[ \boldsymbol{\mathcal{A}}^{\star} - \widehat{\boldsymbol{\mathcal{A}}} \right]_{(k)} \right\rangle$$
$$= \left\langle \Pi_{\mathcal{F}_{\mathcal{S}_4^{\mathrm{II}}}} \left( [\nabla \mathcal{L} \left( \boldsymbol{\mathcal{A}}^{\star} \right)]_{(k)} \right), \left[ \boldsymbol{\mathcal{A}}^{\star} - \widehat{\boldsymbol{\mathcal{A}}} \right]_{(k)} \right\rangle + \left\langle \Pi_{\mathcal{F}_{\mathcal{S}_4^{\mathrm{II}}}} \left( \nabla \mathcal{Q}_\lambda \left( \boldsymbol{A}^{\star}_{(k)} \right) \right), \left[ \boldsymbol{\mathcal{A}}^{\star} - \widehat{\boldsymbol{\mathcal{A}}} \right]_{(k)} \right\rangle + \left\langle \Pi_{\mathcal{F}_{\mathcal{S}_4^{\mathrm{II}}}} \left( \lambda \boldsymbol{G}^{\star} \right), \left[ \boldsymbol{\mathcal{A}}^{\star} - \widehat{\boldsymbol{\mathcal{A}}} \right]_{(k)} \right\rangle$$
$$\leq \left( \left\| \Pi_{\mathcal{F}_{\mathcal{S}_4^{\mathrm{II}}}} \left( [\nabla \mathcal{L} \left( \boldsymbol{\mathcal{A}}^{\star} \right)]_{(k)} \right) \right\|_{\mathrm{sp}} + \left\| \Pi_{\mathcal{F}_{\mathcal{S}_4^{\mathrm{II}}}} \left( \nabla \mathcal{Q}_\lambda \left( \boldsymbol{A}^{\star}_{(k)} \right) \right) \right\|_{\mathrm{sp}} + \left\| \Pi_{\mathcal{F}_{\mathcal{S}_4^{\mathrm{II}}}} \left( \lambda \boldsymbol{G}^{\star} \right) \right\|_{\mathrm{sp}} \right) \cdot \left\| \Pi_{\mathcal{F}_{\mathcal{S}_4^{\mathrm{II}}}} \left( \left[ \boldsymbol{\mathcal{A}}^{\star} - \widehat{\boldsymbol{\mathcal{A}}} \right]_{(k)} \right) \right\|_{\mathrm{nuc}},$$

where the last inequality is derived from the Hölder inequality. Since we have obtained the bound for each term, as in (38) and (39), we have

$$\left\langle \Pi_{\mathcal{F}_{\mathcal{S}_4^{\mathrm{II}}}} \left( [\nabla \mathcal{L} \left( \boldsymbol{\mathcal{A}}^{\star} \right)]_{(k)} + \nabla \mathcal{Q}_\lambda \left( \boldsymbol{A}^{\star}_{(k)} \right) + \lambda \boldsymbol{G}^{\star} \right), \left[ \boldsymbol{\mathcal{A}}^{\star} - \widehat{\boldsymbol{\mathcal{A}}} \right]_{(k)} \right\rangle \leq 3\lambda \left\| \Pi_{\mathcal{F}_{\mathcal{S}_4^{\mathrm{II}}}} \left( \left[ \boldsymbol{\mathcal{A}}^{\star} - \widehat{\boldsymbol{\mathcal{A}}} \right]_{(k)} \right) \right\|_{\mathrm{nuc}}$$
$$\leq 3\lambda \sqrt{|\mathcal{S}_4^{\mathrm{II}}|} \left\| \left[ \boldsymbol{\mathcal{A}}^{\star} - \widehat{\boldsymbol{\mathcal{A}}} \right]_{(k)} \right\|_{\mathrm{F}}$$
$$= 3\lambda \sqrt{|\mathcal{S}_4^{\mathrm{II}}|} \left\| \boldsymbol{\mathcal{A}}^{\star} - \widehat{\boldsymbol{\mathcal{A}}} \right\|_{\mathrm{F}}. \tag{40}$$

where the second inequality utilizes the fact that $\mathrm{rank} \left( \Pi_{\mathcal{F}_{\mathcal{S}_4^{\mathrm{II}}}} \left( \left[ \boldsymbol{\mathcal{A}}^{\star} - \widehat{\boldsymbol{\mathcal{A}}} \right]_{(k)} \right) \right) \leq |\mathcal{S}_4^{\mathrm{II}}|$.

(iii) For $i \in \mathcal{S}_4^c$, which correspond to the projector $\Pi_{\mathcal{F}^\perp}$ since $\sigma_i \left( \Pi_{\mathcal{F}^\perp} \left( \boldsymbol{A}^{\star}_{(k)} \right) \right) = 0$.

Based on Assumption 1, $q_\lambda \left( 0 \right) = q'_\lambda \left( 0 \right) = 0$. We have that $\nabla \mathcal{Q}_\lambda \left( \boldsymbol{A}^{\star}_{(k)} \right) = \boldsymbol{U}^{\star}_{\mathrm{c}} q'_\lambda \left( \boldsymbol{\Sigma}^{\star}_{\mathrm{c}} \right) \boldsymbol{V}^{\star\top}_{\mathrm{c}}$, where $\boldsymbol{\Sigma}^{\star}_{\mathrm{c}} \in \mathbb{R}^{r \times r}$ is a diagonal matrix and $r = \min\{d_k, \Pi_{j \neq k} d_j\}$. Now we have

$$\Pi_{\mathcal{F}^\perp} \left( \nabla \mathcal{Q}_\lambda \left( \boldsymbol{A}^{\star}_{(k)} \right) \right) = \left( \boldsymbol{I}_{\mathrm{c}} - \boldsymbol{U}^{\star}_{\mathrm{c}} \boldsymbol{U}^{\star\top}_{\mathrm{c}} \right) \boldsymbol{U}^{\star}_{k} q'_\lambda \left( \boldsymbol{\Sigma}^{\star}_{\mathrm{c}} \right) \boldsymbol{V}^{\star\top}_{\mathrm{c}} \left( \boldsymbol{I}_{\mathrm{c}} - \boldsymbol{V}^{\star}_{\mathrm{c}} \boldsymbol{V}^{\star\top}_{\mathrm{c}} \right)$$
$$= \left( \boldsymbol{U}^{\star}_{\mathrm{c}} - \boldsymbol{U}^{\star}_{\mathrm{c}} \right) q'_\lambda \left( \boldsymbol{\Sigma}^{\star}_{\mathrm{c}} \right) \left( \boldsymbol{V}^{\star\top}_{\mathrm{c}} - \boldsymbol{V}^{\star\top}_{\mathrm{c}} \right)$$
$$= 0,$$

where $\boldsymbol{I}_{\mathrm{c}}$ is the identity matrix. Meanwhile, since

$$\left\|\Pi_{\mathcal{F}^\perp}\left([\nabla\mathcal{L}\left(\boldsymbol{\mathcal{A}}^\star\right)]_{(k)}\right)\right\|_{\mathrm{sp}} \leq \left\|[\nabla\mathcal{L}\left(\boldsymbol{\mathcal{A}}^\star\right)]_{(k)}\right\|_{\mathrm{sp}} = \frac{\left\|[\mathfrak{X}^\star\left(\boldsymbol{\mathcal{E}}\right)]_{(k)}\right\|_{\mathrm{sp}}}{n} \leq \lambda.$$

For $\boldsymbol{Z}_{\mathrm{c}}^\star = -\lambda^{-1}\Pi_{\mathcal{F}^\perp}\left([\nabla\mathcal{L}\left(\boldsymbol{\mathcal{A}}^\star\right)]_{(k)}\right)$ and $\boldsymbol{G}^\star = \boldsymbol{U}_{\mathrm{c}}^\star\boldsymbol{V}_{\mathrm{c}}^{\star\top} + \boldsymbol{Z}_{\mathrm{c}}^\star \in \partial\left\|\boldsymbol{A}_{(k)}^\star\right\|_{\mathrm{nuc}}$, we have

$$\Pi_{\mathcal{F}^\perp}\left[[\nabla\mathcal{L}\left(\boldsymbol{\mathcal{A}}^\star\right)]_{(k)} + \lambda\boldsymbol{G}^\star\right] = \Pi_{\mathcal{F}^\perp}\left([\nabla\mathcal{L}\left(\boldsymbol{\mathcal{A}}^\star\right)]_{(k)}\right) + \lambda\boldsymbol{Z}_{\mathrm{c}}^\star = 0,$$

which implies that

$$\left\langle\Pi_{\mathcal{F}^\perp}\left([\nabla\mathcal{L}\left(\boldsymbol{\mathcal{A}}^\star\right)]_{(k)} + \lambda\boldsymbol{G}^\star + \nabla\mathcal{Q}_\lambda\left(\boldsymbol{A}_{(k)}^\star\right)\right), \left[\boldsymbol{\mathcal{A}}^\star - \widehat{\boldsymbol{\mathcal{A}}}\right]_{(k)}\right\rangle = \left\langle\boldsymbol{0}, \left[\boldsymbol{\mathcal{A}}^\star - \widehat{\boldsymbol{\mathcal{A}}}\right]_{(k)}\right\rangle = 0. \tag{41}$$

Adding (36), (40) and (41), which indicate that

$$\begin{aligned}
(\mu - \zeta)&\left\|\widehat{\boldsymbol{\mathcal{A}}} - \boldsymbol{\mathcal{A}}^\star\right\|_{\mathrm{F}} \\
&\leq \left\langle\Pi_{\mathcal{F}}\left([\nabla\mathcal{L}\left(\boldsymbol{\mathcal{A}}^\star\right)]_{(k)} + \nabla\mathcal{Q}_\lambda\left(\boldsymbol{A}_{(k)}^\star\right) + \lambda\boldsymbol{G}^\star\right), \left[\boldsymbol{\mathcal{A}}^\star - \widehat{\boldsymbol{\mathcal{A}}}\right]_{(k)}\right\rangle \\
&\leq \sqrt{|\mathcal{S}_4^{\mathrm{I}}|}\left\|\Pi_{\mathcal{S}_4^{\mathrm{I}}}\left([\nabla\mathcal{L}\left(\boldsymbol{\mathcal{A}}^\star\right)]_{(k)}\right)\right\|_{\mathrm{sp}} \cdot \left\|\boldsymbol{\mathcal{A}}^\star - \widehat{\boldsymbol{\mathcal{A}}}\right\|_{\mathrm{F}} + 3\lambda\sqrt{|\mathcal{S}_4^{\mathrm{II}}|}\left\|\boldsymbol{\mathcal{A}}^\star - \widehat{\boldsymbol{\mathcal{A}}}\right\|_{\mathrm{F}} \\
&= \left\|\boldsymbol{\mathcal{A}}^\star - \widehat{\boldsymbol{\mathcal{A}}}\right\|_{\mathrm{F}}\sqrt{|\mathcal{S}_4^{\mathrm{I}}|}\left\|\Pi_{\mathcal{S}_4^{\mathrm{I}}}\left([\nabla\mathcal{L}\left(\boldsymbol{\mathcal{A}}^\star\right)]_{(k)}\right)\right\|_{\mathrm{sp}} + 3\lambda\sqrt{|\mathcal{S}_4^{\mathrm{II}}|}.
\end{aligned}$$

Thus, we have

$$\left\|\widehat{\boldsymbol{\mathcal{A}}} - \boldsymbol{\mathcal{A}}^\star\right\|_{\mathrm{F}} \leq \frac{1}{\mu - \zeta}\left[\sqrt{|\mathcal{S}_4^{\mathrm{I}}|}\left\|\Pi_{\mathcal{S}_4^{\mathrm{I}}}\left([\nabla\mathcal{L}\left(\boldsymbol{\mathcal{A}}^\star\right)]_{(k)}\right)\right\|_{\mathrm{sp}} + 3\lambda\sqrt{|\mathcal{S}_4^{\mathrm{II}}|}\right].$$

$\square$

**Lemma 16.** *Suppose* $\boldsymbol{\mathcal{A}}^\star \in \mathbb{R}^{d_1 \times \cdots \times d_N}$ *with rank of each mode-(k) unfolding* $|\mathcal{S}_4|$. *Then the error bound between the oracle estimator* $\widehat{\boldsymbol{\mathcal{A}}}^O$ *and the true* $\boldsymbol{\mathcal{A}}^\star$ *satisfies*

$$\left\|\widehat{\boldsymbol{\mathcal{A}}}^O - \boldsymbol{\mathcal{A}}^\star\right\|_{\mathrm{F}} = \left\|\left[\widehat{\boldsymbol{\mathcal{A}}}^O - \boldsymbol{\mathcal{A}}^\star\right]_{(k)}\right\|_{\mathrm{F}} \leq \frac{2\sqrt{|\mathcal{S}_4|}\left\|\Pi_{\mathcal{F}}\left([\nabla\mathcal{L}\left(\boldsymbol{\mathcal{A}}^\star\right)]_{(k)}\right)\right\|_{\mathrm{sp}}}{\mu}. \tag{42}$$

*Proof.* Let $\boldsymbol{\mathcal{B}}' = \widehat{\boldsymbol{\mathcal{A}}}^O - \boldsymbol{\mathcal{A}}^\star$. According to the general estimator (2) and the definition of the adjoint operator $\mathfrak{X}(\cdot)$, we can express the difference in loss as follows

$$\begin{aligned}
\mathcal{L}\left(\widehat{\boldsymbol{\mathcal{A}}}^O\right) - \mathcal{L}(\boldsymbol{\mathcal{A}}^\star) &= \frac{1}{2n}\sum_{i=1}^n\left[\mathcal{Y}^{(i)} - \mathfrak{X}^{(i)}\left(\boldsymbol{\mathcal{A}}^\star + \boldsymbol{\mathcal{B}}'\right)\right]^2 - \frac{1}{2n}\sum_{i=1}^n\left[\mathcal{Y}^{(i)} - \mathfrak{X}^{(i)}\left(\boldsymbol{\mathcal{A}}^\star\right)\right]^2 \\
&= \frac{1}{2n}\sum_{i=1}^n\left[\boldsymbol{\mathcal{E}}^{(i)} - \mathfrak{X}^{(i)}\left(\boldsymbol{\mathcal{B}}'\right)\right]^2 - \frac{1}{2n}\sum_{i=1}^n\boldsymbol{\mathcal{E}}^{(i)} \\
&= \frac{1}{2n}\left\|[\mathfrak{X}\left(\boldsymbol{\mathcal{B}}'\right)]_{(k)}\right\|_{\mathrm{sp}}^2 - \frac{1}{n}\left\langle\mathfrak{X}^\star\left(\boldsymbol{\mathcal{E}}\right), \boldsymbol{\mathcal{B}}'\right\rangle.
\end{aligned}$$

Given that $\widehat{\boldsymbol{\mathcal{A}}}^O$ minimizes $\mathcal{L}(\cdot)$ over the subspace $\mathcal{F}$ and $\boldsymbol{A}_{(k)}^\star \in \mathcal{F}$, we have

$$\mathcal{L}\left(\widehat{\boldsymbol{\mathcal{A}}}^O\right) - \mathcal{L}(\boldsymbol{\mathcal{A}}^\star) \leq 0.$$

Thus, it follows that

$$\frac{1}{2n}\left\|\left[\mathfrak{X}\left(\mathcal{B}'\right)\right]_{(k)}\right\|_{\mathrm{sp}}^{2} \leq \frac{1}{n}\left\langle\mathfrak{X}^{\star}\left(\mathcal{E}\right),\mathcal{B}'\right\rangle. \tag{43}$$

By the RSC condition 2, we know that

$$\mathcal{L}\left(\mathcal{A}+\mathcal{B}\right)-\mathcal{L}\left(\mathcal{A}\right) \geq \left\langle\nabla\mathcal{L}\left(\mathcal{A}\right),\mathcal{B}\right\rangle+\frac{\mu}{2}\left\|\mathcal{B}\right\|_{\mathrm{F}}^{2}.$$

Applying this to $\mathcal{B}'$,

$$\begin{aligned}\frac{\mu}{2}\left\|\mathcal{B}'\right\|_{\mathrm{F}}^{2} &\leq \mathcal{L}\left(\mathcal{B}'\right)-\mathcal{L}\left(\mathcal{A}^{\star}\right)-\left\langle\nabla\mathcal{L}\left(\mathcal{A}^{\star}\right),\mathcal{B}'\right\rangle\\ &=\frac{1}{2n}\left\|\left[\mathfrak{X}\left(\mathcal{B}'\right)\right]_{(k)}\right\|_{\mathrm{sp}}^{2}-\frac{1}{n}\left\langle\mathfrak{X}^{\star}\left(\mathcal{E}\right),\mathcal{B}'\right\rangle-\left\langle\nabla\mathcal{L}\left(\mathcal{A}^{\star}\right),\mathcal{B}'\right\rangle.\end{aligned} \tag{44}$$

Substituting (43) into (44) gives

$$\frac{\mu}{2}\left\|\mathcal{B}'\right\|_{\mathrm{F}}^{2} \leq \frac{1}{2n}\left\|\left[\mathfrak{X}\left(\mathcal{B}'\right)\right]_{(k)}\right\|_{\mathrm{sp}}^{2} \leq \frac{1}{n}\left\langle\mathfrak{X}^{\star}\left(\mathcal{E}\right),\mathcal{B}'\right\rangle.$$

Therefore, we have

$$\left\|\mathcal{B}'\right\|_{\mathrm{F}}^{2} \leq \frac{2\left\langle\Pi_{\mathcal{F}}\left(\left[\mathfrak{X}^{\star}\left(\mathcal{E}\right)\right]_{(k)}\right),\mathcal{B}'\right\rangle}{n\mu} \leq \frac{2\left\|\Pi_{\mathcal{F}}\left(\left[\mathfrak{X}^{\star}\left(\mathcal{E}\right)\right]_{(k)}\right)\right\|_{\mathrm{sp}}\cdot\left\|\mathcal{B}'\right\|_{\mathrm{nuc}}}{n\mu}.$$

Using the fact that $\mathrm{rank}\left(\mathcal{B}'\right)=|\mathcal{S}_{4}|$, we have

$$\left\|\mathbf{B}'_{(k)}\right\|_{\mathrm{nuc}} \leq \sqrt{|\mathcal{S}_{4}|}\left\|\mathbf{B}'_{(k)}\right\|_{\mathrm{F}}^{2}.$$

Thus, it follows that

$$\left\|\mathbf{B}'_{(k)}\right\|_{\mathrm{F}}^{2} \leq \frac{2\sqrt{|\mathcal{S}_{4}|}\left\|\Pi_{\mathcal{F}}\left(\left[\mathfrak{X}^{\star}\left(\mathcal{E}\right)\right]_{(k)}\right)\right\|_{\mathrm{sp}}\cdot\left\|\mathbf{B}'_{(k)}\right\|_{\mathrm{F}}^{2}}{n\mu}.$$

Recalling that $\nabla\mathcal{L}\left(\mathcal{A}^{\star}\right)=-\frac{\mathfrak{X}^{\star}\left(\mathcal{E}\right)}{n}$, we conclude

$$\left\|\mathbf{B}'_{(k)}\right\|_{\mathrm{F}} \leq \frac{2\sqrt{|\mathcal{S}_{4}|}\left\|\Pi_{\mathcal{F}}\left(\left[\mathfrak{X}^{\star}\left(\mathcal{E}\right)\right]_{(k)}\right)\right\|_{\mathrm{sp}}}{n\mu} = \frac{2\sqrt{|\mathcal{S}_{4}|}\left\|\Pi_{\mathcal{F}}\left(\left[\nabla\mathcal{L}\left(\mathcal{A}^{\star}\right)\right]_{(k)}\right)\right\|_{\mathrm{sp}}}{\mu}.$$

Thus, since $\left\|\mathbf{B}'_{(k)}\right\|_{\mathrm{F}}=\left\|\mathcal{B}'\right\|_{\mathrm{F}}$, we have the desired error bound

$$\left\|\widehat{\mathcal{A}}^{O}-\mathcal{A}^{\star}\right\|_{\mathrm{F}}=\left\|\mathcal{B}'\right\|_{\mathrm{F}} \leq \frac{2\sqrt{|\mathcal{S}_{4}|}\left\|\Pi_{\mathcal{F}}\left(\left[\nabla\mathcal{L}\left(\mathcal{A}^{\star}\right)\right]_{(k)}\right)\right\|_{\mathrm{sp}}}{\mu}.$$

$\square$

Then, we prove Theorem 6.

*Proof.* Suppose $\widehat{\boldsymbol{G}} \in \partial \left\| \left( \widehat{\boldsymbol{A}}_{(k)} \right) \right\|_{\mathrm{nuc}}$, since $\widehat{\boldsymbol{\mathcal{A}}}$ satisfies the optimality condition, for any $\boldsymbol{\mathcal{A}}' \in \mathbb{R}^{d_1 \times \cdots \times d_N}$, it holds that

$$\max_{\boldsymbol{\mathcal{A}}'} \left\{ \left\langle \nabla \mathcal{L} \left( \widehat{\boldsymbol{\mathcal{A}}} \right), \widehat{\boldsymbol{\mathcal{A}}} - \boldsymbol{\mathcal{A}}' \right\rangle + \left\langle \nabla \mathcal{Q}_\lambda \left( \widehat{\boldsymbol{A}}_{(k)} \right) + \lambda \widehat{\boldsymbol{G}}, \left[ \widehat{\boldsymbol{\mathcal{A}}} - \boldsymbol{\mathcal{A}}' \right]_{(k)} \right\rangle \right\} \leq 0. \tag{45}$$

In the following, we will show some $\widehat{\boldsymbol{G}}^O \in \partial \left\| \widehat{\boldsymbol{A}}_{(k)}^O \right\|_{\mathrm{nuc}}$ satisfy that

$$\max_{\boldsymbol{A}'} \left\{ \left\langle \left[ \nabla \mathcal{L} \left( \widehat{\boldsymbol{\mathcal{A}}}^O \right) \right]_{(k)} + \nabla \mathcal{Q}_\lambda \left( \widehat{\boldsymbol{A}}_{(k)}^O \right) + \lambda \widehat{\boldsymbol{G}}^O, \left[ \widehat{\boldsymbol{\mathcal{A}}}^O - \boldsymbol{A}' \right]_{(k)} \right\rangle \right\} \leq 0. \tag{46}$$

Recall that $\widetilde{\mathcal{L}} (\boldsymbol{\mathcal{A}}) = \mathcal{L} (\boldsymbol{\mathcal{A}}) + \mathcal{Q}_\lambda \left( \boldsymbol{A}_{(k)} \right)$. Projecting the components of the inner product of the LHS in (46) into two complementary spaces $\mathcal{F}$ and $\mathcal{F}^\perp$, we have the following decomposition

$$\left\langle \left[ \nabla \mathcal{L} \left( \widehat{\boldsymbol{\mathcal{A}}}^O \right) \right]_{(k)} + \nabla \mathcal{Q}_\lambda \left( \widehat{\boldsymbol{A}}_{(k)}^O \right) + \lambda \widehat{\boldsymbol{G}}^O, \left[ \widehat{\boldsymbol{\mathcal{A}}}^O - \boldsymbol{\mathcal{A}}' \right]_{(k)} \right\rangle$$

$$= \underbrace{\left\langle \left[ \nabla \mathcal{L} \left( \widehat{\boldsymbol{\mathcal{A}}}^O \right) \right]_{(k)} + \nabla \mathcal{Q}_\lambda \left( \widehat{\boldsymbol{A}}_{(k)}^O \right) + \lambda \widehat{\boldsymbol{G}}^O, \Pi_{\mathcal{F}} \left( \left[ \widehat{\boldsymbol{\mathcal{A}}}^O - \boldsymbol{\mathcal{A}}' \right]_{(k)} \right) \right\rangle}_{P_1}$$

$$+ \underbrace{\left\langle \left[ \nabla \mathcal{L} \left( \widehat{\boldsymbol{\mathcal{A}}}^O \right) \right]_{(k)} + \nabla \mathcal{Q}_\lambda \left( \widehat{\boldsymbol{A}}_{(k)}^O \right) + \lambda \widehat{\boldsymbol{G}}^O, \Pi_{\mathcal{F}^\perp} \left( \left[ \widehat{\boldsymbol{\mathcal{A}}}^O - \boldsymbol{\mathcal{A}}' \right]_{(k)} \right) \right\rangle}_{P_2}. \tag{47}$$

**For Term $P_1$.** By applying Weyl's inequality for singular values, we obtain

$$\max_l \left| \sigma_i \left( \boldsymbol{A}_{(k)}^\star \right) - \sigma_i \left( \widehat{\boldsymbol{A}}_{(k)}^O \right) \right| \leq \left\| \boldsymbol{A}_{(k)}^\star - \widehat{\boldsymbol{A}}_{(k)}^O \right\|_{\mathrm{sp}}.$$

Further, from the properties of the Frobenius norm, we have

$$\left\| \boldsymbol{\mathcal{A}}^\star - \widehat{\boldsymbol{\mathcal{A}}}^O \right\|_{\mathrm{F}} = \left\| \left[ \boldsymbol{\mathcal{A}}^\star - \widehat{\boldsymbol{\mathcal{A}}}^O \right]_{(k)} \right\|_{\mathrm{F}}.$$

From Lemma 16, the estimation error $\boldsymbol{\mathcal{A}}^\star - \widehat{\boldsymbol{\mathcal{A}}}^O$ yields

$$\max_l \left| \sigma_i \left( \boldsymbol{A}_{(k)}^\star \right) - \sigma_i \left( \widehat{\boldsymbol{A}}_{(k)}^O \right) \right| \leq \frac{2 \sqrt{|\mathcal{S}_4|} \left\| [\mathfrak{X}^\star (\boldsymbol{\mathcal{E}})]_{(k)} \right\|_{\mathrm{sp}}}{n \mu},$$

where $|\mathcal{S}_4|$ denotes the rank of the unfolded matrix $\boldsymbol{A}_{(k)}^\star$. Utilizing the weak condition of the singular values, we find

$$\min_{i \in \mathcal{S}_4} \left| \sigma_i \left( \boldsymbol{A}_{(k)}^\star \right) \right| \geq \nu + \frac{2 \sqrt{|\mathcal{S}_4|}}{n \mu} \left\| [\mathfrak{X}^\star (\boldsymbol{\mathcal{E}})]_{(k)} \right\|_{\mathrm{sp}}.$$

Applying the triangle inequality, we derive

$$\min_{i \in \mathcal{S}_4} \left| \sigma_i \left( \widehat{\boldsymbol{A}}_{(k)}^O \right) \right| = \min_{i \in \mathcal{S}_4} \left| \sigma_i \left( \widehat{\boldsymbol{A}}_{(k)}^O \right) - \sigma_i \left( \boldsymbol{A}_{(k)}^\star \right) + \sigma_i \left( \boldsymbol{A}_{(k)}^\star \right) \right|$$

$$\geq - \max_{i \in \mathcal{S}_4} \left| \sigma_i \left( \widehat{\boldsymbol{A}}_{(k)}^O \right) - \sigma_i \left( \boldsymbol{A}_{(k)}^\star \right) \right| + \min_{i \in \mathcal{S}_4} \left| \sigma_i \left( \boldsymbol{A}_{(k)}^\star \right) \right|$$

$$\geq - \frac{2 \sqrt{|\mathcal{S}_4|}}{n \mu} \left\| [\mathfrak{X}^\star (\boldsymbol{\mathcal{E}})]_{(k)} \right\|_{\mathrm{sp}} + \nu + \frac{2 \sqrt{|\mathcal{S}_4|}}{n \mu} \left\| [\mathfrak{X}^\star (\boldsymbol{\mathcal{E}})]_{(k)} \right\|_{\mathrm{sp}}$$

$$= \nu.$$

Considering the definition of oracle estimator , $\widehat{\boldsymbol{\mathcal{A}}}^{O} \in \mathcal{F}$, which implies the tensor rank of each mode-(k) unfolding rank $\left(\widehat{\boldsymbol{A}}_{(k)}^{O}\right) = |\mathcal{S}_4|$. And we have the singular value decomposition $\widehat{\boldsymbol{A}}_{(k)}^{O} = \boldsymbol{U}^{\star} \widehat{\boldsymbol{\Sigma}}^{O} \boldsymbol{V}^{\star\top}$. Since $\mathcal{R}_{\lambda}\left(\boldsymbol{A}_{(k)}\right) = \lambda \left\|\boldsymbol{A}_{(k)}\right\|_{\text{nuc}} + \mathcal{Q}_{\lambda}\left(\boldsymbol{A}_{(k)}\right)$, for $\widehat{\boldsymbol{Z}}^{O} \in \mathcal{F}^{\perp}$, $\left\|\widehat{\boldsymbol{Z}}^{O}\right\|_{\text{sp}} \leq 1$, and $\widehat{\boldsymbol{\Sigma}}_{\mathcal{S}_4}^{O} \in \mathbb{R}^{|\mathcal{S}_4| \times |\mathcal{S}_4|}$ is a diagonal matrix, where $\Pi_{\mathcal{F}}\left(q_{\lambda}'\left(\widehat{\boldsymbol{\Sigma}}^{O}\right)\right) = q_{\lambda}'\left(\widehat{\boldsymbol{\Sigma}}_{\mathcal{S}_4}^{O}\right)$. Based on the definition of $\nabla \mathcal{Q}_{\lambda}\left(\cdot\right)$ and $\partial \left\|\cdot\right\|_{\text{nuc}}$, we have

$$
\begin{aligned}
\Pi_{\mathcal{F}}\left(\nabla \mathcal{R}_{\lambda}\left(\widehat{\boldsymbol{A}}_{(k)}^{O}\right)\right) &= \Pi_{\mathcal{F}}\left(\mathcal{Q}_{\lambda}\left(\widehat{\boldsymbol{A}}_{(k)}^{O}\right)\right) + \lambda \partial \left\|\widehat{\boldsymbol{A}}_{(k)}^{O}\right\|_{\text{nuc}} \\
&= \Pi_{\mathcal{F}}\left(\boldsymbol{U}^{\star} q_{\lambda}'\left(\widehat{\boldsymbol{\Sigma}}^{O}\right) \boldsymbol{V}^{\star\top} + \lambda \boldsymbol{U}^{\star} \boldsymbol{V}^{\star\top} + \lambda \widehat{\boldsymbol{Z}}^{O}\right) \\
&= \boldsymbol{U}^{\star}\left(q_{\lambda}'\left(\widehat{\boldsymbol{\Sigma}}_{\mathcal{S}_4}^{O}\right) + \lambda \boldsymbol{I}_{\mathcal{S}_4}\right) \boldsymbol{V}^{\star\top},
\end{aligned}
\tag{48}
$$

where the second equality in (48) is to simply project each component into the subspace $\mathcal{F}$. $\boldsymbol{I}_{\mathcal{S}_4}$ is the identity matrix and $\boldsymbol{I}_{\mathcal{S}_4} \in \mathbb{R}^{|\mathcal{S}_4| \times |\mathcal{S}_4|}$. Since $p_{\lambda}(t) = q_{\lambda}(t) + \lambda |t|$, we have $p_{\lambda}'(t) = q_{\lambda}'(t) + \lambda t$ for all $t > 0$. Consider the diagonal matrix $q_{\lambda}'\left(\widehat{\boldsymbol{\Sigma}}_{\mathcal{S}_4}^{O}\right) + \lambda \boldsymbol{I}_{\mathcal{S}_4}$, we have the $i$-th $(i \in \mathcal{S}_4)$ entry on the diagonal that

$$
\left[q_{\lambda}'\left(\widehat{\boldsymbol{\Sigma}}_{\mathcal{S}_4}^{O}\right) + \lambda \boldsymbol{I}_{\mathcal{S}_4}\right]_{ii} = q_{\lambda}'\left(\sigma_i\left(\widehat{\boldsymbol{A}}_{(k)}^{O}\right)\right) + \lambda = p_{\lambda}'\left(\sigma_i\left(\widehat{\boldsymbol{A}}_{(k)}^{O}\right)\right).
$$

Since $p_{\lambda}(\cdot)$ satisfies the regularity condition (iii) in Assumption 1 that $p_{\lambda}'(t) = 0$ for all $t \geq \nu$, we have $p_{\lambda}'\left(\sigma_i\left(\widehat{\boldsymbol{A}}_{(k)}^{O}\right)\right) = 0$ for $i \in \mathcal{S}_4$, due to the fact that $\sigma_i\left(\widehat{\boldsymbol{A}}_{(k)}^{O}\right) \geq \nu > 0$.

Therefore, the diagonal matrix $q_{\lambda}'\left(\widehat{\boldsymbol{\Sigma}}_{\mathcal{S}_4}^{O}\right) + \lambda \boldsymbol{I}_{\mathcal{S}_4} = 0$, substituting which in to (48) yields

$$
\Pi_{\mathcal{F}}\left(\nabla \mathcal{R}_{\lambda}\left(\widehat{\boldsymbol{A}}_{(k)}^{O}\right)\right) = 0.
\tag{49}
$$

Since $\widehat{\boldsymbol{\mathcal{A}}}^{O}$ is the estimator over $\mathcal{F}$, we have the optimality condition that for any $\boldsymbol{\mathcal{A}}' \in \mathbb{R}^{d_1 \times \cdots \times d_N}$, it holds that

$$
\max_{\boldsymbol{\mathcal{A}}'}\left\langle \left[\nabla \mathcal{L}\left(\widehat{\boldsymbol{\mathcal{A}}}^{O}\right)\right]_{(k)}, \Pi_{\mathcal{F}}\left(\left[\widehat{\boldsymbol{\mathcal{A}}}^{O} - \boldsymbol{\mathcal{A}}'\right]_{(k)}\right)\right\rangle \leq 0.
\tag{50}
$$

Substitute (49) and (50) into $P_1$, for all $\widehat{\boldsymbol{G}}^{O} \in \partial \left\|\widehat{\boldsymbol{A}}_{(k)}^{O}\right\|_{\text{nuc}}$ we have

$$
\begin{aligned}
&\max_{\boldsymbol{\mathcal{A}}'}\left\langle \left[\nabla \mathcal{L}\left(\widehat{\boldsymbol{\mathcal{A}}}^{O}\right)\right]_{(k)} + \nabla \mathcal{Q}_{\lambda}\left(\widehat{\boldsymbol{A}}_{(k)}^{O}\right) + \lambda \widehat{\boldsymbol{G}}^{O}, \Pi_{\mathcal{F}}\left(\left[\widehat{\boldsymbol{\mathcal{A}}}^{O} - \boldsymbol{\mathcal{A}}'\right]_{(k)}\right)\right\rangle \\
=&\max_{\boldsymbol{\mathcal{A}}'}\left\langle \left[\nabla \mathcal{L}\left(\widehat{\boldsymbol{\mathcal{A}}}^{O}\right)\right]_{(k)}, \Pi_{\mathcal{F}}\left(\left[\widehat{\boldsymbol{\mathcal{A}}}^{O} - \boldsymbol{\mathcal{A}}'\right]_{(k)}\right)\right\rangle + \max_{\boldsymbol{\mathcal{A}}'}\left\langle \Pi_{\mathcal{F}}\left(\nabla \mathcal{R}_{\lambda}\left(\widehat{\boldsymbol{A}}_{(k)}^{O}\right)\right), \Pi_{\mathcal{F}}\left(\left[\widehat{\boldsymbol{\mathcal{A}}}^{O} - \boldsymbol{\mathcal{A}}'\right]_{(k)}\right)\right\rangle \\
\leq&0.
\end{aligned}
\tag{51}
$$

**For Term $P_2$.** By definition of $\nabla \mathcal{Q}_{\lambda}\left(\cdot\right)$, and the regularity condition (v) in Assumption 1, we do the decomposition that $\nabla \mathcal{Q}_{\lambda}\left(\widehat{\boldsymbol{A}}_{(k)}^{O}\right) = \boldsymbol{U}^{\star} q_{\lambda}'\left(\widehat{\boldsymbol{\Sigma}}^{O}\right) \boldsymbol{V}^{\star\top}$, where $\widehat{\boldsymbol{\Sigma}}^{O}$ is diagonal matrix. Projecting $\nabla \mathcal{Q}_{\lambda}\left(\widehat{\boldsymbol{A}}_{(k)}^{O}\right)$ into $\mathcal{F}^{\perp}$ yields that

$$
\begin{aligned}
\Pi_{\mathcal{F}^{\perp}}\left(\nabla \mathcal{Q}_{\lambda}\left(\widehat{\boldsymbol{A}}_{(k)}^{O}\right)\right) &= \left(\boldsymbol{I}_{\mathcal{S}_4} - \boldsymbol{U}^{\star} \boldsymbol{U}^{\star\top}\right) \boldsymbol{U}^{\star} q_{\lambda}'\left(\widehat{\boldsymbol{\Sigma}}^{O}\right) \boldsymbol{V}^{\star\top}\left(\boldsymbol{I}_{\mathcal{S}_4} - \boldsymbol{V}^{\star} \boldsymbol{V}^{\star\top}\right) \\
&= \left(\boldsymbol{U}^{\star} - \boldsymbol{U}^{\star}\right) q_{\lambda}'\left(\widehat{\boldsymbol{\Sigma}}_{\mathcal{S}_4}^{O}\right)\left(\boldsymbol{V}^{\star\top} - \boldsymbol{V}^{\star\top}\right) \\
&= 0.
\end{aligned}
$$

Therefore,

$$P_2 = \left\langle \Pi_{\mathcal{F}^\perp} \left( \left[ \nabla \mathcal{L} \left( \widehat{\boldsymbol{\mathcal{A}}}^O \right) \right]_{(k)} + \lambda \widehat{\boldsymbol{G}}^O \right), \Pi_{\mathcal{F}^\perp} \left( \left[ \widehat{\boldsymbol{\mathcal{A}}}^O - \boldsymbol{\mathcal{A}}' \right]_{(k)} \right) \right\rangle.$$

Moreover, with the triangle inequality, we have

$$\left\| \left[ \nabla \mathcal{L} \left( \widehat{\boldsymbol{\mathcal{A}}}^O \right) \right]_{(k)} \right\|_{\mathrm{sp}} \leq \left\| \left[ \nabla \mathcal{L} \left( \boldsymbol{\mathcal{A}}^\star \right) \right]_{(k)} \right\|_{\mathrm{sp}} + \left\| \left[ \nabla \mathcal{L} \left( \boldsymbol{\mathcal{A}}^\star \right) \right]_{(k)} - \left[ \nabla \mathcal{L} \left( \widehat{\boldsymbol{\mathcal{A}}}^O \right) \right]_{(k)} \right\|_{\mathrm{sp}}$$

$$\leq \left\| \left[ \nabla \mathcal{L} \left( \boldsymbol{\mathcal{A}}^\star \right) \right]_{(k)} \right\|_{\mathrm{sp}} + \left\| \left[ \nabla \mathcal{L} \left( \boldsymbol{\mathcal{A}}^\star \right) \right]_{(k)} - \left[ \nabla \mathcal{L} \left( \widehat{\boldsymbol{\mathcal{A}}}^O \right) \right]_{(k)} \right\|_{\mathrm{F}}, \tag{52}$$

where the second inequality comes from the fact that

$$\left\| \left[ \nabla \mathcal{L} \left( \boldsymbol{\mathcal{A}}^\star \right) \right]_{(k)} - \left[ \nabla \mathcal{L} \left( \widehat{\boldsymbol{\mathcal{A}}}^O \right) \right]_{(k)} \right\|_{\mathrm{sp}} \leq \left\| \left[ \nabla \mathcal{L} \left( \boldsymbol{\mathcal{A}}^\star \right) \right]_{(k)} - \left[ \nabla \mathcal{L} \left( \widehat{\boldsymbol{\mathcal{A}}}^O \right) \right]_{(k)} \right\|_{\mathrm{F}}.$$

From Restricted Smoothness in Assumption 3 where $\left\| \nabla \mathcal{L} \left( \boldsymbol{\mathcal{A}} \right) - \nabla \mathcal{L} \left( \boldsymbol{\mathcal{A}} + \boldsymbol{\mathcal{B}}' \right) \right\|_{\mathrm{F}} \leq \left\| \boldsymbol{\mathcal{B}}' \right\|_{\mathrm{F}}$, we can substitute it into (52), and we have

$$\left\| \left[ \nabla \mathcal{L} \left( \widehat{\boldsymbol{\mathcal{A}}}^O \right) \right]_{(k)} \right\|_{\mathrm{sp}} \leq \left\| \left[ \nabla \mathcal{L} \left( \boldsymbol{\mathcal{A}}^\star \right) \right]_{(k)} \right\|_{\mathrm{sp}} + L \left\| \boldsymbol{\mathcal{A}}^\star - \widehat{\boldsymbol{\mathcal{A}}}^O \right\|_{\mathrm{F}}. \tag{53}$$

Since $\Pi_{\mathcal{F}^\perp} \left( \boldsymbol{\mathcal{B}}' \right) = 0$, it is evident that $\boldsymbol{\mathcal{B}}' \in \mathcal{C}$. Substitute (42) from Lemma 16 into (53), from the choice of $\lambda$, we have

$$\left\| \Pi_{\mathcal{F}^\perp} \left( \left[ \nabla \mathcal{L} \left( \widehat{\boldsymbol{\mathcal{A}}}^O \right) \right]_{(k)} \right) \right\|_{\mathrm{sp}} \leq \left\| \left[ \nabla \mathcal{L} \left( \widehat{\boldsymbol{\mathcal{A}}}^O \right) \right]_{(k)} \right\|_{\mathrm{sp}}$$

$$\leq \left\| \left[ \nabla \mathcal{L} \left( \boldsymbol{\mathcal{A}}^\star \right) \right]_{(k)} \right\|_{\mathrm{sp}} + \frac{2 \sqrt{|\mathcal{S}_4|} L}{n \mu} \left\| \left[ \mathfrak{X}^\star \left( \boldsymbol{\mathcal{E}} \right) \right]_{(k)} \right\|_{\mathrm{sp}}$$

$$\leq \lambda.$$

By setting $\widehat{\boldsymbol{Z}}^O = -\lambda^{-1} \Pi_{\mathcal{F}^\perp} \left( \left[ \nabla \mathcal{L} \left( \widehat{\boldsymbol{\mathcal{A}}}^O \right) \right]_{(k)} \right)$, such that $\widehat{\boldsymbol{G}}^O = \boldsymbol{U}^\star \boldsymbol{V}^{\star\top} + \widehat{\boldsymbol{Z}}^O \in \partial \left\| \widehat{\boldsymbol{A}}^O_{(k)} \right\|_{\mathrm{nuc}}$, since $\widehat{\boldsymbol{Z}}^O$ satisfies the condition $\widehat{\boldsymbol{Z}}^O \in \mathcal{F}^\perp$. $\left\| \widehat{\boldsymbol{Z}}^O \right\|_{\mathrm{sp}} \leq 1$, we have

$$\Pi_{\mathcal{F}^\perp} \left( \left[ \nabla \mathcal{L} \left( \widehat{\boldsymbol{\mathcal{A}}}^O \right) \right]_{(k)} + \lambda \widehat{\boldsymbol{G}}^O \right) = 0,$$

which implies that

$$P_2 = \left\langle \boldsymbol{0}, \Pi_{\mathcal{F}^\perp} \left( \left[ \boldsymbol{0} - \boldsymbol{\mathcal{A}}' \right]_{(k)} \right) \right\rangle = 0. \tag{54}$$

Substituting (51) and (54) into (47), we obtain (46) that

$$\max_{\boldsymbol{\mathcal{A}}'} \left\langle \left[ \nabla \mathcal{L} \left( \widehat{\boldsymbol{\mathcal{A}}}^O \right) \right]_{(k)} + \mathcal{Q}_\lambda \left( \widehat{\boldsymbol{A}}^O_{(k)} \right) + \lambda \widehat{\boldsymbol{G}}^O, \left[ \widehat{\boldsymbol{\mathcal{A}}}^O - \boldsymbol{\mathcal{A}}' \right]_{(k)} \right\rangle \leq 0.$$

Now we are going to prove that $\widehat{\boldsymbol{\mathcal{A}}}^O = \widehat{\boldsymbol{\mathcal{A}}}$ and the error bound between $\widehat{\boldsymbol{\mathcal{A}}}^O$ and $\boldsymbol{\mathcal{A}}^\star$.

Similar to the proof of Lemma 15, since $\| \cdot \|_{\mathrm{nuc}}$ is convex, and applying Lemma 13, we have

$$0 \geq \left\langle \nabla \mathcal{L} \left( \widehat{\boldsymbol{\mathcal{A}}} \right), \widehat{\boldsymbol{\mathcal{A}}}^O - \widehat{\boldsymbol{\mathcal{A}}} \right\rangle + \left\langle \nabla \mathcal{Q}_\lambda \left( \widehat{\boldsymbol{A}}_{(k)} \right) + \lambda \widehat{\boldsymbol{G}}, \left[ \widehat{\boldsymbol{\mathcal{A}}}^O - \widehat{\boldsymbol{\mathcal{A}}} \right]_{(k)} \right\rangle$$

$$+ \left\langle \nabla \mathcal{L} \left( \widehat{\boldsymbol{\mathcal{A}}}^O \right), \widehat{\boldsymbol{\mathcal{A}}} - \widehat{\boldsymbol{\mathcal{A}}}^O \right\rangle + \left\langle \nabla \mathcal{Q}_\lambda \left( \widehat{\boldsymbol{A}}^O_{(k)} \right) + \lambda \widehat{\boldsymbol{G}}^O, \left[ \widehat{\boldsymbol{\mathcal{A}}} - \widehat{\boldsymbol{\mathcal{A}}}^O \right]_{(k)} \right\rangle + \left( \mu - \varsigma \right) \left\| \widehat{\boldsymbol{\mathcal{A}}}^O - \widehat{\boldsymbol{\mathcal{A}}} \right\|_{\mathrm{F}}^2. \tag{55}$$

From (45), we have

$$
\left\langle \nabla \mathcal{L}\left(\widehat{\mathcal{A}}\right), \widehat{\mathcal{A}} - \widehat{\mathcal{A}}^{O}\right\rangle + \left\langle \nabla \mathcal{Q}_{\lambda}\left(\widehat{\mathbf{A}}_{(k)}\right) + \lambda\widehat{\mathbf{G}}, \left[\widehat{\mathcal{A}} - \widehat{\mathcal{A}}^{O}\right]_{(k)}\right\rangle
$$
$$
\leq \max_{\mathcal{A}'}\left\{\left\langle \nabla \mathcal{L}\left(\widehat{\mathcal{A}}\right), \widehat{\mathcal{A}} - \mathcal{A}'\right\rangle + \left\langle \nabla \mathcal{Q}_{\lambda}\left(\widehat{\mathbf{A}}_{(k)}\right) + \lambda\widehat{\mathbf{G}}, \left[\widehat{\mathcal{A}} - \mathcal{A}'\right]_{(k)}\right\rangle\right\} \leq 0. \tag{56}
$$

Therefore, in (55),

$$
\left\langle \nabla \mathcal{L}\left(\widehat{\mathcal{A}}\right), \widehat{\mathcal{A}}^{O} - \widehat{\mathcal{A}}\right\rangle + \left\langle \nabla \mathcal{Q}_{\lambda}\left(\widehat{\mathbf{A}}_{(k)}\right) + \lambda\widehat{\mathbf{G}}, \left[\widehat{\mathcal{A}}^{O} - \widehat{\mathcal{A}}\right]_{(k)}\right\rangle \geq 0.
$$

From (46), we have

$$
\left\langle \nabla \mathcal{L}\left(\widehat{\mathcal{A}}^{O}\right), \widehat{\mathcal{A}}^{O} - \widehat{\mathcal{A}}\right\rangle + \left\langle \nabla \mathcal{Q}_{\lambda}\left(\widehat{\mathbf{A}}_{(k)}^{O}\right) + \lambda\widehat{\mathbf{G}}, \left[\widehat{\mathcal{A}}^{O} - \widehat{\mathcal{A}}\right]_{(k)}\right\rangle
$$
$$
\leq \max_{\mathcal{A}'}\left\{\left\langle \nabla \mathcal{L}\left(\widehat{\mathcal{A}}^{O}\right), \widehat{\mathcal{A}}^{O} - \mathcal{A}'\right\rangle + \left\langle \nabla \mathcal{Q}_{\lambda}\left(\widehat{\mathbf{A}}_{(k)}^{O}\right) + \lambda\widehat{\mathbf{G}}, \left[\widehat{\mathcal{A}}^{O} - \mathcal{A}'\right]_{(k)}\right\rangle\right\} \leq 0. \tag{57}
$$

Therefore, in (55),

$$
\left\langle \nabla \mathcal{L}\left(\widehat{\mathcal{A}}^{O}\right), \widehat{\mathcal{A}} - \widehat{\mathcal{A}}^{O}\right\rangle + \left\langle \nabla \mathcal{Q}_{\lambda}\left(\widehat{\mathbf{A}}_{(k)}^{O}\right) + \lambda\widehat{\mathbf{G}}^{O}, \left[\widehat{\mathcal{A}} - \widehat{\mathcal{A}}^{O}\right]_{(k)}\right\rangle \geq 0.
$$

Substituting (55) and (56) into (57) such that

$$
(\mu - \zeta)\left\|\widehat{\mathcal{A}}^{O} - \widehat{\mathcal{A}}\right\|_{\mathrm{F}}^{2} \geq 0.
$$

Since $\mu > \zeta$, the inequation holds only if

$$
\widehat{\mathcal{A}}^{O} = \widehat{\mathcal{A}}.
$$

And by Lemma 16, we obtain the statistical oracle bound for the penalty

$$
\left\|\widehat{\mathcal{A}}^{O} - \mathcal{A}^{\star}\right\|_{\mathrm{F}} = \left\|\left[\widehat{\mathcal{A}} - \mathcal{A}^{\star}\right]_{(k)}\right\|_{\mathrm{F}} \leq \frac{2\sqrt{|\mathcal{S}_{4}|}\left\|\Pi_{\mathcal{F}}\left([\mathfrak{X}^{\star}(\boldsymbol{\mathcal{E}})]_{(k)}\right)\right\|_{\mathrm{sp}}}{n\mu}.
$$

Furthermore, we can have

$$
\left\|\widehat{\mathcal{A}}^{O} - \mathcal{A}^{\star}\right\|_{\mathrm{F}} = \frac{2\sqrt{|\mathcal{S}_{4}|}\tau_{k}}{n\mu},
$$

where $\tau_{k} = \left\|\Pi_{\mathcal{F}}\left([\mathfrak{X}^{\star}(\boldsymbol{\mathcal{E}})]_{(k)}\right)\right\|_{\mathrm{sp}}$, which completes the proof. $\qquad\square$

### B.5. Proof of Theorem 9

Recall that the proposed slice-wise low-rankness penalty can be reformulated as the sum of the $\ell_1$ penalty and a concave part. Specifically, we have:

$$
\mathcal{R}_{\lambda}\left(\boldsymbol{\mathcal{A}}\right) = \sum_{l=1}^{\Pi_{m \neq j,k} d_{m}} \sum_{s=1}^{s^{\mathrm{all}}} p_{\lambda}\left(\sigma_{s}\left([\boldsymbol{\mathcal{A}}_{(j,k)}]_{\cdot,\cdot,l}\right)\right) = \sum_{l=1}^{\Pi_{m \neq j,k} d_{m}}\left[\lambda\left\|[\boldsymbol{\mathcal{A}}_{(j,k)}]_{\cdot,\cdot,l}\right\|_{\mathrm{nuc}} + \mathcal{Q}_{\lambda}\left([\boldsymbol{\mathcal{A}}_{(j,k)}]_{\cdot,\cdot,l}\right)\right],
$$

where $s^{\mathrm{all}} = \min\left\{ d_j d_k, \prod_{l \neq j,k} d_l \right\}$, $\left[\boldsymbol{\mathcal{A}}_{(j,k)}\right]_{\cdot,\cdot,l}$ denotes the $l$-th slice of the mode-$(j,k)$ unfolding $\boldsymbol{\mathcal{A}}_{(j,k)}$ and $\sigma_s\left(\left[\boldsymbol{\mathcal{A}}_{(j,k)}\right]_{\cdot,\cdot,l}\right)$ denotes the $s$-th singular value of the slice. For the estimation problem, we define

$$\widetilde{\mathcal{L}}(\boldsymbol{\mathcal{A}}) = \mathcal{L}(\boldsymbol{\mathcal{A}}) + \sum_{l=1}^{\prod_{m \neq j,k} d_m} \mathcal{Q}_\lambda\left(\left[\boldsymbol{\mathcal{A}}_{(j,k)}\right]_{\cdot,\cdot,l}\right),$$

where $\mathcal{Q}_\lambda\left(\left[\boldsymbol{\mathcal{A}}_{(j,k)}\right]_{\cdot,\cdot,l}\right) = \sum_{s=1}^{s^{\mathrm{all}}} q_\lambda\left(\sigma_s\left(\left[\boldsymbol{\mathcal{A}}_{(j,k)}\right]_{\cdot,\cdot,l}\right)\right)$.

Based on Lemma 13, for slice-wise lowrankness regularizer, we can similarly prove the following lemmas

**Lemma 17.** *Under Assumption 2 , $\mu > \zeta$, and the regularization parameter $\lambda \geq \frac{\left\|\left[[\mathfrak{X}^\star(\boldsymbol{\mathcal{E}})]_{(j,k)}\right]_{\cdot,\cdot,l}\right\|_{\mathrm{sp}}}{2n}$, we have*

$$\left\|\Pi_{\mathcal{F}^\perp}\left(\left[\widehat{\boldsymbol{\mathcal{A}}}_{(j,k)}\right]_{\cdot,\cdot,l} - \left[\boldsymbol{\mathcal{A}}^\star_{(j,k)}\right]_{\cdot,\cdot,l}\right)\right\|_{\mathrm{nuc}} \leq 5 \left\|\Pi_{\mathcal{F}}\left(\left[\widehat{\boldsymbol{\mathcal{A}}}_{(j,k)}\right]_{\cdot,\cdot,l} - \left[\boldsymbol{\mathcal{A}}^\star_{(j,k)}\right]_{\cdot,\cdot,l}\right)\right\|_{\mathrm{nuc}}.$$

*Proof.* Similar to the proof of Lemma 14, we can prove the Lemma 17

$\square$

From Lemma 13 and Lemma 17, we can prove the following general deterministic bound.

**Lemma 18.** *For the estimated parameter tensor $\widehat{\boldsymbol{\mathcal{A}}}$ and the true parameter tensor $\boldsymbol{\mathcal{A}}^\star$, we have*

$$\left\|\widehat{\boldsymbol{\mathcal{A}}} - \boldsymbol{\mathcal{A}}^\star\right\|_{\mathrm{F}} \leq \frac{1}{(\mu - \zeta)}\sqrt{\sum_{l=1}^{\prod_{m \neq j,k} d_m}\left[\sqrt{|\mathcal{S}_5^{\mathrm{I}}|}\left\|\Pi_{\mathcal{F}}\left(\left[[\mathfrak{X}^\star(\boldsymbol{\mathcal{E}})]_{(j,k)}\right]_{\cdot,\cdot,l}\right)\right\|_{\mathrm{sp}} + 3\lambda\sqrt{|\mathcal{S}_5^{\mathrm{II}}|}\right]^2}.$$

*Proof.* Similar to the proof for Lemma 16, we can derive the error bound for the slice-wise low-rankness regularizer. $\square$

**Lemma 19.** *Suppose $\boldsymbol{\mathcal{A}}^\star \in \mathbb{R}^{d_1 \times \cdots \times d_N}$ with rank of each slices $|\mathcal{S}_5|$. Then the error bound between the oracle estimator $\widehat{\boldsymbol{\mathcal{A}}}^O$ and the true $\boldsymbol{\mathcal{A}}^\star$ satisfies*

$$\left\|\widehat{\boldsymbol{\mathcal{A}}}^O - \boldsymbol{\mathcal{A}}^\star\right\|_{\mathrm{F}} = \sqrt{\sum_{l=1}^{\prod_{m \neq j,k} d_m}\left\|\left[\left[\widehat{\boldsymbol{\mathcal{A}}}^O - \boldsymbol{\mathcal{A}}^\star\right]_{(j,k)}\right]_{\cdot,\cdot,l}\right\|_{\mathrm{F}}^2} \lesssim \frac{2\sqrt{|\mathcal{S}_5|}\left\|\Pi_{\mathcal{F}}\left(\left[[\mathfrak{X}^\star(\boldsymbol{\mathcal{E}})]_{(j,k)}\right]_{\cdot,\cdot,l}\right)\right\|_{\mathrm{sp}}}{n\mu}.$$

*Proof.* With Lemma 19, we can also obtain that $\widehat{\boldsymbol{\mathcal{A}}}^O = \widehat{\boldsymbol{\mathcal{A}}}$. Similarly, we can prove the Theorem 9. $\square$

## C. Complementary Experimental Results

In this section, we present additional results for the proposed penalties introduced in Section A. In Section C.1, we evaluate the performance of these penalties on synthetic data and provide a detailed analysis of the experimental findings. Furthermore, Section C.2 demonstrates the effectiveness of the penalties on real-world data.

### C.1. Synthetic Data

Figure 5 illustrates the impact of the fiber-wise sparsity regularizer on estimation accuracy. In these experiments, we consider 3rd-order tensor $\boldsymbol{\mathcal{A}} \in \mathbb{R}^{d \times d \times d}$, with $d = 16$. We display the results of the Mean Squared Frobenius norm Error (MSFE) when varying the tensor dimension $d$, the fiber-wise sparsity $|\mathcal{S}_3|$, and the sample size $n$, respectively. In Figures 5a and 5b, we fix the fiber-wise sparsity $|\mathcal{S}_3| = 4$. The three lines in different colors represent varying sample sizes $n = \{1000, 2000, 3000\}$. From Figure 5a, we observe that MSFE increases as the tensor dimension $d$ increases. From Figure 5b, we find that increasing the sample size $n$ decreases the MSFE. This demonstrates that larger sample sizes improve the accuracy of the tensor estimation, as expected. In Figure 5c, we see that increasing the fiber-wise-sparsity $|\mathcal{S}_3|$ leads

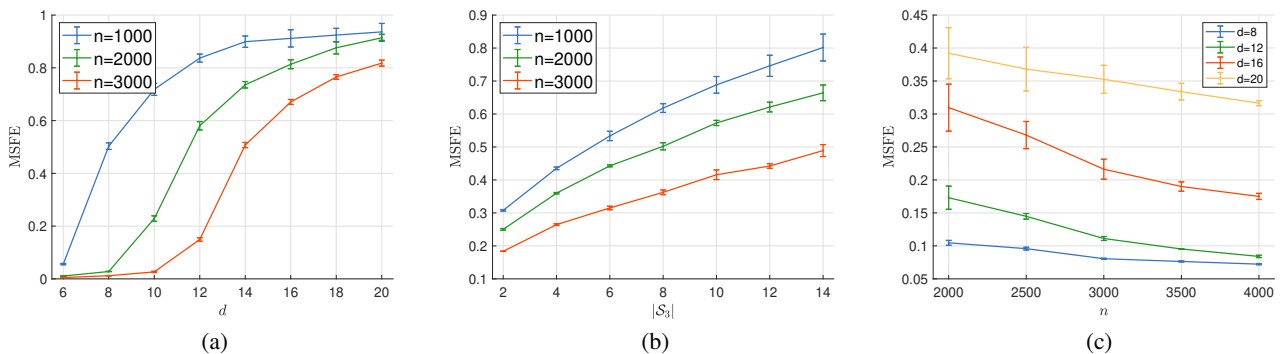

Figure 5: Fiber-wise sparsity regularizer with the error bars of MSFE $\pm$ standard deviation .

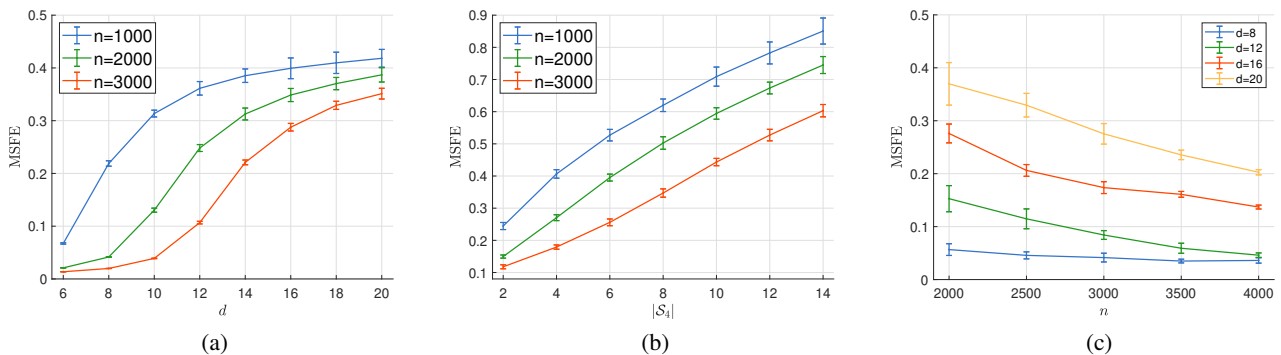

Figure 6: Slice-wise sparsity regularizer with the error bars of MSFE $\pm$ standard deviation .

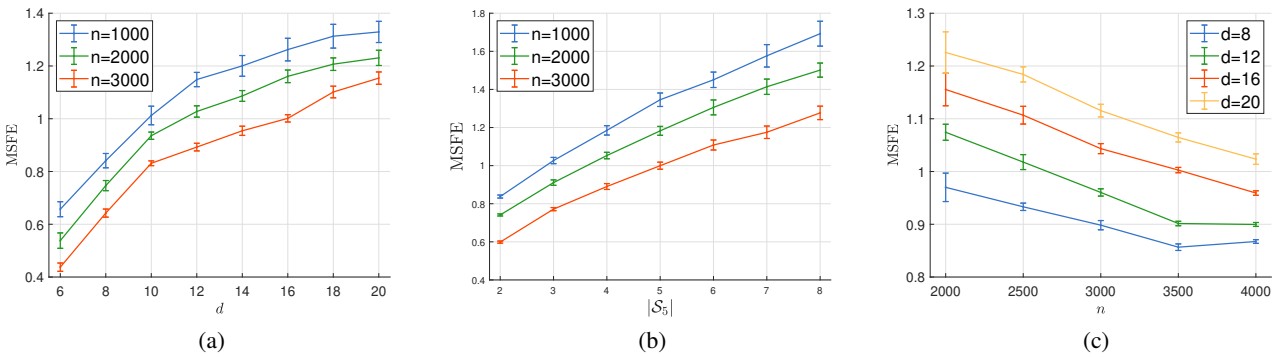

Figure 7: Slice-wise low-rankness penalty with the error bars of MSFE $\pm$ standard deviation .

Table 2: Comparisons between proposed nonconvex penalties and convex penalties.

| Structures | Methods | | Synthetic Data | | | | | Real-world Data | |
|---|---|---|---|---|---|---|---|---|---|
| | | | size | $|\mathcal{S}|$ | $\eta$ | MSFE | RMSE | MSFE | MPRE |
| Sparsity | Entry-wise | Nonconvex | $16 \times 16 \times 16$ | 2048 | 0.1 | $\mathbf{0.4042 \pm 0.0201}$ | $\mathbf{0.0992 \pm 0.0021}$ | $\mathbf{134.5864 \pm 11.2950}$ | $\mathbf{7.6072 \pm 0.0301}$ |
| | | Convex | | | | $0.6938 \pm 0.0297$ | $0.1004 \pm 0.0023$ | $144.7160 \pm 14.9947$ | $7.7498 \pm 0.0457$ |
| | Fiber-wise | Nonconvex | $16 \times 16 \times 16$ | 8 | 0.1 | $\mathbf{0.4406 \pm 0.0157}$ | $\mathbf{0.0993 \pm 0.0012}$ | $\mathbf{90.3068 \pm 7.3006}$ | $\mathbf{4.7161 \pm 0.0103}$ |
| | | Convex | | | | $0.7512 \pm 0.0439$ | $0.0995 \pm 0.0019$ | $102.7019 \pm 9.2188$ | $5.0330 \pm 0.0118$ |
| | Slice-wise | Nonconvex | $16 \times 16 \times 20$ | 8 | 0.1 | $\mathbf{0.5761 \pm 0.0289}$ | $\mathbf{0.0997 \pm 0.0027}$ | $\mathbf{43.8705 \pm 3.0257}$ | $\mathbf{1.8909 \pm 0.0043}$ |
| | | Convex | | | | $0.7201 \pm 0.0314$ | $0.1005 \pm 0.0039$ | $48.4585 \pm 3.8834$ | $1.9250 \pm 0.0045$ |
| Low-rankness | Mode-wise | Nonconvex | $16 \times 16 \times 16$ | 5 | 1 | $\mathbf{0.5482 \pm 0.0395}$ | $\mathbf{0.1002 \pm 0.0012}$ | $\mathbf{35.5536 \pm 1.4889}$ | $\mathbf{1.0330 \pm 0.0022}$ |
| | | Convex | | | | $1.7411 \pm 0.0953$ | $0.1096 \pm 0.0020$ | $41.2719 \pm 3.5079$ | $1.1027 \pm 0.0024$ |
| | Slice-wise | Nonconvex | $16 \times 16 \times 20$ | 5 | 1 | $\mathbf{0.9214 \pm 0.0736}$ | $\mathbf{0.1004 \pm 0.0010}$ | $\mathbf{8.9348 \pm 0.7493}$ | $\mathbf{0.0436 \pm 0.0002}$ |
| | | Convex | | | | $1.8261 \pm 0.1066$ | $0.1113 \pm 0.0031$ | $10.1655 \pm 0.9050$ | $0.6348 \pm 0.0009$ |

to an increase in the estimate error. Furthermore, the standard deviation of the estimation error follows the same trend, increasing with fiber-wise sparsity.

Figure 6 presents the results of the slice-wise sparsity regularizer. In Figures 6a and 6b, the number of slices is uniformly set to $s = 20$. And we set the slice-wise sparsity $|\mathcal{S}_4| = 4$ in Figures 6a and 6c. We select three sample sizes while varying the dimension $d$ or the number of non-zero slices $|\mathcal{S}_4|$. The results indicate that the estimation error increases with increments in $d$ or $|\mathcal{S}_4|$. The standard deviation of the MSFE also rises as the MSFE increases. Furthermore, as observed in Figure 6c, increasing the sample size reduces estimation errors when the dimension is fixed.

Figure 7 demonstrates the results of the slice-wise low-rankness penalty. In Figure 7b, the $x$-axis $|\mathcal{S}_5|$ represents the rank of each slice of the tensor $\mathcal{A} \in \mathbb{R}^{d \times d \times d}$. Figures 7a and 7c fix the rank of each slice to 5. The three distinct lines correspond to the estimation errors for sample size $n = \{1000, 2000, 3000\}$. From Figure 7a, we observe that with a fixed rank and sample size, the estimation error increases as the dimension $d$ enlarges. Furthermore, Figure 7b shows that the estimation errors increase with the rank. Figure 7c demonstrates that with more samples, the estimation errors decrease.

In Table 2, we compare the performance of our proposed nonconvex penalties against traditional convex penalties. For sparsity penalties, we set the $\eta = 0.1$, and for low-rankness penalties, we set $\eta = 1$. We configure the tensor dimension such that tensors with slices-wise structures $\mathcal{A} \in \mathbb{R}^{d \times d \times s}$ and the others $\mathcal{A} \in \mathbb{R}^{d \times d \times d}$, where $d = 16$, $s = 20$. Depending on the tensor structure, the sparsity or the rank of the tensors varies accordingly. The results in Table 2 demonstrate that nonconvex penalties achieve lower MSFE for parameter estimation and lower RMSE for predictions compared to their convex counterparts. These empirical findings are in strong agreement with our theoretical analysis.

Table 3: The Frobenius norm $\left\| \widehat{\mathcal{A}} - \mathcal{A}^{\star} \right\|_{\mathrm{F}}$ with standard variance changing the noise parameter

| Structures | Methods | $d = 10 \times 10 \times 10$ | | | $d = 20 \times 20 \times 20$ | | |
|---|---|---|---|---|---|---|---|
| | | $\eta = 0.1$ | $\eta = 1$ | $\eta = 5$ | $\eta = 0.1$ | $\eta = 1$ | $\eta = 5$ |
| Sparsity | Entry-wise | $1.8909 \pm 0.2004$ | $1.9021 \pm 0.2214$ | $1.9015 \pm 0.2084$ | $19.4215 \pm 2.2710$ | $18.6899 \pm 2.7556$ | $19.2904 \pm 2.3523$ |
| | Fiber-wise | $1.8622 \pm 0.2813$ | $1.8461 \pm 0.2783$ | $1.8500 \pm 0.2431$ | $19.8920 \pm 2.6721$ | $19.3542 \pm 2.8909$ | $19.7628 \pm 2.4745$ |
| | Slice-wise | $2.0509 \pm 0.3510$ | $2.0421 \pm 0.3242$ | $1.9927 \pm 0.3666$ | $20.0062 \pm 2.7153$ | $20.4267 \pm 2.7248$ | $19.4231 \pm 2.6420$ |
| Low-rankness | Mode-wise | $2.6311 \pm 0.3008$ | $2.6947 \pm 0.3254$ | $2.6265 \pm 0.3410$ | $24.6129 \pm 2.7010$ | $23.8932 \pm 2.7108$ | $24.6571 \pm 2.5114$ |
| | Slice-wise | $3.0150 \pm 0.3227$ | $3.0045 \pm 0.3754$ | $3.0184 \pm 0.3365$ | $25.8502 \pm 3.7601$ | $25.6691 \pm 3.5732$ | $25.9134 \pm 3.8282$ |

**Additional experiments.** In this paper, we derive five corollaries that establish error bounds involving the noise parameter $\eta$. The analysis of these bounds is nontrivial due to the interplay among conjugate operators, projection operators, and nuclear norm regularization. From equation (20), we observe that increasing $\eta$ enlarges the associated error term, thereby worsening the overall error bound. To illustrate this effect, Table 3 presents results from synthetic data experiments conducted under varying noise levels, which confirm the anticipated impact of $\eta$ on the error magnitude.

As outlined in the five corollaries of our paper, our theoretical framework is inherently generalizable to tensors of any order. Although the scope of this paper did not include experimental results for higher-order tensors, in Table 4, we have conducted supplementary experiments that demonstrate promising outcomes for these cases.

Also, to explore whether the proposed methods perform robustness under an unknown structure, in table 5, we implement

Table 4: The Frobenius norm $\left\|\widehat{\mathcal{A}} - \mathcal{A}^{\star}\right\|_{\mathrm{F}}$ with standard variance for higher dimension

| Structures | Methods | 3-order | | 4-order | | 5-order | |
|---|---|---|---|---|---|---|---|
| | | $d = 8$ | $d = 16$ | $d = 8$ | $d = 16$ | $d = 8$ | $d = 16$ |
| Sparsity | Entry-wise | $1.0509 \pm 0.1004$ | $1.9021 \pm 0.2214$ | $7.9015 \pm 1.2084$ | $19.4215 \pm 2.2710$ | $58.6899 \pm 7.7556$ | $192.2904 \pm 17.3523$ |
| | Fiber-wise | $1.0622 \pm 0.0813$ | $1.8461 \pm 0.2783$ | $7.8500 \pm 1.2431$ | $19.8920 \pm 2.6721$ | $59.3542 \pm 8.2909$ | $190.7628 \pm 16.4745$ |
| | Slice-wise | $1.0909 \pm 0.1510$ | $2.0421 \pm 0.3242$ | $8.0927 \pm 1.3666$ | $20.0062 \pm 2.7153$ | $60.4267 \pm 8.1248$ | $193.4231 \pm 17.6420$ |
| Low-rankness | Mode-wise | $1.6311 \pm 0.3008$ | $2.6947 \pm 0.3254$ | $8.6265 \pm 1.3410$ | $34.6129 \pm 3.4010$ | $63.8932 \pm 9.7108$ | $224.6571 \pm 21.5114$ |
| | Slice-wise | $1.8150 \pm 0.3227$ | $2.7045 \pm 0.3754$ | $9.0184 \pm 1.3365$ | $35.8502 \pm 3.7601$ | $65.6691 \pm 9.5732$ | $245.9134 \pm 22.8282$ |

Table 5: The Frobenius norm $\left\|\widehat{\mathcal{A}} - \mathcal{A}^{\star}\right\|_{\mathrm{F}}$ with standard variance changing ground data structure of our proposed methods

| Structures | Methods | Tensor Data Structures | | | | |
|---|---|---|---|---|---|---|
| | | entry-sp | fiber-sp | slice-sp | lr-mode | lr-slice |
| Sparsity | Entry-wise | $\mathbf{1.0509 \pm 0.1004}$ | $1.0680 \pm 0.1027$ | $1.1263 \pm 0.2046$ | $1.8991 \pm 0.4002$ | $1.9367 \pm 0.3979$ |
| | Fiber-wise | $1.0931 \pm 0.1421$ | $\mathbf{1.0622 \pm 0.0813}$ | $1.1305 \pm 0.2488$ | $1.9054 \pm 0.3865$ | $1.9274 \pm 0.4410$ |
| | Slice-wise | $1.1014 \pm 0.1852$ | $1.1226 \pm 0.2200$ | $\mathbf{1.0909 \pm 0.1510}$ | $2.0221 \pm 0.4518$ | $2.3185 \pm 0.4477$ |
| Low-rankness | Mode-wise | $6.8502 \pm 1.2101$ | $6.9333 \pm 1.2565$ | $6.9068 \pm 1.1987$ | $\mathbf{1.6311 \pm 0.3008}$ | $14.9490 \pm 1.4555$ |
| | Slice-wise | $7.1481 \pm 1.3061$ | $7.1636 \pm 1.2989$ | $7.0701 \pm 1.3004$ | $15.5770 \pm 1.3435$ | $\mathbf{1.8150 \pm 0.3227}$ |

Table 6: The average computational time with standard variance comparing nonconvex algorithm and MATLAB CVX solver.

| Structures | Methods | $d = 8 \times 8 \times 18$ | | $d = 16 \times 16 \times 16$ | |
|---|---|---|---|---|---|
| | | Iterations | Total time | Iterations | Total time |
| Entry-wise Sparse | Nonconvex (APG Algorithm) | $17.5000 \pm 0.5415$ | $\mathbf{9.5374 \pm 0.2927}$ | $27.8345 \pm 0.3892$ | $\mathbf{16.5968 \pm 0.6731}$ |
| | Convex (CVX solver) | \ | $76.0145 \pm 1.2664$ | \ | $254.2588 \pm 2.5022$ |
| Mode-wise Lowrank | Nonconvex (APG Algorithm) | $22.3333 \pm 0.3258$ | $\mathbf{12.5169 \pm 0.4709}$ | $27.9677 \pm 0.1796$ | $\mathbf{17.3974 \pm 0.5930}$ |
| | Convex (CVX solver) | \ | $77.5089 \pm 1.0035$ | \ | $229.7236 \pm 2.7010$ |

experiments on the proposed methods for each tensor structure, and the results are shown in the table.

### C.2. Real-world Data

We have chosen several real-world images from the ImageNet 2012 dataset (Russakovsky et al., 2015) besides the image used in Section 6.2. We implement experiments based on different penalties, revealing the following performance. In Figure 8, we pick one image "rabbit" from the dataset, and the denoised results are shown in the figure.

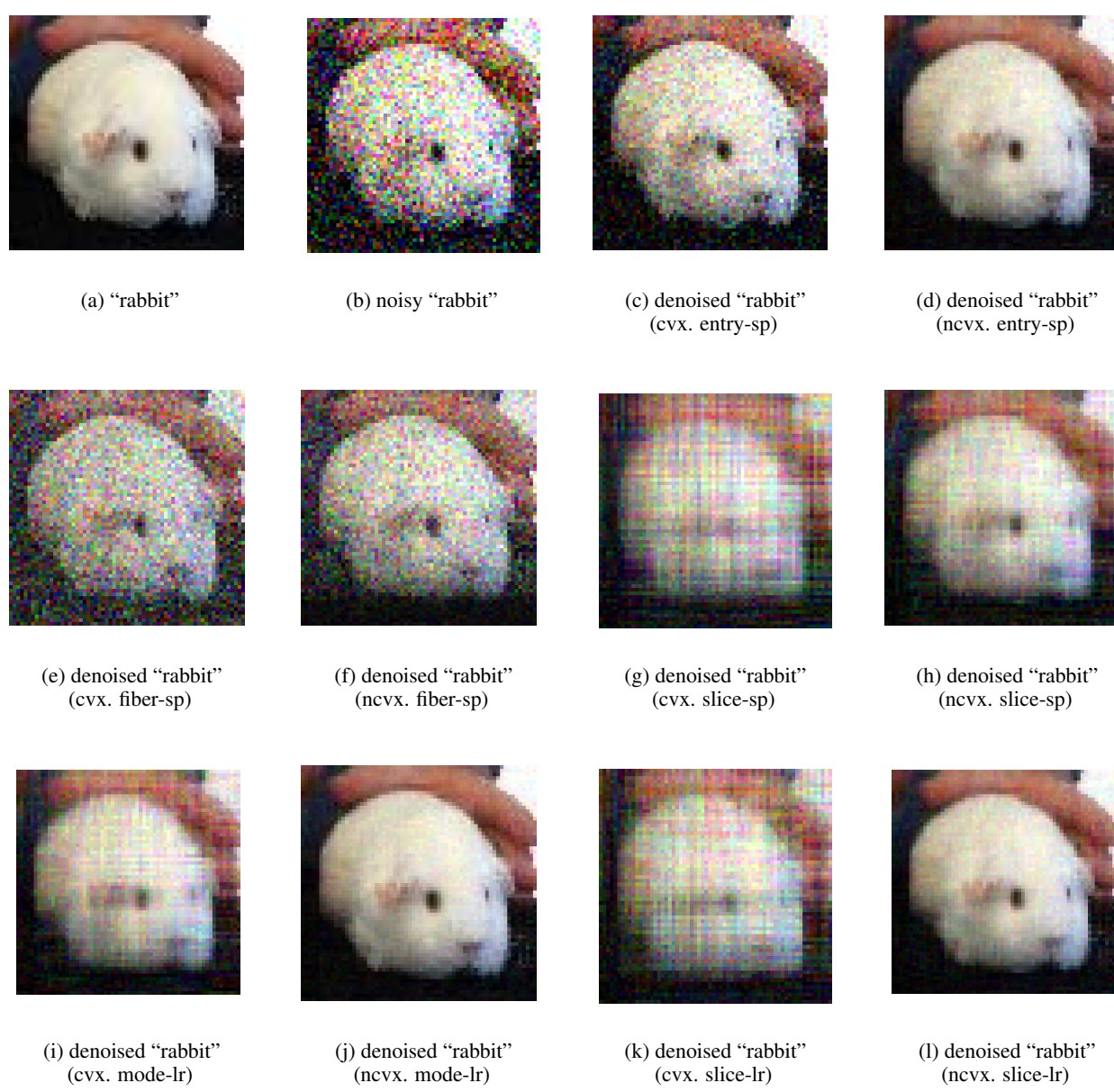

(a) "rabbit"

(b) noisy "rabbit"

(c) denoised "rabbit"
(cvx. entry-sp)

(d) denoised "rabbit"
(ncvx. entry-sp)

(e) denoised "rabbit"
(cvx. fiber-sp)

(f) denoised "rabbit"
(ncvx. fiber-sp)

(g) denoised "rabbit"
(cvx. slice-sp)

(h) denoised "rabbit"
(ncvx. slice-sp)

(i) denoised "rabbit"
(cvx. mode-lr)

(j) denoised "rabbit"
(ncvx. mode-lr)

(k) denoised "rabbit"
(cvx. slice-lr)

(l) denoised "rabbit"
(ncvx. slice-lr)

Figure 8: The denoising results with the fiber-wise sparsity regularizer.

We have also implemented additional real-world data experiments with the proposed methods. In Table 7, the real data is considered the tensor to be estimated. Regarding the initialization of the covariate tensors $\mathcal{A}$ in the real-data experiments, the number of covariate tensors $\mathcal{A}$ corresponds to the sample size $n = 5000$, and the noise term $\mathcal{E}$ are drawn independently from a Gaussian distribution with mean $0$ and variance equal to $\eta = 0.01$.

Table 7: The MSFE of the climate data (10 years) observation and alcoholic genetic predisposition data with our proposed methods

| Dataset | Penalties | Sparsity | | | Low-rankness | |
|---------|-----------|----------|--------|-------|--------------|-------|
| | | Entry | Fiber | Slice | Mode | Slice |
| NA-1990-2002-Monthly | SCAD | **7.4556 ± 0.8235** | 8.0716 ± 0.9456 | 8.4187 ± 0.9491 | **11.6809 ± 2.0437** | **9.9750 ± 1.2203** |
| | MCP | 7.6087 ± 1.0003 | **7.9554 ± 0.9884** | **8.1305 ± 0.9050** | 12.6281 ± 2.1882 | 10.2314 ± 2.0015 |
| | Convex | 8.2502 ± 1.4887 | 8.7425 ± 1.7264 | 9.4577 ± 1.3004 | 14.5808 ± 3.3435 | 13.4508 ± 2.6688 |
| EEG Database | SCAD | **12.6865 ± 2.3544** | **13.8878 ± 2.2412** | **14.5640 ± 2.4898** | 18.7983 ± 5.0977 | **16.3202 ± 4.8331** |
| | MCP | 13.0024 ± 1.9973 | 14.0368 ± 2.0241 | 16.0431 ± 3.9314 | **18.4546 ± 4.8020** | 16.4002 ± 4.7771 |
| | Convex | 13.8001 ± 2.6764 | 14.5716 ± 2.1379 | 16.8890 ± 4.3051 | 19.6404 ± 5.3317 | 18.5783 ± 5.3854 |

The experimental data employed in this study were sourced from the University of Southern California's Viterbi School of Engineering repository and the UCI Machine Learning Repository's EEG Database. Specifically, the datasets can be accessed via the following links: https://archive.ics.uci.edu/dataset/121/eeg+database, https://viterbi-web.usc.edu/~liu32/data.html.

NA-1990-2002-Monthly is a monthly climatological dataset (size of $22 \times 19500$) that includes monthly observations of time series data of 18 climate agents in 125 locations in North America. The original data size for one location in 12 years is a $22 \times 156$ matrix. To fit our model, we segment it into twelve $22 \times 13$ matrices and merge them into a $22 \times 13 \times 12$ tensor to predict. The estimation results are shown in Table 7.

EEG Database is an alcoholic genetic predisposition dataset that contains the EEG images of 64 channels sampled at 256 Hz for 77 subjects suffering from alcoholism and 44 normal controls. In the dataset, there are 10 alcoholic subjects, and each sample is a third-order tensor (Channels× Time × Voltage). We take each sample as the tensor to estimate, and the result is revealed in Table 7.

