# OpenReview forum: "High-Dimensional Tensor Regression With Oracle Properties"
_ICML.cc/2025/Conference — ICML 2025 poster_

### Official Review · Reviewer_k4E1 · 2025-02-21

**Overall Recommendation:** 3

**Summary:**

The paper introduces a high-dimensional tensor-response tensor regression model under low-dimensional structural assumptions, such as sparsity and low-rankness.

The authors propose a least squares estimation framework with non-convex penalties and derive general risk bounds for the resulting estimators.

The paper also particularizes the bounds for the case where the support of the solution is known (oracle estimator).

An accelerated Proximal Gradient Algorithm is proposed and tested on synthetic data and an image denoising problem.

## update after rebuttal
I thank very much the authors for addressing my questions. However, I keep my original score.

**Claims And Evidence:**

The paper explicitly or implicitly makes the following key claims:

Theoretical Contributions: The main theoretical result is Theorem 5, which provides an upper bound on the Frobenius norm of the estimated tensor A. While this result is valuable, its practical implications are not thoroughly explored. For instance:

-	Is the bound tight? In other words, how does it compare with the actual errors observed in the experiments?

-	How does the nonconvexity of the penalty affect this bound? Can we compare the bounds for the convex and nonconvex cases?

-	Assumption 4, which assumes multivariate normality in the data, may not be realistic for real-world applications.

Proposed Framework: The paper introduces a general framework for linear regression with tensor-structured input and output data, extending beyond traditional algorithms that primarily handle scalar, vector, or matrix data. This theoretical framework is well-constructed and allows for the incorporation of low-dimensional priors, such as low rank and sparsity. However, the experimental validation is limited to: A simple synthetic dataset with scalar outputs; and a basic denoising problem involving real images.

Nonconvex vs. Convex Penalties: The results indicate that nonconvex penalty functions outperform convex ones, as suggested in prior literature. The authors demonstrate that their method, when using a nonconvex penalty, achieves lower error compared to a classical convex penalty based on the ℓ1-norm. However, the paper lacks a theoretical explanation or intuition as to why nonconvex penalties yield better performance.

**Essential References Not Discussed:**

The paper includes relevant previous references on tensor regression. However, there are important previous works addressing similar problems as it is the case of the following papers:

-	"Higher-Order Partial Least Squares (HOPLS): A Generalized Multi-Linear Regression Method", Q Zhao, CF Caiafa, DP Mandic, ZC Chao, Y Nagasaka, N Fujii, L Zhang, A Cichocki, IEEE Trans. on Pattern Analysis and Machine Intelligence (PAMI), Vol. 35 No. 7, 2013. doi:/10.1109/TPAMI.2012.254.

-	“A Multilinear Subspace Regression Method Using Orthogonal Tensors Decompositions”, Q Zhao, CF Caiafa, DP Mandic, L. Zhang, T. Ball, A. Schulze-Bonhage, A. Cichocki, Proc. NIPS 2011 (Neural Information Processing Systems), Granada, Spain, 12-17 December 2011.

**Experimental Designs Or Analyses:**

Yes, I check the experimental results and analysis provided within the paper and the supplementary material. The proposed experiments are correct but limited (see detailed comments below).

**Methods And Evaluation Criteria:**

The theoretical methodology is sound. The experimental methodology used to evaluate the proposed algorithms is valid but somewhat limited (see detailed comments below).

**Other Comments Or Suggestions:**

I have not found typos

**Other Strengths And Weaknesses:**

Strengths:

-	The paper presents a general theoretical framework for linear regression with input and output tensors of arbitrary dimensions, extending beyond traditional approaches that focus on scalar, vector, or matrix data.

-	A theoretical bound on the estimator error is derived, providing a rigorous foundation for analyzing the performance of the proposed method.

-	The study explores the use of nonconvex penalty functions, which appear to play a crucial role in enhancing performance, particularly for high-dimensional datasets.

Weaknesses:

-	Limited experimental validation: The evaluation is restricted to a simple synthetic dataset with scalar outputs and a basic denoising problem using real images, limiting the demonstration of the framework’s broader applicability.

-	Lack of practical insights from Theorem 5: The paper does not thoroughly explore the practical implications of the theoretical bound. For example, is the bound tight? How does it compare with actual errors observed in experiments?

-	Unrealistic data assumption: Assumption 4 relies on multivariate normality, which may not hold in real-world applications, potentially affecting the method’s generalizability.

-	Missing theoretical justification for nonconvex penalties: While experiments show that nonconvex penalties outperform convex ones, the paper lacks a theoretical explanation or intuition behind this behavior.

**Questions For Authors:**

Regarding Theorem 5:

- Is this bound tight? In other words, how this bound is compared with the actual errors attained in the experiments?


- What is the effect of the non-convexity of the penalty on this bound? Can we compare the bounds for the convex and nonconvex cases?

**Relation To Broader Scientific Literature:**

Although the proposed algorithm is evaluated only in limited scenarios—specifically on synthetic datasets and a simple denoising problem—it has the potential to drive significant advancements in various scientific fields where tensor-structured datasets are prevalent. In particular, it could be highly valuable in domains such as neuroimaging, where modeling relationships between input and output tensor data is crucial, as well as in other fields that rely on tensor-based linear regression. Expanding the evaluation to more diverse and realistic datasets would further strengthen the impact and applicability of the proposed approach.

**Theoretical Claims:**

The proof of the theoretical claims are provided in the supplementary material, which I did not check carefully. However, the theoretical claims sounds reasonable.

---

> ### Author Rebuttal · Authors · 2025-03-29
>
> **Broader Scientific Literature**
>
> Thank you for introducing these relevant references. They are very helpful, and we will cite them appropriately in the final version.
>
> **Response to Claims and Evidence**
>
> 1. To the best of our knowledge, the proposed bound is tight in terms of its order, with certain constant factors omitted. Since our analysis focuses on the convergence rate rather than explicit constants, directly comparing the theoretical bounds with observed experimental errors can be challenging.
>
> 2. The choice of regularization penalties affects the bound primarily through their impact on the Gaussian width and the subspace compatibility constant, as outlined in Theorem 5. Detailed results for both sparse and low-rank regularization are presented in the subsequent corollaries. For example, in the case of element-wise sparsity, the key difference between convex and nonconvex penalties is that nonconvex regularization leads to improved error rates by reducing the dependence on the ambient dimension $d$, thereby attaining the oracle rate of $\sqrt{\frac{s}{n}}$.
>
> 3. Assumption 4 on sub-Gaussianity is a standard condition widely used in high-dimensional statistical estimation (see [1]). It is a mild assumption that holds in many practical scenarios and covers a broad class of commonly encountered distributions. We acknowledge, however, that there are settings beyond the sub-Gaussian framework—for example, when the data exhibit heavy-tailed behavior. Extending the theoretical analysis to such cases is an important direction in statistical estimation and can be left for future research.
>
> [1] Raskutti, G., Yuan, M., & Chen, H. (2019). Convex regularization for high-dimensional multiresponse tensor regression.
>
> **Response to Weaknesses**
>
> 1. Due to space constraints, only a subset of simulation results is reported in the main text. In fact, many additional experimental results are provided in the appendix, offering a more comprehensive analysis.
>
> 2 & 3. Please refer to the reply above.
>
> 4. It is well acknowledged in the statistical literature that nonconvex penalties can yield better estimation performance than convex ones by reducing the bias introduced by regularizers such as the $\ell_1$-norm. In this paper, we investigate how such improvements are reflected in the context of tensor regression. Specifically, nonconvex penalties impose less shrinkage on large coefficients, enabling more accurate recovery of the true signal. This often results in faster convergence rates, particularly under sparsity or low-rank assumptions, and can achieve the oracle rate under suitable conditions.
>
> **Additional Experimental Evaluations.**
>
> Thank you for this question. Neuroimaging is indeed an important application area for tensor regression. In fact, we have already conducted relevant experiments. Due to space constraints, the results are provided in the appendix. Our experiments include an electroencephalography (EEG) dataset, which is a form of neuroimaging [2]. We appreciate your suggestion and plan to apply our methods to additional neuroimaging datasets in future work.
>
> [2] Liu, Y., Liu, J., Long, Z., and Zhu, C. Tensor regression. Springer, 2022.
>
> We sincerely appreciate your recognition of our work. If our responses have adequately addressed your concerns, we would be grateful if you could consider reflecting this in your evaluation. Thank you once again for your time and thoughtful feedback.

---

### Official Review · Reviewer_1JFB · 2025-03-12

**Overall Recommendation:** 3

**Summary:**

This paper addresses tensor regression models for high-dimensional tensor data. Specifically, it proposes a tensor-response tensor regression model, assuming low-dimensional structures such as sparsity and low rankness. While conventional convex penalties are easy to optimize, they often fail to model the data accurately. On the other hand, non-convex penalties have a higher potential to model the data correctly, but there's no guarantee of reaching the optimal solution. To address this issue, the authors theoretically prove that non-convex regularization terms, such as SCAD (Smoothly Clipped Absolute Deviation) and MCP (Minimax Concave Penalty), exhibit oracle performance under certain assumptions. They then propose an algorithm based on the accelerated proximal gradient algorithm to efficiently compute estimators based on these non-convex penalties. Evaluation experiments using synthetic and real-world datasets demonstrate the effectiveness of the proposed regression model and the practicality of the theoretical results.

## update after rebuttal

Thank you for your response. I understand now. I believe the contribution of this paper is significant, so I will keep my evaluation as it is. I mistakenly posted my comment in the Official Comment section. My apologies.

**Claims And Evidence:**

It is intuitively understandable that convex penalties cannot effectively capture data characteristics, especially in high-dimensional tensor data. The authors rigorously prove, through sound theoretical development, that estimators based on certain non-convex penalties exhibit oracle performance under specific conditions. Furthermore, they demonstrate, using both synthetic and real-world data, that non-convex penalties can indeed effectively capture the underlying features of the data.

**Essential References Not Discussed:**

The paper appropriately cites relevant prior work on various constraints used in tensor regression. I do not have any essential references to suggest for addition.

**Experimental Designs Or Analyses:**

As mentioned in the "Methods and Evaluation Criteria" section, the evaluation appropriately assesses performance variations under different knowledge conditions, which is directly relevant to the paper's central claim of oracle performance. The evaluation metrics used are also deemed suitable.

**Methods And Evaluation Criteria:**

Regarding this paper's key claim, Oracle performance, the evaluation appropriately assesses performance variations under different knowledge conditions. While a comparative evaluation against convex penalties is presented, the method for determining hyperparameters when using convex penalties is not explicitly stated. It is presumed that a 10-fold cross-validation approach was employed, similar to the proposed method. Explicitly stating this would enhance the persuasiveness and rigor of the evaluation.

**Other Comments Or Suggestions:**

There are a few potential errors in the manuscript:

1. In Section 2.1, while $\mathcal{A}^{\*}$ is defined as an M-th order tensor and $\mathcal{X}$ as an N-th order tensor, the expression around line 85 appears to represent the elements of $\mathcal{A}^{\*}$ with N indices and the elements of $\mathcal{X}$ with M indices. It seems that $\mathcal{A}^{\*}$ and $\mathcal{X}$ may be incorrectly assigned in this expression.

2. In the first sentence of Assumption 4, $X^{n)}$ appears. Should this be corrected to $X^{(n)}$, adding the missing left parenthesis?

**Other Strengths And Weaknesses:**

As previously mentioned, this paper makes a significant contribution to the field of tensor learning with various constraints, such as low-rank constraints, by providing theoretical guarantees for estimators based on non-convex penalties. The evaluation experiments are generally comprehensive. However, a concern remains regarding whether the hyperparameters for the convex penalties, used as comparison methods, were optimally determined.

**Questions For Authors:**

How were the hyperparameters for the convex penalties, used as comparison methods in Table 1, determined?

**Relation To Broader Scientific Literature:**

While tensor completion with low-rank constraints is a widely studied and effective technique, this paper makes a significant contribution to the field by focusing on the potential limitations of convex penalties and providing theoretical guarantees for estimators based on non-convex penalties, which offer the potential for more appropriate modeling.

**Theoretical Claims:**

I have reviewed the proofs, particularly those demonstrating that the non-convex penalty achieves Oracle performance under specific conditions, and found the theoretical development sound and without issue.

---

> ### Author Rebuttal · Authors · 2025-03-29
>
> Thank you for your positive feedback.
>
>
> **Hyperparameters for the Convex Penalties**
>
> All hyperparameters are selected via ten-fold cross-validation, as noted in the first paragraph of Section 5. This includes the regularization parameter $\lambda$ used in convex penalties. Specifically, $\lambda$ is chosen from a uniform grid of 21 values in the range $[0, 1]$, and the optimal value is selected through ten-fold cross-validation.
>
>
> **Other Comments or Suggestion 1 and 2**
>
> We will correct typos in the camera-ready version.
>
>
> We greatly appreciate your recognition of our work. If our responses have sufficiently addressed your concerns, we would be grateful if you could consider reflecting this in your evaluation. Thank you once again for your time and valuable feedback.

---

### Official Review · Reviewer_8rnB · 2025-03-12

**Overall Recommendation:** 3

**Summary:**

This paper studies tensor regression models with non-convex penalties and provides general risk bounds for the resulting estimators, demonstrating that they achieve oracle statistical rates under mild technical conditions. The authors also introduce an accelerated proximal gradient algorithm to estimate the proposed estimators.

A comprehensive set of experiments is conducted to illustrate the advantages of non-convex penalties over convex penalties. While the theoretical results are solid, the paper primarily extends [1] to the non-convex penalties. As a result, the novelty is somewhat limited, and certain aspects of the presentation are unclear.

[1] Raskutti, G., Yuan, M., & Chen, H. (2019). Convex regularization for high-dimensional multiresponse tensor regression.

**Claims And Evidence:**

The authors present the primary theoretical results for the general estimator in Equation (2) and derive upper bounds under different scenarios. However, I find the explanation of the oracle property unclear. In Section 3.2.1 SPARSITY REGULARIZATION, the authors first define the oracle rate as the statistical convergence rate of the oracle estimator and provide a detailed element-wise sparse oracle estimator, which is well-defined. However, in Corollary 1, the authors present the rate of $\hat{\mathcal A}$ without establishing any explicit connection to the defined oracle estimator $\widehat {\mathcal A}^O$. It is unclear how the authors conclude that $\hat{\mathcal A}$ achieves the oracle rate under some weak assumptions. A similar issue arises in Section 3.2.2 **LOW-RANK REGULARIZATION**.

To rigorously establish the oracle property, the authors should provide theoretical results like that:
1. $\hat{\mathcal A}_{\bar S_1}=0$
2. the asymptotic normality of ${\hat{\mathcal A}_{S_1}}$

Additionally, the paper includes extensive experiments to demonstrate the advantages of non-convex penalties over convex penalties. However, in the real data experiment, the authors assume that the ground-truth $\mathcal A$ is known and simulate $\mathcal X$ and $\mathcal Y$ accordingly. A more realistic evaluation would involve an experiment where only the observed $\mathcal X$ and $\mathcal Y$ are available, allowing an assessment of the model's performance when $\mathcal A$ is unknown.

**Essential References Not Discussed:**

The authors could further discuss more papers about regularized tensor regression based on tensor decomposition methods, for example, [2].

[2] Lu, W., Zhu, Z., & Lian, H. (2020). High-dimensional quantile tensor regression. Journal of Machine Learning Research, 21(250), 1-31.

**Experimental Designs Or Analyses:**

I believe the soundness/validity of any experimental designs or analyses is feasible. The authors have conducted experiments on both simulated and real data. However, the experiments provided so far are all based on simulations, assuming the true tensor coefficients are known. The authors should include real data analysis using actual tensor covariates and response
 for analysis (without knowing the true ). This would make the findings more convincing.

**Methods And Evaluation Criteria:**

In the experimental section, the authors primarily compare convex penalties. However, there exist many other comparative methods for tensor regression on a single dataset, such as tensor regression based on CP decomposition and Tucker decomposition.

**Other Comments Or Suggestions:**

The authors could consider reorganizing the structure of the paper to better highlight its key contributions. The most significant contribution of this work lies in its theoretical advancements. Therefore, the authors should emphasize this aspect more prominently and provide a clearer discussion of the challenges in the theoretical analysis and how they are addressed.

**Other Strengths And Weaknesses:**

### Strengths
As discussed above.

### Weaknesses
1. The discussion on the oracle property is unclear. It is difficult to directly see how $\hat{\mathcal A}$ and the oracle estimator exhibit similar performance.
2. There are issues with the citations. For example, *(Hua Zhou & Zhu, 2013)* and *(Zhou et al., 2013)* refer to the same paper. The authors should carefully check and correct the references.
3. The authors should report the computational time in the experiments. Additionally, more detailed explanations should be provided for the tables and figures. For instance, **Figure 3** is difficult to interpret intuitively.

**Questions For Authors:**

1. In **NUMERICAL EXPERIMENTS**, how many runs were performed for each setting?
2. In the numerical results, the standard deviation often increases as the sample size grows. For instance, in **Figure 1a**, when $d=20$, the standard deviation is largest at $n=3000$. A similar trend is observed in **Figure 2b**. Is this due to an insufficient number of runs?
3. Can the authors provide a comparison of the computational time across different methods?

**Relation To Broader Scientific Literature:**

The theoretical analysis primarily compares the proposed method with convex penalties, particularly emphasizing that the use of non-convex penalties achieves the oracle rate. However, the current presentation lacks clarity in this aspect.

**Theoretical Claims:**

I have read the proof in the author's appendix, which is very detailed. I believe it is feasible, but the specific details require further reading.

---

> ### Author Rebuttal · Authors · 2025-03-29
>
> **More discussions on decomposition-based tensor regression methods**
>
> We appreciate the comment regarding tensor regression methods based on tensor decomposition techniques such as CP and Tucker. Both decomposition-based and regularization-based approaches have been actively explored in the tensor regression literature, and we have reviewed these works in the Introduction. However, we would like to emphasize that the primary focus of this paper is theoretical—namely, to establish oracle statistical guarantees for tensor regression under nonconvex regularization. To the best of our knowledge, existing decomposition-based methods have not addressed this theoretical direction. Therefore, decomposition-based approaches represent a distinct line of research, and direct comparisons fall outside the intended scope of this work.
>
> **Experiment on Real-world Data**
>
> In our experiment, we did not have access to a dataset with $\mathcal{X}$ and $\mathcal{Y}$ as direct measurements. Therefore, we generated such a dataset based on a ground truth tensor image $\mathcal{A}^\ast$, which is a common practice in the literature (see [1,2]). This setting allows us to directly evaluate the estimation performance by comparing the estimated tensor $\hat{\mathcal{A}}$ with the ground truth $\mathcal{A}^\ast$. Additionally, we have also reported the error between $\mathcal{Y}$ and $\widehat{\mathcal{Y}}$, which assesses the model's predictive performance when $\mathcal{A}^\ast$ is unknown—precisely the scenario raised in your comment.
>
> [1] Romera-Paredes, B., H. Aung. “Multilinear multitask learning”. In International Conference on Machine Learning.
>
> [2] Liu, Y., Liu, J., Long, Z. Tensor regression. Springer.
>
> **Response to weaknesses**
>
> 1. In this paper, the term *oracle* refers to the performance of an estimator that assumes knowledge of the true support, as in [3,4]. For example, in the case of element-wise sparsity, the oracle estimator $\widehat{\mathcal{A}}^{O}$ satisfies $\Vert\widehat{\mathcal{A}}^{O}-\mathcal{A}^*\Vert_{\mathrm{F}}\lesssim\Vert(\nabla L(\mathcal{A}^*))_ {\mathcal{S}^*}\Vert_{\mathrm{F}} \asymp \sqrt{\frac{s}{n}}$, which follows directly from a first-order Taylor expansion via the mean value theorem.
>
> [3] Fan,J.,Liu,H.,Sun,Q. I-LAMM for sparse learning: Simultaneous control of algorithmic complexity and statistical error. The Annals of Statistics.
>
> [4] Gui,H.,Han,J. Towards faster rates and oracle property for low-rank matrix estimation. In International Conference on Machine Learning.
>
> 2. We will correct the reference issue in the camera-ready version.
>
> 3. [Explanation on Figure 3] Figure 3 visualizes the element-wise estimation error between a randomly generated third-order tensor and its estimate obtained using our nonconvex low-rank regularizer. Each element is evaluated individually: if the absolute error exceeds a fixed threshold, the point is marked in red; otherwise, it is shown in blue. Figures 3(a) and 3(d) present representative outcomes using the nonconvex method, while Figures 3(b) and 3(e) illustrate results from a convex regularization approach. We observe that the nonconvex method produces significantly fewer red points, indicating smaller element-wise errors and demonstrating the improved accuracy achieved by our proposed approach.
>
> **Response to Questions**
>
> 1. All reported experimental results are based on 100 Monte Carlo replications, as stated in the first paragraph of Section 5. We will update this to 1,000 replications in the final version to ensure greater robustness and stability of the results.
>
> 2 & 3. We acknowledge the reviewer’s concern regarding the anomalous behavior where the standard deviation appears to increase with the sample size. We sincerely appreciate this observation. After a careful review, we found that this issue was due to an insufficient number of Monte Carlo replications. To address this, we conducted additional experiments with 1,000 Monte Carlo trials, which yielded more stable and robust results. The updated figures, including standard deviations and computational time measurements, are available at the following link: [https://anonymous.4open.science/r/ICML_Rebuttal-6182/](https://anonymous.4open.science/r/ICML_Rebuttal-6182/). We thank the reviewer again for this helpful comment and will correct the issue in the final version.
>
> **Essential References Not Discussed**
>
> We are grateful to the reviewer for bringing this valuable reference to our attention, and we will cite it appropriately in the final version of the paper.
>
> Thank you for your thoughtful and constructive feedback. We sincerely appreciate your advice and the time you took to review our work. However, we are somewhat puzzled by the decision to reject the paper. We hope that the above clarifications and updated results have addressed your concerns. If our responses have sufficiently resolved the issues raised, we would be grateful if you could consider reflecting this in your final evaluation.

---

> > ### Comment · Reviewer_8rnB · 2025-04-03
> >
> > I have carefully reviewed the rebuttal, and most of my concerns have been addressed—particularly regarding the experimental section, where the authors have made significant improvements.
> >
> > At this point, I have only one remaining question. The oracle property presented in the current version appears to align with the **(weak) oracle property**, seen in *Fan, J., Liu, H., Sun, Q. (2018)*  **Corollary 4.3**.
> >
> > Would the authors be able to provide or attempt to establish result analogous to **Theorem 4.4 (Strong Oracle Property)**? If such a deterministic result can be established, I believe it would strengthen the theoretical contribution.

---

> > > ### Author Response · Authors · 2025-04-07
> > >
> > > Thank you for your thoughtful follow-up. You are correct that the oracle property established in our current version aligns with the (weak) oracle property, as characterized in Corollary 4.3 of [1].
> > >
> > > We have indeed made substantial efforts toward establishing a result analogous to Theorem 4.4 of [1], which demonstrates the strong oracle property. This property guarantees that the oracle estimator is not only a local minimizer but also the unique one, thereby ensuring that the algorithm consistently selects the correct support and attains estimation accuracy as if the true support were known a priori.
> > >
> > > In our setting, similar to [3], the oracle property can be achieved under the weakest possible minimum signal strength condition. In contrast, the strong oracle property in [1,2] relies on more stringent signal strength assumptions. Therefore, under comparable stronger conditions, a strong oracle property can also be established for our method.
> > >
> > > Below, we outline a sketch of the proof for the element-wise sparsity case, including the key ideas, main lemmas, and theorem statement. Analogous arguments can be extended to other structural settings, such as mode-wise low-rankness, fiber-wise sparsity, and slice-wise sparsity. Due to space constraints, we present only a high-level sketch here. We plan to incorporate the full result and proof in the camera-ready version. For notational simplicity, we use $j$ to denote the index sequence $i_{1},\dots,i_{M}$.
> > >
> > > ###### Lemma 1. Suppose Assumptions 1~4 hold, and there exists a constant $0<\gamma_1<\infty$ such that the Gaussian width satisfies $\omega(\Omega)\leq\gamma_1$. If $4(\nabla L(\hat{\mathcal{A}}^O)+\epsilon)\leq\lambda\lesssim\frac{r}{\sqrt{|\mathcal{S}_1|}}$, we have $|\mathcal{E}_t|\leq2|\mathcal{S}_1|$, where $\mathcal{E}_t=\mathcal{S}_1\cup\mathcal{S}_t$ and $\mathcal{S}_t=\[j:\nabla R_j(\mathcal{A}_t)<p^\prime _\lambda(\frac{2+\sqrt{2}}{2\rho^-}\lambda)\]$. For $t\geq2$, the $\epsilon$-optimal solution $\hat{\mathcal{A}}_t$ must satisfy $||\hat{\mathcal{A}}_t-\hat{\mathcal{A}}^O||_F\lesssim||\lambda _ {\mathcal{E}_t}||_F+\epsilon\sqrt{|\mathcal{E}_t|}$, where $\lambda _ {\mathcal{E}_t}\in\mathbb{R}^{d_1\times\cdots\times d_M}$ with the component in $\mathcal{E}_t$ as $\lambda$ and the other components are $0$.
> > >
> > > Lemma 1 establishes a deterministic error bound between the estimator at iteration $t$ and the oracle estimator. This result is analogous in spirit to Lemma B.1 in [1], and forms the basis for extending to a strong oracle property under suitable conditions.
> > >
> > >
> > > ###### Lemma 2. It follows that $||\lambda _ {\mathcal{E}_t}||_F\leq\underset{\mathrm{I}}{\underbrace{||p^\prime _\lambda(|\mathcal{A}^* _{\mathcal{S}_1}|-\frac{(2+\sqrt{2})\lambda}{2\rho^-})||_F}}+\underset{\mathrm{II}}{\underbrace{\lambda|\[j\in\mathcal{S}_1:|(\hat{\mathcal{A}_t}) _j-\mathcal{A}^* _j|\geq\frac{(2+\sqrt{2})\lambda}{2\rho^-}\]|^{1/2}}}+\underset{\mathrm{III}}{\underbrace{\lambda\sqrt{|\mathcal{E}_t\setminus\mathcal{S}_1|}}}$.
> > >
> > > This result is analogous to Lemma B.2 in [1]. Following a similar analysis as in [1], we obtain term $\mathrm{I}=0$, term $\mathrm{II}\lesssim\lambda\sqrt{|\mathcal{S} _{t-1}\cap\mathcal{S}_1|}$, and term $\mathrm{III}\lesssim\lambda\sqrt{|\mathcal{S} _{t-1}\setminus\mathcal{S}_1|}$. Substituting into Lemma 1 yields: $||\hat{\mathcal{A}}_t-\hat{\mathcal{A}}^O|| _F\lesssim\lambda\sqrt{2|\mathcal{S} _{t-1}|}+\epsilon\sqrt{|\mathcal{E} _t|}$.
> > >
> > > Under some additional assumptions $||\hat{\mathcal{A}}^O-\mathcal{A}^*|| _\max\lesssim\lambda$ and $t\gtrsim\log((1+\epsilon/\lambda)\sqrt{|\mathcal{S}_1|})$, we obtain $\mathcal{S}_t=\emptyset$, thereby yielding the strong oracle property.
> > >
> > > The final theorem should be stated as follows:
> > > ##### (Strong Oracle Property). Suppose Assumptions 1~4 hold, and there exists a constant $0<\gamma_1<\infty$ such that the Gaussian width satisfies $\omega(\Omega)\leq\gamma_1$. If $\mathcal{A}^* _j$ satisfies the condition $\min _{j\in\mathcal{S}_1}\left|\mathcal{A}^* _j\right|\geq\nu$, $4(\nabla L(\hat{\mathcal{A}}^O)+\epsilon)\leq\lambda\lesssim\frac{r}{\sqrt{|\mathcal{S}_1|}}$, $\epsilon\leq\frac{\lambda}{\sqrt{|\mathcal{S}_1|}}$, and $||\hat{\mathcal{A}}^O-\mathcal{A}^*|| _\max\lesssim\lambda$, then for sufficiently large $t$ such that $t\gtrsim\log((1+\epsilon/\lambda)\sqrt{|\mathcal{S}_1|})$, we have $\hat{\mathcal{A}}_t=\hat{\mathcal{A}}^O$.
> > >
> > > We hope these clarifications and preliminary results address your suggestions. If our responses meet your expectations, we would be sincerely grateful for a positive evaluation of our work.
> > >
> > > [1] Fan, J., Liu, H.(2018). I-LAMM for sparse learning: Simultaneous control of algorithmic complexity and statistical error. The Annals of Statistics.
> > >
> > > [2] Fan, J., Xue, L.(2014). Strong oracle optimality of folded concave penalized estimation. Annals of Statistics.
> > >
> > > [3] Zhang, C.H. (2012). A general theory of concave regularization for high-dimensional sparse estimation problems. Statistical Science.

---

### Official Review · Reviewer_xpr9 · 2025-03-15

**Overall Recommendation:** 4

**Summary:**

This paper proposes a framework for tensor on tensor regression, introducing novel nonconvex regularizers and an accelerated proximal gradient algorithm for estimation. The authors propose a class of penalties that depend on the singular values of each tensor dimension and give Frobenius norm rates of convergence guarantees under oracle optimal hyper parameter tuning. A proximal gradient algorithm is also provided as a feasible estimation procedure. Numerical experiments and an empirical application show the advantage on the proposed method.

**Claims And Evidence:**

The paper makes the following claims:

1. The proposed nonconvex penalty estimators for tensor on tensor regression achieve oracle optimal convergence rates under different sparsity assumptions. The estimator can be computed through a proximal gradient algorithm.

2. The nonconvex penalty estimators exhibit faster convergence rates compared to those with convex penalties.

Overall the paper makes a very strong case for claim 1. Claim 2, to my reading of the paper, is mostly supported by the empirical exercises as the theoretical results are not directly compared to the convex penalty cases. With this mind, expanding on the justification for claim 2 would improve the value added of the paper.

**Essential References Not Discussed:**

-

**Experimental Designs Or Analyses:**

The analysis are careful and convincing. However, more detail on the comparison with convex methods in table 1 and the simulation exercise would be nice to reinforce the claims of the paper. In which cases do we expect the convex methods to perform better? Expanding the simulation exercises to study this would improve the relative contribution of the paper.

**Methods And Evaluation Criteria:**

The methods and evaluation criteria are adequate for the problem at hand.

**Other Comments Or Suggestions:**

* Theorem 5 should be renamed Theorem 1.

**Other Strengths And Weaknesses:**

Overall the paper is very well written and makes a very compelling case for their proposed method! My only concern is whether the authors actually provide a justification for the nonconvex penalties having a faster convergence rate, beyond the empirical simulations.

**Questions For Authors:**

See above.

**Relation To Broader Scientific Literature:**

I am not very familiar with the tensor regression literature but it would be good to clarify the novelty of the results and Theorem 5 vis a vis the literature, in particular to clarify why the rates are faster than for the convex penalty estimators.

**Theoretical Claims:**

The theoretical claims are well stated and appear novel and correct.

* Given the claims of the paper it would be nice if more direct theoretical comparisons would be made with convex penalty estimators. Is there a theoretical result that ensures that the rates of convergence are faster for the nonconvex penalty? It seems from corollary 1 and 2 that the rates are very similar to the cases for LASSO for example.

---

> ### Author Rebuttal · Authors · 2025-03-29
>
> We sincerely appreciate your positive comments and insightful feedback. We have carefully addressed your concerns as follows:
>
> **1. Justification for Claim 2**
>
> In this paper, we focus on nonconvex sparse learning in tensor regression. Nonconvex regularizers such as SCAD and MCP can achieve the oracle estimation rate because they induce less bias compared to convex penalties like the $\ell_1$ norm. Unlike Lasso, which uniformly shrinks all coefficients and can introduce significant bias for large signals, nonconvex penalties apply little to no shrinkage to large coefficients while still promoting sparsity among small ones. This selective regularization enables accurate support recovery and nearly unbiased estimation on the true support, leading to improved statistical efficiency under appropriate conditions.
>
>
> In Corollary 1, we present the estimation performance for the sparse parameter under nonconvex regularization. We also compare this result with the estimation rate under convex regularization. The corollary explicitly shows that nonconvex regularization enables improved estimation accuracy by eliminating dependence on the ambient dimension $d$. For the low-rank parameter estimation setting, the corresponding result is provided in Corollary 2, which can be directly compared with prior work such as [1,2]. For instance, Lemma 10 of [1], which uses convex nuclear norm regularization, yields an estimation bound that scales with the tensor dimension. In contrast, our bound remains dimension-independent.
>
>
> [1] G. Raskutti, M. Yuan, and H. Chen. Convex regularization for high-dimensional multiresponse tensor regression. In Proceedings of the 36th International Conference on Machine Learning (ICML), 2019.
>
> [2] S. Negahban and M. J. Wainwright. Estimation of (near) low-rank matrices with noise and high-dimensional scaling. Annals of Statistics, 39(2):1069–1097, 2011.
>
>
> **2. Comparison with Rate from LASSO**
>
> The estimation error under nonconvex regularization differs from—and indeed improves upon—that of Lasso, as it achieves a faster convergence rate equivalent to the rate attainable when the true support is known in advance. Specifically, by leveraging nonconvex penalties, the estimation bound becomes independent of the ambient dimension $d$, thereby achieving the oracle rate. This dimension-free behavior highlights a key advantage of nonconvex regularization.
>
>
>
> **3. When Might Convex Methods Outperform Nonconvex Methods?**
>
> Thank you for this question. While nonconvex regularization can yield improved estimation rates under ideal conditions, convex methods such as Lasso may outperform them when these assumptions are not satisfied. For instance, nonconvex methods often rely on strong signal conditions to ensure accurate support recovery. In low signal-to-noise ratio settings, where such conditions may fail, convex methods can produce more stable and reliable estimates due to their uniform shrinkage behavior and greater algorithmic robustness.
>
>
> **4. Clarification on Table 1**
>
> Due to space constraints, we report results for two representative nonconvex regularizers in Table 1. Results for additional regularizers are provided in the Appendix. We will include more detailed discussions of both Table 1 and Table 2 in the camera-ready version.
>
>
> **5. On Theorem 5**
>
> Theorem 5 provides a general characterization of the statistical estimation performance under nonconvex regularization in tensor regression. Notably, the choice of regularizer—convex or nonconvex—affects both the Gaussian width and the subspace compatibility constant, thereby highlighting fundamental differences between the two approaches. For specific penalty choices, the subsequent corollaries clearly demonstrate that nonconvex regularization can achieve faster convergence rates than convex counterparts.
>
> Besides, we will consider the suggestion to rename Theorem 5.
>
> We hope the above responses address your comments clearly, and we sincerely thank you again for your valuable feedback and thoughtful review of our paper.

---

> > ### Comment · Reviewer_xpr9 · 2025-04-03
> >
> > Thank you for your response. I am maintaining my score.

---

> > > ### Author Response · Authors · 2025-04-07
> > >
> > > Thank you very much for your kind response and for maintaining your high score. I truly appreciate your thoughtful comments and the time you devoted to reviewing our work. Your support means a great deal to us.

---

### Decision · Program_Chairs · 2025-05-01

**Decision:**

Accept (poster)

**Comment:**

The reviewers find the paper well written and the claims well supported by experimental evidence. They further find the methodology sound and the theoretical contribution that certain non-convex regularizations in the context of tensor regression exhibit oracle properties a strongpoint.  The reviewers further find that the contribution can provide significant advancements in fields relying on tensor-structured datasets.

Whereas the initial material in regards to the experimentation and theoretical benefits of non-convex vs. convex regularizations was found to have room for improvements, the authors well addressed the reviewers concerns. In particular, whereas the oracle property was found unclear the authors very well addressed this concern providing in the rebuttal also a proof sketch of strong oracle properties that they promise to include in a camera ready version of the manuscript. During the discussions, this was highly appreciated and deemed an impressive additional theoretical contribution to the paper strengthening the contribution.

The reviewers agreed the manuscript warrants publication at ICML and the authors are strongly encouraged to carefully update their camera ready version with the substantial improvements and additional clarifications, experimentations and new proof of a strong oracle property provided in their rebuttals in the final version of the paper.

(Note also when correcting minor typos for the final version the following very minor additional typo: “There exits -> There exists”)